# CONSISTENT ALGORITHMS FOR MULTI-LABEL CLASSIFICATION WITH MACRO-AT-$k$ METRICS

**Erik Schultheis**
Aalto University
Helsinki, Finland
erik.schultheis@aalto.fi

**Wojciech Kotłowski**
Poznan University of Technology
Poznan, Poland
wkotlowski@cs.put.poznan.pl

**Marek Wydmuch**
Poznan University of Technology
Poznan, Poland
mwydmuch@cs.put.poznan.pl

**Rohit Babbar**
University of Bath / Aalto University
Bath, UK / Helsinki, Finland
rb2608@bath.ac.uk

**Strom Borman**
Yahoo Research
Champaign, USA
strom.borman@yahooinc.com

**Krzysztof Dembczyński**
Yahoo Research / Poznan University of Technology
New York, USA / Poznan, Poland
krzysztof.dembczynski@yahooinc.com

## ABSTRACT

We consider the optimization of complex performance metrics in multi-label classification under the population utility framework. We mainly focus on metrics linearly decomposable into a sum of binary classification utilities applied separately to each label with an additional requirement of exactly $k$ labels predicted for each instance. These "macro-at-$k$" metrics possess desired properties for extreme classification problems with long tail labels. Unfortunately, the at-$k$ constraint couples the otherwise independent binary classification tasks, leading to a much more challenging optimization problem than standard macro-averages. We provide a statistical framework to study this problem, prove the existence and the form of the optimal classifier, and propose a statistically consistent and practical learning algorithm based on the Frank-Wolfe method. Interestingly, our main results concern even more general metrics being non-linear functions of label-wise confusion matrices. Empirical results provide evidence for the competitive performance of the proposed approach.

## 1 INTRODUCTION

Various real-world applications of machine learning require performance measures of a complex structure, which, unlike misclassification error, do not decompose into an expectation over instance-wise quantities. Examples of such performance measures include the area under the ROC curve (AUC) (Drummond & Holte, 2005), geometric mean (Drummond & Holte, 2005; Wang & Yao, 2012; Menon et al., 2013; Cao et al., 2019), the $F$-measure (Lewis, 1995) or precision at the top (Kar et al., 2015). The theoretical analysis of such measures, as well as the design of consistent and efficient algorithms for them, is a non-trivial task.

In multi-label classification, one can consider a wide spectrum of measures that are usually divided into three categories based on the averaging scheme, namely instance-wise, micro, and macro averaging. Instance-wise measures are defined, as the name suggests, on the level of a single instance. Typical examples are Hamming loss, precision@$k$, recall@$k$, and the instance-wise $F$-measure. Micro-averages are defined on a confusion matrix that accumulates true positives, false positives, false negative, and true negatives from all the labels. Macro-averages require a binary metric to be applied to each label separately and then averaged over the labels. In general, any binary metric can be applied in any of the above averaging schemes. Not surprisingly, some of the metrics, for example misclassification error, lead to the same form of the final metric regardless of the scheme

used. One can also consider the wider class of measures that are defined as general aggregation functions of label-wise confusion matrices. This includes the measures described above, but also, e.g., the geometric mean of label-wise metrics or a specific variant of the $F$-measure (Opitz & Burst, 2021) being a harmonic mean of macro-precision and macro-recall.

In this paper, we target the setting of prediction with a budget. Specifically, we require the predictions to be "budgeted-at-$k$," meaning that for each instance, exactly $k$ labels need to be predicted. The budget of $k$ requires the prediction algorithm to choose the labels "wisely". It is also important in many real-world scenarios. For instance, in recommendation systems or extreme classification, there is a fixed number of slots (e.g., indicated by a user interface) required to be filled with related products/searches/ads (Cremonesi et al., 2010; Chang et al., 2021). Furthermore, having a fixed prediction budget is also interesting from a methodological perspective, as various metrics which lead to degenerate solutions without a budget, e.g., predict nothing (macro-precision) or everything (macro-recall), become meaningful when restricted to predict $k$ labels per instance.

While all our theoretical results and algorithms apply to a general class of multi-label measures, we focus in this paper on macro-averaged metrics. If no additional requirements are imposed on the classifier, the linear nature of the macro-averaging means that a binary problem for each label can be solved independently, and existing techniques (Koyejo et al., 2015; Kotłowski & Dembczyński, 2017) are sufficient. In turn, if we require predictions to be budgeted-at-$k$, the task becomes much more difficult, as this constraint tightly couples the different binary problems together. In general, they cannot be solved independently for each label, requiring instead more involved techniques to find the optimal classifier.

The macro-at-$k$ metrics seem to be very attractive in the context of multi-label classification. Macro-averaging treats all the labels equally important. This prevents ignoring labels with a small number of positive examples (Schultheis et al., 2022), so-called tail labels, which are very common in applications of multi-label classification, particularly in the extreme setting when the number of all labels is very large (Jain et al., 2016; Babbar & Schölkopf, 2019). Furthermore, it can be shown that one can remove tail labels from the training set with almost no drop of performance in terms of popular metrics, such as precision@$k$ and nDCG@$k$, on extreme multi-label data sets (Wei & Li, 2019; Schultheis et al., 2023). The macro-at-$k$ metrics, on the other hand, are sensitive to the lack of tail labels in the training set.[1]

We aim at delivering consistent algorithms for macro-at-$k$ metrics, i.e., algorithms that converge in the limit of infinite training data to the optimal classifier for the metrics. Our main theoretical results are stated in a very general form, concerning the large class of aggregation functions of label-wise confusion matrices. Our starting point of the analysis are results obtained in the multi-class setting (Narasimhan et al., 2015; 2022), concerning consistent algorithms for complex performance measures with additional constraints. Nevertheless, they do not consider budgeted-at-$k$ predictions, which do not apply to multi-class classification, while they play an important role in the multi-label setting. Furthermore, using arguments from functional analysis, we managed to significantly simplify the line of reasoning in the proofs. We first show that the problem can be transformed from optimizing over classifiers to optimizing over the set of feasible confusion matrices, and that the optimal classifier optimizes an unknown *linear* confusion-matrix metric. In the multi-label setting, interestingly, such a classifier corresponds to a prediction rule, which has the appealingly simple form: selecting the $k$ highest-scoring labels based on an *affine transformation* of the marginal label probabilities. Combining this result with the optimization of confusion matrices, we state a Frank-Wolfe based algorithm that is consistent for finding the optimal classifier also for *nonlinear* metrics. Empirical studies provide evidence that the proposed approach can be applied in practical settings and obtains competitive performance in terms of the macro-at-$k$ metrics.

## 2 RELATED WORK

The problem of optimizing complex performance metrics is well-known, with many articles published for a variety of metrics and different classification problems. It has been considered for binary (Ye et al., 2012; Koyejo et al., 2014; Busa-Fekete et al., 2015; Dembczynski et al., 2017), multi-class (Narasimhan et al., 2015; 2022), multi-label (Waegeman et al., 2014; Koyejo et al., 2015; Kotłowski & Dembczyński, 2017), and multi-output (Wang et al., 2019) classification.

---

[1]Results and description of such an experiment are given in Appendix I.

Initially, the main focus was on designing algorithms, without a conscious emphasis on statistical consequences of choosing models and their asymptotic behavior. Notable examples of such contributions are the SVMperf algorithm (Joachims, 2005), approaches suited to different types of the F-measure (Dembczynski et al., 2011; Natarajan et al., 2016; Jasinska et al., 2016), or precision at the top (Kar et al., 2015). Wide use of such complex metrics has caused an increasing interest in investigating their theoretical properties, which can then serve as a guide to design practical algorithms.

The consistency of learning algorithms is a well-established problem. The seminal work of Bartlett et al. (2006) was studying this problem for binary classification under the misclassification error. Since then a wide spectrum of learning problems and performance metrics has been analyzed in terms of consistency. These results concern ranking (Duchi et al., 2010; Ravikumar et al., 2011; Calauzenes et al., 2012; Yang & Koyejo, 2020), multi-class (Zhang, 2004; Tewari & Bartlett, 2007) and multi-label classification (Koyejo et al., 2015; Kotłowski & Dembczyński, 2017), classification with abstention (Yuan & Wegkamp, 2010; Ramaswamy et al., 2018), or constrained classification problems (Agarwal et al., 2018; Kearns et al., 2018; Narasimhan et al., 2022). Nevertheless, the problem of designing consistent algorithms for budgeted-at-$k$ macro averages is relatively new.

Optimizing non-decomposable metrics can be considered in two distinct frameworks (Dembczynski et al., 2017): population utility (PU) and expected test utility (ETU). The PU framework focuses on estimation, in the sense that a consistent PU classifier is one which correctly estimates the population optimal utility as the size of the training set increases. A consistent ETU classifier is one which optimizes the expected prediction error over a *given* test set. The latter might get better results, as the optimization is performed on the test set directly. Optimization of budgeted-at-$k$ metrics in this framework has been recently considered in Schultheis et al. (2023). The former framework, which we focus on in this paper, has the advantage that prediction can be made for each test example separately, without knowing the entire test set in advance.

## 3 PROBLEM STATEMENT

Let $\boldsymbol{x} \in \mathcal{X}$ denote an input instance, and $\boldsymbol{y} \in \{0,1\}^m$ the vector indicating the relevant labels, jointly distributed according to $(\boldsymbol{x}, \boldsymbol{y}) \sim \mathbb{P}$. Let $\boldsymbol{h} \colon \mathcal{X} \to [0,1]^m$ be a *randomized multi-label classifier* which, given instance $\boldsymbol{x}$, predicts a possibly randomized class label vector $\widehat{\boldsymbol{y}} \in \{0,1\}^m$, such that $\mathbb{E}_{\widehat{\boldsymbol{y}}|\boldsymbol{x}}[\widehat{\boldsymbol{y}}] = \boldsymbol{h}(\boldsymbol{x})$. We assume that the predictions are *budgeted at* $k$, that is exactly $k$ labels are always predicted as relevant, which means that $\|\widehat{\boldsymbol{y}}\|_1 = \sum_{j=1}^m \widehat{y}_j = k$ with probability 1. It turns out that this is *equivalent* to assuming $\|\boldsymbol{h}(\boldsymbol{x})\|_1 = \sum_{j=1}^m h_j(\boldsymbol{x}) = k$ for all $\boldsymbol{x} \in \mathcal{X}$. Indeed, $\|\boldsymbol{h}(\boldsymbol{x})\|_1 = k$ is *necessary*, because $k = \mathbb{E}_{\widehat{\boldsymbol{y}}|\boldsymbol{x}}[\|\widehat{\boldsymbol{y}}\|_1] = \|\boldsymbol{h}(\boldsymbol{x})\|_1$; but it also *suffices* as for any real-valued vector $\boldsymbol{\pi} \in [0,1]^m$ with $\|\boldsymbol{\pi}\|_1 = k$, one can construct a distribution over binary vectors $\widehat{\boldsymbol{y}} \in \{0,1\}^m$ with $\|\widehat{\boldsymbol{y}}\|_1 = k$ and marginals $\mathbb{E}_{\widehat{\boldsymbol{y}}}[\widehat{\boldsymbol{y}}] = \boldsymbol{\pi}$; this can be accomplished using, e.g., *Madow's sampling scheme* (see Appendix A for the actual efficient algorithm). Thus, using notation $\Delta_m^k := \{\boldsymbol{v} \in [0,1]^m : \|\boldsymbol{v}\|_1 = k\}$, the randomized classifiers budgeted at $k$ are then all (measurable) functions of the form $\boldsymbol{h} \colon \mathcal{X} \to \Delta_m^k$. We denote the set of such functions as $\mathcal{H}$.

For any $\boldsymbol{x} \in \mathcal{X}$, let $\boldsymbol{\eta}(\boldsymbol{x}) := \mathbb{E}_{\boldsymbol{y}|\boldsymbol{x}}[\boldsymbol{y}]$ denote the vector of conditional label marginals. Given a randomized classifier $\boldsymbol{h} \in \mathcal{H}$, we define its *multi-label confusion tensor* $\mathbf{C}(\boldsymbol{h}) = (\boldsymbol{C}^1(h_1), \ldots, \boldsymbol{C}^m(h_m))$ as a sequence of $m$ binary classification confusion matrices associated with each label $j \in [m]$, that is $C_{uv}^j(h_j) = \mathbb{P}[y_j = u, \widehat{y}_j = v]$ for $u, v \in \{0,1\}$. Note that using the marginals and the definition of the randomized classifier,

$$\boldsymbol{C}^j(h_j) = \begin{pmatrix} \mathbb{E}_{\boldsymbol{x}}[(1 - \eta_j(\boldsymbol{x}))(1 - h_j(\boldsymbol{x}))] & \mathbb{E}_{\boldsymbol{x}}[(1 - \eta_j(\boldsymbol{x}))h_j(\boldsymbol{x})] \\ \mathbb{E}_{\boldsymbol{x}}[\eta_j(\boldsymbol{x})(1 - h_j(\boldsymbol{x}))] & \mathbb{E}_{\boldsymbol{x}}[\eta_j(\boldsymbol{x})h_j(\boldsymbol{x})] \end{pmatrix}. \tag{1}$$

The set of all possible binary confusion matrices is written as $\mathcal{C} := \{\boldsymbol{C} \in [0,1]^{2 \times 2} \mid \|\boldsymbol{C}\|_{1,1} = 1\}$, and is used to define the set of possible confusion tensors for predictions at $k$ through $\mathcal{C}^k := \{\mathbf{C} \in [0,1]^{m \times 2 \times 2} \mid \forall j \in [m] : \boldsymbol{C}^j \in \mathcal{C}, \sum_{j=1}^m C_{01}^j + C_{11}^j = k\}$

In this work, we are interested in optimizing performance metrics that do not decompose over individual instances, but are general functions of the confusion tensor of the classifier $\boldsymbol{h}$. While in general, given two confusion matrices, we cannot say which one is better than the other without knowing the specific application, it is possible to impose a *partial* order that any reasonable performance metric should respect. To that end, define:

Table 1: Examples of binary confusion matrix measures, which can be used as building blocks for confusion tensor measures. For clarity, we denote $\text{tn} = C_{00}, \text{fp} = C_{01}, \text{fn} = C_{10}, \text{tp} = C_{11}$.

| Metric | $\psi(\boldsymbol{C})$ | Metric | $\psi(\boldsymbol{C})$ |
|---|---|---|---|
| Accuracy | $\text{tp} + \text{tn}$ | Recall | $\frac{\text{tp}}{\text{tp}+\text{fn}}$ |
| Precision | $\frac{\text{tp}}{\text{tp}+\text{fp}}$ | Balanced accuracy | $\frac{\text{tp}}{2(\text{tp}+\text{fn})} + \frac{\text{tn}}{2(\text{tn}+\text{fp})}$ |
| $F_\beta$ | $\frac{(1+\beta^2)\text{tp}}{(1+\beta^2)\text{tp}+\beta^2\text{fn}+\text{fp}}$ | G-Mean | $\sqrt{\frac{\text{tp}\cdot\text{tn}}{(\text{tp}+\text{fn})(\text{tn}+\text{fp})}}$ |
| Jaccard | $\frac{\text{tp}}{\text{tp}+\text{fp}+\text{fn}}$ | AUC | $\frac{2\cdot\text{tp}\cdot\text{tn}+\text{tp}\cdot\text{fp}+\text{fn}\cdot\text{tn}}{2(\text{tp}+\text{fn})(\text{fp}+\text{tn})}$ |

**Definition 3.1** (Binary Confusion Matrix Measure). *Let $\mathcal{C} = \left\{ \boldsymbol{C} \in [0,1]^{2\times2} \mid \|\boldsymbol{C}\|_{1,1} = 1 \right\}$ be the set of all possible binary confusion matrices, and $\boldsymbol{C}, \boldsymbol{C}' \in \mathcal{C}$. Then we say that $\boldsymbol{C}'$ is* at least as good *as $\boldsymbol{C}$, $\boldsymbol{C}' \succeq \boldsymbol{C}$, if there exists constants $\epsilon_1, \epsilon_2$ such that*

$$\boldsymbol{C}' = \begin{pmatrix} C_{00} + \epsilon_1 & C_{01} - \epsilon_1 \\ C_{10} - \epsilon_2 & C_{11} + \epsilon_2 \end{pmatrix}, \tag{2}$$

*i.e., if $\boldsymbol{C}'$ can be generated from $\boldsymbol{C}$ by turning some false positives to true negatives and false negatives to true positives. A function $\psi\colon \mathcal{C} \longrightarrow [0,1]$ is called a* binary confusion matrix measure *(Singh & Khim, 2022) if it respects that ordering, i.e., if for $\boldsymbol{C}' \succeq \boldsymbol{C}$ we have $\psi(\boldsymbol{C}') \geq \psi(\boldsymbol{C})$.*

Similarly, in the multi-label case we cannot compare arbitrary confusion tensors, where one is better on some labels than on others,[2] but we can recognize if one is better on *all* labels:

**Definition 3.2** (Confusion Tensor Measure). *For a given number of labels $m \in \mathbb{N}$, and two confusion tensors $\mathbf{C}, \mathbf{C}' \in \mathcal{C}^m$, we say that $\mathbf{C}'$ is* at least good as $\mathbf{C}$, $\mathbf{C}' \succeq \mathbf{C}$, *if for all labels $j \in [m]$ it holds that $\boldsymbol{C}^{j\prime} \succeq \boldsymbol{C}^j$. A function $\Psi\colon \mathcal{C}^m \longrightarrow [0,1]$ is called a* confusion tensor measure *if it respects this ordering, i.e., if for $\mathbf{C}' \succeq \mathbf{C}$ we have $\Psi(\mathbf{C}') \geq \Psi(\mathbf{C})$.*

Of particular interest in this paper are functions which linearly decompose over the labels, that is *macro-averaged multi-label metrics* (Manning et al., 2008; Parambath et al., 2014; Koyejo et al., 2015; Kotłowski & Dembczyński, 2017) of the form:

$$\Psi(\boldsymbol{h}) = \Psi(\mathbf{C}(\boldsymbol{h})) = m^{-1} \sum_{j=1}^{m} \psi\big(\boldsymbol{C}^j(h_j)\big), \tag{3}$$

where $\psi$ is some binary confusion matrix measure. If one takes a binary confusion matrix measure (e.g., any of those define in Table 1), then the resulting macro-average will be a valid confusion tensor measure. A more thorough discussion of these conditions can be found in Appendix H.

Macro-averaged metrics find numerous applications in multi-label classification, mainly due to their balanced emphasis across labels independent of their frequencies, and thus can potentially alleviate the "long-tail" issues in problems with many rare labels (Schultheis et al., 2022).

Denote the optimal value of the metric among all classifiers budgeted at $k$ as:

$$\Psi^\star := \sup_{\boldsymbol{h}\in\mathcal{H}} \Psi(\boldsymbol{h}), \tag{4}$$

and let $\boldsymbol{h}^\star \in \arg\max_{\boldsymbol{h}} \Psi(\boldsymbol{h})$ be an optimal (Bayes) classifier for which $\Psi(\boldsymbol{h}^\star) = \Psi^\star$, if it exists. For any classifier $\boldsymbol{h}$, define its $\Psi$-*regret* as $\Delta\Psi(\boldsymbol{h}) = \Psi^\star - \Psi(\boldsymbol{h})$ to measure the suboptimality of $\boldsymbol{h}$ with respect to $\Psi$: from the definition, $\Delta\Psi(\boldsymbol{h}) \geq 0$ for every classifier $\boldsymbol{h}$, and $\Delta\Psi(\boldsymbol{h}) = 0$ if and only if $\boldsymbol{h}$ is optimal. If the $\Psi$-regret of a learning algorithm converges to zero with the sample size tending to infinity, it is called *(statistically) consistent*. We consider such algorithms in Section 5. Even though the objective (3) decomposes onto $m$ binary problems, these are still coupled by the budget constraint, $\|\boldsymbol{h}(\boldsymbol{x})\|_1 = k$ for all $\boldsymbol{x} \in \mathcal{X}$, and cannot be optimized independently as we show later in the paper.

---

[2]This is specifically the trade-off we want to achieve for tail labels!

## 4 THE OPTIMAL CLASSIFIER

Finding the form of the optimal classifier for general macro-averaged performance metrics is difficult. For instance, when $\psi(\boldsymbol{C})$ is the $F_\beta$ measure, the objective to be optimized is a sum of linear fractional functions, which is known to be NP-hard in general (Schaible & Shi, 2003). We are, however, able to fully determine the optimal classifier for the specific class of *linear utilities*, which are metrics depending linearly on the confusion tensor of the classifier. Furthermore, we also show that for a general class of macro-averaged metrics, under mild assumptions on the data distribution, the optimal classifier exists and turns out to also be the maximizer of some linear utility, whose coefficients, however, depend on its (unknown a priori) confusion tensor.

We start with a metric of the form[3] $\Psi(\mathbf{C}) = \mathbf{G} \cdot \mathbf{C} = \sum_{j=1}^m \boldsymbol{G}^j \cdot \boldsymbol{C}^j$ for some vector of *gain matrices* (*gain tensor*) $\mathbf{G} = (\boldsymbol{G}^1, \ldots, \boldsymbol{G}^m)$, possibly depending on the data distribution $\mathbb{P}$. We call such a utility *linear* as it linearly depends on the confusion matrices of the classifier. Note that we allow the gain matrix $\boldsymbol{G}$ to be different for each label, making this more general than linear macro-averages. We need to consider this more general case, because it will appear as a subproblem when finding optimal predictions for non-linear macro-averages as presented below.

Linear metrics are decomposable over instances, and thus the optimal classifier has an appealingly simple form: It boils down to simply sorting the labels by an affine function of the marginals, and returning the top $k$ elements.

**Theorem 4.1.** *The optimal classifier* $\boldsymbol{h}^\star := \operatorname{argmax}_{\boldsymbol{h} \in \mathcal{H}} \Psi(\boldsymbol{h})$ *for* $\Psi(\boldsymbol{h}) = \mathbf{G} \cdot \mathbf{C}(\boldsymbol{h})$ *is given by*

$$\boldsymbol{h}^\star(\boldsymbol{x}) = \operatorname{top}_k(\boldsymbol{a} \odot \boldsymbol{\eta}(\boldsymbol{x}) + \boldsymbol{b}), \tag{5}$$

*where* $\odot$ *denotes the coordinate-wise product of vectors, while the vectors* $\boldsymbol{a}$ *and* $\boldsymbol{b}$ *are given by:*

$$a_j = G_{00}^j + G_{11}^j - G_{01}^j - G_{10}^j, \qquad b_j = G_{01}^j - G_{00}^j, \tag{6}$$

*and* $\operatorname{top}_k(\boldsymbol{v})$ *returns a k-hot vector extracting the top k largest entries of* $\boldsymbol{v}$ *(ties broken arbitrarily).*

*Proof (sketch, full proof in Appendix B).* After simple algebraic manipulations, the objective can be written as $\Psi(\boldsymbol{h}) = \mathbb{E}\left[\sum_{j=1}^m (a_j \eta_j(\boldsymbol{x}) + b_j) h_j(\boldsymbol{x})\right] + R$, where $a_j$ and $b_j$ are as stated in the theorem, while $R$ does not depend on the classifier. For each $\boldsymbol{x} \in \mathcal{X}$, the objective can thus be independently maximized by the choice of $\boldsymbol{h}(\boldsymbol{x}) \in \Delta_m^k$ which maximizes $\sum_{j=1}^m (a_j \eta_j(\boldsymbol{x}) + b_j) h_j(\boldsymbol{x})$. But this is achieved by sorting $a_j \eta_j(\boldsymbol{x}) + b_j$ in descending order, and setting $h_j(\boldsymbol{x}) = 1$ for the top $k$ coordinates, and $h_j(\boldsymbol{x}) = 0$ for the remaining coordinates (with ties broken arbitrarily). $\square$

Examples of binary metrics for which their macro averages can be written in the linear form include:

- the accuracy $\psi(\boldsymbol{C}) = C_{00} + C_{11}$ (which leads to the *Hamming utility* after macro-averaging) with $a_j = 2, b_j = -1$, and thus for any $\boldsymbol{x} \in \mathcal{X}$, the optimal prediction $\boldsymbol{h}^\star(\boldsymbol{x})$ returns $k$ labels with the largest marginals $\eta_j(\boldsymbol{x})$;

- the same prediction rule is obtained for the *TP* metric $\psi(\boldsymbol{C}) = C_{00}$ (that leads to *precision@k*) with $a_j = 1, b_j = 0$;

- the recall $\psi(\boldsymbol{C}) = \mathbb{P}(y=1)^{-1} C_{11}$ (macro-averaged to *recall@k*) has $a_j = \mathbb{P}(y_j = 1)^{-1}, b_j = 0$, so that the optimal classifiers returns top $k$ labels sorted according to $\frac{\eta_j(\boldsymbol{x})}{\mathbb{P}(y_j=1)}$;

- the balanced accuracy $\psi(\boldsymbol{C}) = \frac{C_{11}}{2\mathbb{P}(y=1)} + \frac{C_{00}}{2\mathbb{P}(y=0)}$, gives $a_j = \frac{1}{2\mathbb{P}(y_j=1)} + \frac{1}{2\mathbb{P}(y_j=0)}, b_j = -\frac{1}{2\mathbb{P}(y_j=0)}$, with the optimal prediction sorting labels according to $\frac{\eta_j(\boldsymbol{x})}{\mathbb{P}(y_j=1)} - \frac{1-\eta_j(\boldsymbol{x})}{1-\mathbb{P}(y_j=1)}$.

We now switch to general case, in which the base binary metrics are not necessarily decomposable over instances, and optimizing their macro averages with budgeted predictors is a challenging task. We make the following mild assumptions on the data distribution and performance metric:

---

[3]We use $\boldsymbol{A} \cdot \boldsymbol{B} = \sum_{uv} A_{uv} B_{uv}$ to denote a dot product over matrices, and a concise notation $\mathbf{A} \cdot \mathbf{B} = \sum_j \boldsymbol{A}^j \cdot \boldsymbol{B}^j$ for 'dot product' over matrix sequences $\mathbf{A} = (\boldsymbol{A}^1, \ldots, \boldsymbol{A}^m)$ and $\mathbf{B} = (\boldsymbol{B}^1, \ldots, \boldsymbol{B}^m)$.

**Assumption 4.2.** *The label conditional marginal vector $\boldsymbol{\eta}(\boldsymbol{x}) = \mathbb{E}_{\boldsymbol{y}|\boldsymbol{x}}[\boldsymbol{y}]$ is absolutely continuous with respect to the Lebesgue measure on $[0, 1]^m$ (i.e., has a density over $[0, 1]^m$).*

A similar assumption was commonly used in the past works (Koyejo et al., 2014; Narasimhan et al., 2015; Dembczynski et al., 2017).

**Assumption 4.3.** *The performance metric $\Psi$ is differentiable and fulfills for all labels $j \in [m]$*

$$\frac{\partial}{\partial \epsilon} \Psi \left( \boldsymbol{C}^1, \ldots, \boldsymbol{C}^j + \epsilon \begin{pmatrix} 1 & -1 \\ -1 & 1 \end{pmatrix}, \ldots, \boldsymbol{C}^m \right) \Bigg|_{\epsilon=0} > 0 \,. \tag{7}$$

Assumption 4.3 is essentially a 'strictly monotonic and differentiable' version of Definition 3.2, and is satisfied by all macro-averaged metrics given in Table 1.

Our first main result concerns the form of the optimal classifier for general confusion tensor measures, of which macro-averaged binary confusion matrix measures are special cases. To state the result, we define $\mathcal{C}_{\mathbb{P}} := \{\mathbf{C}(\boldsymbol{h}) : \boldsymbol{h} \in \mathcal{H}\}$, the set of confusion tensors achievable by randomized $k$-budgeted classifiers on distribution $\mathbb{P}$. Clearly, maximizing $\Psi(\boldsymbol{h})$ over $\boldsymbol{h} \in \mathcal{H}$ is equivalent to maximizing $\Psi(\mathbf{C})$ over $\mathbf{C} \in \mathcal{C}_{\mathbb{P}}$.

**Theorem 4.4.** *Let the data distribution $\mathbb{P}$ and metric $\Psi$ satisfy Assumption 4.2 and Assumption 4.3 respectively. Then, there exists an optimal $\mathbf{C}^\star \in \mathcal{C}_{\mathbb{P}}$, that is $\Psi(\mathbf{C}^\star) = \Psi^\star$. Moreover, any classifier $\boldsymbol{h}^\star$ maximizing the linear utility $\mathbf{G} \cdot \mathbf{C}(\boldsymbol{h})$ over $\boldsymbol{h} \in \mathcal{H}$ with $\mathbf{G} = (\boldsymbol{G}^1, \ldots, \boldsymbol{G}^m)$ given by $\boldsymbol{G}^j = \nabla_{\boldsymbol{C}^j} \Psi(\mathbf{C}^\star)$, also maximizes $\Psi(\boldsymbol{h})$ over $\boldsymbol{h} \in \mathcal{H}$.*

*Proof (sketch, full proof in Appendix C.* We first prove that $\mathcal{C}_{\mathbb{P}}$ is a compact set by using certain properties of continuous linear operators in Hilbert space. Due to continuity of $\Psi$ and the compactness of $\mathcal{C}_{\mathbb{P}}$, there exists a maximizer $\mathbf{C}^\star = \operatorname{argmax}_{\mathbf{C} \in \mathcal{C}_{\mathbb{P}}} \Psi(\mathbf{C})$. By the first order optimality and convexity of $\mathcal{C}_{\mathbb{P}}$, $\nabla \Psi(\mathbf{C}^\star) \cdot \mathbf{C}^\star \geq \nabla \Psi(\mathbf{C}^\star) \cdot \mathbf{C}$ for all $\mathbf{C} \in \mathcal{C}_{\mathbb{P}}$, so $\mathbf{C}^\star$ maximizes a linear utility $\mathbf{G} \cdot \mathbf{C}^\star$ with gain matrices given by $\mathbf{G} = \nabla \Psi(\mathbf{C}^\star)$. A careful analysis under Assumption 4.2 shows that $\mathbf{C}^\star$ uniquely maximizes $\mathbf{G} \cdot \mathbf{C}$ over $\mathbf{C} \in \mathcal{C}_{\mathbb{P}}$. $\square$

Theorem 4.4 reveals that $\Psi$-optimal classifier exists and can be found by maximizing a linear utility, that is, by predicting the top $k$ labels sorted according to an affine function of the label marginals: $\boldsymbol{h}^\star(\boldsymbol{x}) = \operatorname{top}_k(\boldsymbol{a}^\star \odot \boldsymbol{\eta}(\boldsymbol{x}) + \boldsymbol{b}^\star)$ for vectors $\boldsymbol{a}^\star$ and $\boldsymbol{b}^\star$ defined for gain matrices $\mathbf{G} = \nabla \Psi(\mathbf{C}^\star)$ as in Theorem 4.1. Unfortunately, since $\mathbf{C}^\star$ is unknown in advance, the coefficients $\boldsymbol{a}^\star, \boldsymbol{b}^\star$ are also unknown, and the optimal classifier is not directly available. However, knowing that $\boldsymbol{h}^\star$ optimizes a linear utility induced by the gradient of $\Psi$ leads to a consistent algorithm described in the next section.

Although the optimal solution is expressed by affine functions of label marginals, in general, it cannot be obtained by solving the problem independently for each label, i.e., the values of $a_j$ and $b_j$ may depend on labels other than $j$. Let $\boldsymbol{h}^\star(\boldsymbol{x})$ and $\boldsymbol{h}'^\star(\boldsymbol{x})$ be optimal for distributions $\mathbb{P}$ and $\mathbb{P}'$, respectively. Let $\mathbb{P}'$ differ from $\mathbb{P}$ only on a single label $j$. If we could solve the problem independently for each label, then $\boldsymbol{h}^\star(\boldsymbol{x})$ and $\boldsymbol{h}'^\star(\boldsymbol{x})$ would be the same up to label $j$, in the sense that the ordering relation between all other labels would not change. In Appendix E we show that this is not the case, presenting a simple counterexample showing that a different distribution on a single label changes the solution with respect to the other labels.

## 5 CONSISTENT ALGORITHMS

As any algorithm we propose has to operate on a finite sample, we need to introduce empirical counterparts for our quantities of interest. For example, we use $\widehat{\boldsymbol{\eta}}(\boldsymbol{x})$ to denote the estimate of $\boldsymbol{\eta}(\boldsymbol{x})$ given by a label probability estimator trained on some set of training data $\mathcal{S} = \{(\boldsymbol{x}_1, \boldsymbol{y}_1), \ldots, (\boldsymbol{x}_n, \boldsymbol{y}_n)\}$. We also define the empirical multi-label confusion tensor $\widehat{\mathbf{C}}(\boldsymbol{h}, \mathcal{S})$ of a classifier $\boldsymbol{h}$ with respect to some set $\mathcal{S}$ of $n$ instances. In this case, we have:

$$\widehat{C}_{uv}^j(h_j, \mathcal{S}) = \frac{1}{n} \sum_{i=1}^n \mathbb{1}[y_{ij} = u, h_j(\boldsymbol{x}_i) = v] \,. \tag{8}$$

Following Narasimhan et al. (2015), we use the Frank-Wolfe algorithm (Frank & Wolfe, 1956) to perform an implicit optimization over feasible confusion tensors $\mathcal{C}_{\mathbb{P}}$, without having to explicitly

construct $\mathcal{C}_{\mathbb{P}}$. This is possible, because Frank-Wolfe only requires us to be able to solve two sub-problems: First, given a classifier $\boldsymbol{h}$, we need to calculate its empirical confusion tensor, which is straight-forward. Second, given a classifier and its corresponding confusion tensor, we need to solve a *linearized* version of the optimization problem, which is possible due to Theorem 4.1.

Consequently, our algorithm, presented in Algorithm 1, proceeds as follows: In the beginning, we split the available training data into two subsets. One for estimating label probabilities $\widehat{\boldsymbol{\eta}}$, and one for tuning the actual classifier. After determining $\widehat{\boldsymbol{\eta}}$, we initialize $\boldsymbol{h}$ to be the standard top-k classifier, which will get iteratively refined as follows. For the confusion tensor of the current classifier, we can determine a linear objective based on its gradient. Because we can linearly interpolate stochastic classifiers, which will lead to linearly interpolated confusion tensors, this gives us a descent direction over which we can optimize a step-size,[4] and the confusion tensor at this classifier. Based on this confusion tensor, we can do the next linearized optimization step, until we reach a fixed limit for the iteration count. We represent the randomized classifier as a set of deterministic classifiers $\boldsymbol{h}^i$, and corresponding sampling weights $\alpha^i$ obtained across all iterations of the algorithm. The Frank-Wolfe algorithm scales to the larger problems as it only requires $\mathcal{O}(nm)$ time per iteration.

---

**Algorithm 1** Multi-label Frank-Wolfe algorithm for complex performance measures

---

**Require:** Dataset $\mathcal{S} := \{(\boldsymbol{x}_1, \boldsymbol{y}_1), \ldots, (\boldsymbol{x}_n, \boldsymbol{y}_n)\}$, number of iterations $t \in \mathbb{N}$, stopping condition $\epsilon \in \mathbb{R}$
1: Split dataset $\mathcal{S}$ into $\mathcal{S}_1$ and $\mathcal{S}_2$
2: Learn label marginals model $\widehat{\boldsymbol{\eta}} : \mathcal{X} \to \mathbb{R}^m$ on $\mathcal{S}_1$
3: Initialize $\boldsymbol{h}^0 : \mathcal{X} \to \widehat{\mathcal{Y}}_k$         ▷ Initial deterministic classifier
4: Initialize $\alpha^0 \leftarrow 1$         ▷ Initial probability of selecting the initial classifier $\boldsymbol{h}^0$
5: $\mathbf{C}^0 \leftarrow \widehat{\mathbf{C}}(h^0, \mathcal{S}_2)$         ▷ Calculate the initial confusion tensor
6: **for** $i \in \{1, \ldots, t\}$ **do**         ▷ Perform $t$ iterations
7:     $\mathbf{G}^i \leftarrow \nabla_{\mathbf{C}} \Psi(\mathbf{C}^{i-1})$         ▷ Calculate tensor of gradients of $\mathbf{C}^{i-1}$ in respect to $\Psi$ (gain tensor)
8:     $\boldsymbol{a}^i \leftarrow \mathbf{G}^i_{11} + \mathbf{G}^i_{00} - \mathbf{G}^i_{01} - \mathbf{G}^i_{10}$, $\boldsymbol{b}^i \leftarrow \mathbf{G}^i_{01} - \mathbf{G}^i_{00}$
9:     $\boldsymbol{h}^i(\boldsymbol{x}) \leftarrow \text{top}_k(\boldsymbol{a}^i \odot \widehat{\boldsymbol{\eta}}(\boldsymbol{x}) + \boldsymbol{b}^i)$         ▷ Construct the next classifier $\boldsymbol{h}^i$
10:     $\mathbf{C}' \leftarrow \widehat{\mathbf{C}}(h^i, \mathcal{S}_2)$         ▷ Calculate the confusion tensor of the next classifier $\boldsymbol{h}^i$
11:     $\alpha^i \leftarrow \arg\max_{\alpha \in [0,1]} \Psi((1-\alpha)\mathbf{C}^{i-1} + \alpha\mathbf{C}')$ ▷ Find the best combination of $\mathbf{C}^{i-1}$ and $\mathbf{C}'$ (step-size)
12:     **if** $\alpha^i < \epsilon$ **then break**         ▷ Stop if the step-size $\alpha^i$ is smaller then $\epsilon$
13:     $\mathbf{C}^i \leftarrow (1 - \alpha^i)\mathbf{C}^{i-1} + \alpha^i\mathbf{C}'$         ▷ Calculate a new confusion tensor based on the best $\alpha^i$
14:     **for** $j \in \{0, \ldots, i-1\}$ **do**         ▷ Update all the previous
15:         $\alpha^j \leftarrow \alpha^j(1 - \alpha^i)$         ▷ probabilities of selecting corresponding $\boldsymbol{h}$
16: **return** $(\{\boldsymbol{h}^0, \ldots, \boldsymbol{h}^i\}, \{\alpha^0, \ldots, \alpha^i\})$         ▷ Return randomized classifier

---

This algorithm can consistently optimize a confusion tensor measure if it fulfills certain conditions:

**Theorem 5.1** (Consistency of Frank-Wolfe). *Assume the utility function* $\Psi : [0,1]^{m \times 2 \times 2} \longrightarrow \mathbb{R}_{\geq 0}$ *is concave over* $\mathcal{C}_{\mathbb{P}}$, *L-Lipschitz, and* $\beta$-*smooth w.r.t. the 1-norm. Let* $\mathcal{S} = (\mathcal{S}_1, \mathcal{S}_2)$ *be a sample drawn i.i.d. from* $\mathbb{P}$. *Further, let* $\widehat{\boldsymbol{\eta}}$ *be a label probability estimator learned from* $\mathcal{S}_1$, *and* $\boldsymbol{h}^{\text{FW}}_{\mathcal{S}}$ *be the classifier obtained after* $\kappa n$ *iterations. Then, for any* $\delta \in (0,1]$, *with probability of at least* $1 - \delta$ *over draws of* $\mathcal{S}$,

$$\Delta\Psi(\boldsymbol{h}^{\text{FW}}_{\mathcal{S}}) \leq \mathcal{O}(\mathbb{E}_{\boldsymbol{x}}[\|\boldsymbol{\eta}(\boldsymbol{x}) - \widehat{\boldsymbol{\eta}}(\boldsymbol{x})\|_1]) + \tilde{\mathcal{O}}\left(m^2 \sqrt{\frac{m \cdot \log m \cdot \log n - \log \delta}{n}}\right) + \frac{8\beta m}{\kappa n + 2}. \quad (9)$$

The proof of this theorem, given in Appendix D, broadly follows (Narasimhan et al., 2015): First, we show that *linear* metrics can be estimated consistently with a regret growing with the $L_1$ error of the LPE. Then, we prove a uniform convergence result for estimating the multi-label confusion tensor. As a prerequisite, we derive the VC-dimension of the class of classifiers based on top-k scoring, i.e., those classifiers that minimize some linear confusion tensor metric as shown in Theorem 4.1.

**Lemma 5.2** (VC dimension for linear top-k classifiers). *For* $\boldsymbol{\eta} : \mathcal{X} \longrightarrow [0,1]^m$, *define*

$$\mathcal{H}^j_{\boldsymbol{\eta}} := \bigcup_{\boldsymbol{a}, \boldsymbol{b} \in \mathbb{R}^m} \{h : \mathcal{X} \longrightarrow \{0,1\} : h(\boldsymbol{x}) = \mathbb{1}[j \in \text{top}_k(\boldsymbol{a} \odot \boldsymbol{\eta} + \boldsymbol{b})]\}. \quad (10)$$

*The VC-complexity of this class is* $\text{VC}(\mathcal{H}^j_{\boldsymbol{\eta}}) \leq 6m \log(em)$.

---

[4]The classical version of FW uses a fixed step-size schedule of $\frac{2}{t+1}$ instead of an inner optimization, but we find the latter to give better results empirically. However, for the convergence result, fixed steps are assumed.

Table 2: Results of different inference strategies on measure calculated at $\{3, 5, 10\}$. Notation: P—precision, R—recall, F1—F1-measure. The green color indicates cells in which the strategy matches the metric. The best results are in **bold** and the second best are in *italic*. We additionally report basic statistics of the benchmarks: number of labels and instances in train and test sets, and average number of positive labels per instance, average number of positive instances per label.

| Inference strategy | Instance @3 P | R | Macro @3 P | R | F1 | Instance @5 P | R | Macro @5 P | R | F1 | Instance @10 P | R | Macro @10 P | R | F1 |
|---|---|---|---|---|---|---|---|---|---|---|---|---|---|---|---|
| **MEDIAMILL** ($m = 101$, $n_{\text{train}} = 30993$, $n_{\text{test}} = 12914$, $\mathbb{E}[\|\boldsymbol{y}\|_1] = 4.36$, $\mathbb{E}[y \times n_{\text{train}}] = 1338.8$) | | | | | | | | | | | | | | | |
| Top-K | **66.25** | *49.55* | 8.96 | 4.81 | 4.95 | *51.96* | *62.04* | 12.85 | 8.75 | 7.71 | **33.63** | 76.60 | 11.46 | 19.68 | 11.28 |
| Top-K+$\boldsymbol{w}^{\text{POW}}$ | 57.36 | 42.51 | 15.31 | 11.84 | *10.54* | 47.68 | 56.62 | 13.00 | 17.37 | *12.64* | 32.18 | 72.98 | 9.64 | 29.43 | *13.07* |
| Top-K+$\boldsymbol{w}^{\text{LOG}}$ | 39.72 | 27.32 | 14.43 | 10.10 | 9.41 | 35.40 | 39.96 | 11.38 | 15.33 | 10.95 | 28.45 | 63.36 | 9.86 | 26.25 | 12.26 |
| Top-K+$\ell_{\text{FOCAL}}$ | 65.87 | **49.60** | 10.08 | 4.87 | 4.94 | **52.08** | **62.16** | 11.99 | 8.93 | 7.90 | *33.61* | 76.65 | 10.76 | 20.08 | 11.37 |
| Top-K+$\ell_{\text{ASYM}}$ | *65.88* | 49.48 | 10.31 | 4.58 | 4.80 | 51.55 | 61.87 | 11.10 | 8.50 | 7.48 | 33.54 | **76.75** | 10.73 | 19.55 | 11.16 |
| Macro-P$_{\text{FW}}$ | 7.94 | 6.13 | **19.33** | 6.06 | 2.87 | 6.99 | 8.96 | **17.29** | 8.79 | 3.17 | 6.02 | 14.14 | **17.38** | 17.24 | 5.23 |
| Macro-R$_{\text{PRIOR}}$ | 6.37 | 3.67 | 8.81 | **19.82** | 5.31 | 7.38 | 7.25 | 8.91 | **26.50** | 6.71 | 8.31 | 17.42 | 10.53 | **39.24** | 8.85 |
| Macro-R$_{\text{FW}}$ | 6.37 | 3.67 | 8.81 | **19.82** | 5.31 | 7.38 | 7.25 | 8.91 | **26.50** | 6.71 | 8.31 | 17.42 | 10.53 | **39.24** | 8.85 |
| Macro-F1$_{\text{FW}}$ | 45.20 | 33.05 | *15.42* | 11.17 | **12.21** | 43.57 | 51.60 | *15.20* | 15.05 | **13.82** | 28.12 | 64.23 | *13.93* | 23.32 | **14.81** |
| **FLICKR** ($m = 195$, $n_{\text{train}} = 56359$, $n_{\text{test}} = 24154$, $\mathbb{E}[\|\boldsymbol{y}\|_1] = 1.34$, $\mathbb{E}[y \times n_{\text{train}}] = 412.6$) | | | | | | | | | | | | | | | |
| Top-K | **23.94** | **56.96** | 23.04 | 38.41 | *26.56* | **16.99** | **66.01** | 17.12 | 47.03 | 23.49 | **10.16** | **77.35** | 10.72 | 59.37 | 17.24 |
| Top-K+$\boldsymbol{w}^{\text{POW}}$ | 22.35 | 53.44 | 17.96 | 44.26 | 24.21 | 16.10 | 62.80 | 13.76 | 52.39 | 20.68 | 9.77 | 74.54 | 9.08 | 63.98 | 15.08 |
| Top-K+$\boldsymbol{w}^{\text{LOG}}$ | 23.57 | 56.17 | 19.86 | 41.36 | 25.49 | 16.76 | 65.21 | 15.05 | 49.75 | 22.00 | *10.06* | 76.63 | 9.79 | 61.80 | 16.10 |
| Top-K+$\ell_{\text{FOCAL}}$ | *23.64* | 56.27 | 24.90 | 36.67 | 26.42 | *16.89* | 65.62 | 18.53 | 45.67 | *24.16* | 10.05 | 76.63 | 11.77 | 57.90 | *18.14* |
| Top-K+$\ell_{\text{ASYM}}$ | 23.37 | 55.65 | 23.09 | 37.00 | 26.12 | 16.74 | 65.04 | 17.39 | 45.61 | 23.60 | 10.06 | *76.63* | 10.91 | 58.36 | 17.48 |
| Macro-P$_{\text{FW}}$ | 4.65 | 11.49 | **39.34** | 6.63 | 8.06 | 5.66 | 22.75 | **41.74** | 9.70 | 10.57 | 2.83 | 22.26 | **37.59** | 10.68 | 8.50 |
| Macro-R$_{\text{PRIOR}}$ | 16.14 | 38.62 | 17.58 | **45.50** | 22.27 | 12.17 | 47.48 | 13.98 | **53.83** | 19.72 | 7.89 | 60.42 | 9.57 | **64.66** | 15.07 |
| Macro-R$_{\text{FW}}$ | 16.14 | 38.62 | 17.58 | **45.50** | 22.27 | 12.17 | 47.48 | 13.98 | **53.83** | 19.72 | 7.89 | 60.42 | 9.57 | **64.66** | 15.07 |
| Macro-F1$_{\text{FW}}$ | 17.59 | 41.60 | *35.28* | 29.28 | **29.43** | 12.22 | 47.31 | *34.13* | 32.70 | **29.43** | 5.92 | 45.77 | *34.55* | 33.08 | **29.02** |
| **RCV1X** ($m = 2456$, $n_{\text{train}} = 623847$, $n_{\text{test}} = 155962$, $\mathbb{E}[\|\boldsymbol{y}\|_1] = 4.80$, $\mathbb{E}[y \times n_{\text{train}}] = 1218.6$) | | | | | | | | | | | | | | | |
| Top-K | **72.99** | **75.32** | 13.06 | 4.67 | 5.43 | **52.30** | **81.96** | 12.77 | 7.61 | 7.64 | **32.98** | **89.70** | 11.35 | 14.75 | 10.28 |
| Top-K+$\boldsymbol{w}^{\text{POW}}$ | 65.99 | 69.11 | 18.58 | 12.78 | **13.09** | 48.48 | 77.18 | 14.69 | 17.66 | *13.64* | 31.43 | 87.14 | 10.63 | 26.05 | *12.82* |
| Top-K+$\boldsymbol{w}^{\text{LOG}}$ | 70.70 | 73.37 | 19.97 | 8.10 | 9.80 | 51.18 | 80.49 | 16.03 | 11.75 | 11.29 | *32.66* | *89.14* | 11.96 | 19.01 | 12.06 |
| Top-K+$\ell_{\text{FOCAL}}$ | *71.99* | *74.38* | 14.06 | 4.83 | 5.76 | *51.46* | *80.94* | 12.49 | 7.65 | 7.75 | 32.38 | 88.75 | 10.59 | 14.42 | 10.06 |
| Top-K+$\ell_{\text{ASYM}}$ | 71.14 | 73.60 | 14.40 | 5.44 | 6.46 | 50.81 | 80.13 | 12.27 | 8.52 | 8.41 | 31.88 | 87.85 | 9.64 | 15.16 | 10.03 |
| Macro-P$_{\text{FW}}$ | 46.36 | 50.11 | *21.11* | 5.61 | 5.84 | 29.40 | 49.81 | *21.69* | 5.72 | 5.31 | 19.45 | 60.40 | *21.66* | 6.03 | 5.78 |
| Macro-R$_{\text{PRIOR}}$ | 44.26 | 46.10 | 14.60 | *18.24* | 12.04 | 34.77 | 56.28 | 13.13 | *24.59* | 12.77 | 24.08 | 70.51 | 10.66 | *34.34* | 12.39 |
| Macro-R$_{\text{FW}}$ | 43.28 | 44.99 | 14.56 | **18.41** | 11.95 | 34.15 | 55.24 | 13.15 | **24.89** | 12.73 | 23.78 | 69.71 | 10.76 | **34.66** | 12.44 |
| Macro-F1$_{\text{FW}}$ | 58.20 | 61.22 | **21.45** | 10.37 | *12.09* | 44.42 | 71.86 | **21.96** | 12.25 | **13.68** | 27.26 | 78.88 | **22.10** | 14.86 | **15.12** |
| **AMAZONCAT** ($m = 13330$, $n_{\text{train}} = 1186239$, $n_{\text{test}} = 306782$, $\mathbb{E}[\|\boldsymbol{y}\|_1] = 5.04$, $\mathbb{E}[y \times n_{\text{train}}] = 448.6$) | | | | | | | | | | | | | | | |
| Top-K | **78.29** | 59.29 | 35.73 | 12.44 | 16.52 | **63.63** | **74.54** | 46.43 | 32.72 | 35.06 | **39.16** | **85.18** | 39.52 | 51.69 | 40.39 |
| Top-K+$\boldsymbol{w}^{\text{POW}}$ | 66.32 | 49.76 | 50.21 | 45.79 | **45.70** | 57.12 | 67.49 | 44.85 | 53.78 | **46.30** | 37.31 | 82.20 | 30.13 | 63.53 | 37.15 |
| Top-K+$\boldsymbol{w}^{\text{LOG}}$ | *72.56* | *54.56* | 50.30 | 32.06 | 36.94 | *61.15* | *71.83* | 48.93 | 42.87 | *43.05* | *38.71* | *84.49* | 36.84 | 56.71 | *40.60* |
| Macro-P$_{\text{FW}}$ | 47.00 | 35.57 | *56.47* | 23.74 | 29.62 | 41.04 | 50.74 | *55.85* | 27.45 | 30.23 | 30.66 | 69.67 | **55.27** | 29.09 | 34.51 |
| Macro-R$_{\text{PRIOR}}$ | 48.58 | 34.93 | 37.16 | **59.97** | 42.02 | 40.67 | 47.35 | 28.17 | **66.98** | 35.75 | 28.06 | 62.91 | 17.62 | **73.98** | 25.04 |
| Macro-R$_{\text{FW}}$ | 48.58 | 34.93 | 37.15 | *59.97* | 42.02 | 40.67 | 47.35 | 28.17 | **66.98** | 35.75 | 28.06 | 62.91 | 17.62 | **73.98** | 25.04 |
| Macro-F1$_{\text{FW}}$ | 68.59 | 51.49 | **56.75** | 34.68 | **40.90** | 55.73 | 65.60 | **56.62** | 36.40 | **41.92** | 35.30 | 78.34 | *54.67* | 39.93 | **43.26** |

## 6 EXPERIMENTS

In this section, we empirically evaluate the proposed Frank-Wolfe algorithm on a variety of multi-label benchmark tasks that differ substantially in the number of labels and imbalance of the label distribution: MEDIAMILL (Snoek et al., 2006), FLICKR (Tang & Liu, 2009), RCV1X (Lewis et al., 2004), and AMAZONCAT (McAuley & Leskovec, 2013; Bhatia et al., 2016). For the first three datasets we use a multi-layer neural network for estimating $\widehat{\boldsymbol{\eta}}(\boldsymbol{x})$. For the last and largest dataset, we use a sparse linear label tree model, which is a common baseline in extreme multi-label classification (Jasinska-Kobus et al., 2020).[5] In Appendix F we include all the details regarding the setup of probability estimators.

We evaluate the following classifiers optimizing the macro-at-$k$ measures:

- MACRO-P$_{\text{FW}}$, MACRO-R$_{\text{FW}}$, MACRO-F1$_{\text{FW}}$: randomized classifiers found by the Frank-Wolfe algorithm (Algorithm 1) for optimizing macro precision, recall, and $F_1$, respectively, based on $\widehat{\boldsymbol{\eta}}(\boldsymbol{x})$ coming from the model trained with binary cross-entropy loss (BCE).

---

[5]Code to reproduce the experiments: https://github.com/mwydmuch/xCOLUMNs

- MACRO-R$_{\text{PRIOR}}$: implements the optimal strategy for macro recall, which selects $k$ labels with the highest $\widehat{p}_j^{-1}\widehat{\eta}_j$; $\widehat{p}_j$s are estimates of label priors obtained on a training set and $\widehat{\eta}(x)$ are given by the model trained with BCE loss.

As baselines, we use the following algorithms:

- TOP-K: selects $k$ labels with the highest $\widehat{\eta}_j$ coming from the model trained with BCE loss; the optimal strategy for instance-wise precision at $k$ (Wydmuch et al., 2018).
- TOP-K+$w^{\text{POW}}$, TOP-K+$w^{\text{LOG}}$: similarly to TOP-K, selects $k$ labels with the highest $w_j\widehat{\eta}_j$, where $w_j$ are calculated as a function of label priors corresponding to the power-law, $w_j^{\text{pow}} = \widehat{p}_j^{-\beta}$, and log weights, $w_j^{\text{log}} = -\log\widehat{p}_j$, with $\widehat{p}$ estimated on the training set. For power-law weights, we use $\beta = 0.5$. This kind of weighting aims to put more emphasis on less frequent labels.
- TOP-K+$\ell_{\text{FOCAL}}$, TOP-K+$\ell_{\text{ASYM}}$: multi-label focal loss and asymmetric loss (Lin et al., 2017; Ridnik et al., 2021) are variants of BCE loss, commonly used in multi-label classification to improve classification performance on harder, less frequent labels. Here, we train models using these losses and select $k$ labels with the highest output scores.

For all baselines and MACRO-R$_{\text{PRIOR}}$, we always train the label probability estimator on the whole training set. For MACRO-P$_{\text{FW}}$, MACRO-R$_{\text{FW}}$, and MACRO-F1$_{\text{FW}}$, we tested different ratios (50/50 or 75/25) of splitting training data into sets used for training the label probability estimators and estimating confusion matrix $C$, as well as a variant where we use the whole training set for both steps. We also investigated two strategies for initialization of classifier $h$ by either using equal weights (resulting in a TOP-K classifier) or random weights. Finally, we terminate the algorithm if we do not observe sufficient improvement in the objective. In practice, we found that Frank-Wolfe converges within 3–10 iterations. Because of the nature of the random classifier, we repeat the inference on the test set 10 times and report the mean results. In Table 2 we present the variant achieving the best results, and report all the results including standard deviations, running times, number of Frank-Wolfe iterations in Appendix G.

The randomized classifiers obtained via the Frank-Wolfe algorithm achieve, in most cases, the best results for the measure they aim to optimize, at the cost of loosing on some instance-wise measures. However, they sometimes fail to obtain the best results on the largest dataset, where the majority of labels have only a few (less than 10) positive instances in the training set, preventing them from obtaining accurate estimates of $\eta$ and $C$. In this case, simple heuristics like TOP-K+$w^{\text{POW}}$ might work better. Popular Focal loss and Asymmetric loss preserve the performance on instance-wise metrics, but improvement on the macro measures is usually small. It is also worth noting that, as expected, MACRO-R$_{\text{FW}}$ recovers the solution of MACRO-R$_{\text{PRIOR}}$ in all cases.

## 7    CONCLUSIONS

In this paper, we have focused on developing a consistent algorithm for complex macro-at-$k$ metrics in the framework of population utility (PU). Our main results have been obtained by following the line of research conducted in the context of multi-class classification with additional constraints. However, these previous works do not address the specific scenario of budgeted predictions at $k$, which commonly arises in multi-label classification problems. For the complex macro-at-$k$ metrics, we have introduced a consistent Frank-Wolfe algorithm, which is capable of finding an optimal randomized classifier by transforming the problem of optimizing over classifiers to optimizing over the set of feasible confusion matrices and using the fact that the optimal classifier optimizes (unknown) linear confusion-matrix. Our empirical studies show that the introduced approach effectively optimizes macro-measures and it scales to even larger datasets with thousands of labels.

## ACKNOWLEDGMENTS

A part of computational experiments for this paper had been performed in Poznan Supercomputing and Networking Center. We want to acknowledge the support of Academy of Finland via grants 347707 and 348215.

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

## A  MADOW'S SAMPLING SCHEME

In this section we present a sampling scheme for the following sampling problem: given a real-valued vector $\boldsymbol{\pi} \in [0, 1]^m$ of marginal probabilities with $\|\boldsymbol{\pi}\|_1 = k$, sample binary vectors $\widehat{\boldsymbol{y}} \in \{0, 1\}^m$ such that the distribution of $\widehat{\boldsymbol{y}}$ has $\boldsymbol{\pi}$ as the marginals, $\mathbb{E}[\widehat{\boldsymbol{y}}] = \boldsymbol{\pi}$

**Theorem A.1.** *Let $m \geq 1$. Given a vector $\boldsymbol{\pi} \in [0, 1]^m$ satisfying $\|\boldsymbol{\pi}\|_1 = k$, Algorithm 2 returns a randomized binary vector $\widehat{\boldsymbol{y}} \in \{0, 1\}^m$ of size $k$, $\|\widehat{\boldsymbol{y}}\|_1 = k$, with marginals given by $\boldsymbol{\pi}$, $\mathbb{E}[\widehat{\boldsymbol{y}}] = \boldsymbol{\pi}$. The algorithm runs in $\mathcal{O}(m)$ time.*

---

**Algorithm 2** Madow's sampling scheme

---

**Require:** Vector of marginals $\boldsymbol{\pi} \in [0, 1]^m$ with $\|\boldsymbol{\pi}\|_1 = k$
**Ensure:** A random vector $\widehat{\boldsymbol{y}} \in \{0, 1\}^m$ with $\|\widehat{\boldsymbol{y}}\|_1 = k$ such that $\mathbb{E}[\widehat{\boldsymbol{y}}] = \boldsymbol{\pi}$
1: Compute $\Pi_0 = 0$ and $\Pi_j = \Pi_{j-1} + \pi_j$ for $j = 1, \ldots, m$
2: Sample a uniformly distributed random variable $U$ from the interval $[0, 1]$
3: $\widehat{\boldsymbol{y}} = \mathbf{0}$
4: **for** $i \in \{0, 1, \ldots, k - 1\}$ **do**
5:     Select $j$ such that $\Pi_{j-1} < U + i \leq \Pi_j$
6:     Set $\widehat{y}_j = 1$
7: **return** $\widehat{\boldsymbol{y}}$

---

The algorithm is due to Madow (Madow, 1949; Mukhopadhyay et al., 2022), and the considered sampling problem has been studied in the statistical literature under the name *unequal probability sampling design* (Hanif & Brewer, 1980). Below we give a simple proof of correctness of the algorithm for completeness.

*Proof.* First note that for any $i \in \{0, 1, \ldots, k - 1\}$, there exists unique $j$ for which $\Pi_{j-1} < U + i \leq \Pi_j$. This is because due to $\sum_{j=1}^{m} \pi_j = k$, the intervals $(\Pi_0, \Pi_1], (\Pi_1, \Pi_2], \ldots, (\Pi_{m-1}, \Pi_m]$ are disjoint and cover $(0, k]$, whereas $U + i \in (0, k]$ with probability one. Furthermore, the algorithm will select distinct $j$'s for distinct $i$'s. This is because the condition $\Pi_{j-1} < U + i \leq \Pi_j$ is equivalent to $i \in (\Pi_{j-1} - U, \Pi_j - U]$, and the interval $(\Pi_{j-1} - U, \Pi_j - U]$ have width $\pi_j \leq 1$ and thus can contain at most one integer. So the algorithm will return $\widehat{\boldsymbol{y}}$ with exactly $k$ ones.

Since $\mathbb{E}[\widehat{y}_j] = \mathbb{P}[\widehat{y}_j = 1]$, we need to show that this probability is equal to $\pi_j$ for each $j$. We have

$$\mathbb{P}[\widehat{y}_j = 1] = \mathbb{P}\left[U \in \bigcup_{i=0}^{k-1} (\Pi_{j-1} - i, \Pi_j - i]\right]$$

$$= (0, 1] \cap \bigcup_{i=0}^{k-1} (\Pi_{j-1} - i, \Pi_j - i] = \Pi_j - \Pi_{j-1} = \pi_j. \tag{11}$$

$\square$

The theorem and the algorithm from its proof can then be used to generate prediction vectors independently for any instance of interest $\boldsymbol{x}$ by setting $\boldsymbol{\pi} = \boldsymbol{h}(\boldsymbol{x})$.

## B  THE OPTIMAL CLASSIFIER FOR LINEAR METRICS

**Theorem 4.1.** *The optimal classifier $\boldsymbol{h}^\star := \operatorname{argmax}_{\boldsymbol{h} \in \mathcal{H}} \Psi(\boldsymbol{h})$ for $\Psi(\boldsymbol{h}) = \mathbf{G} \cdot \mathbf{C}(\boldsymbol{h})$ is given by*

$$\boldsymbol{h}^\star(\boldsymbol{x}) = \operatorname{top}_k(\boldsymbol{a} \odot \boldsymbol{\eta}(\boldsymbol{x}) + \boldsymbol{b}), \tag{5}$$

*where $\odot$ denotes the coordinate-wise product of vectors, while the vectors $\boldsymbol{a}$ and $\boldsymbol{b}$ are given by:*

$$a_j = G_{00}^j + G_{11}^j - G_{01}^j - G_{10}^j, \qquad b_j = G_{01}^j - G_{00}^j, \tag{6}$$

*and $\operatorname{top}_k(\boldsymbol{v})$ returns a $k$-hot vector extracting the top $k$ largest entries of $\boldsymbol{v}$ (ties broken arbitrarily).*

*Proof.* The linear metric is *decomposable over instances* as:

$$\boldsymbol{G}^j \cdot \boldsymbol{C}^j(h_j) = G^j_{00} \, \mathbb{E}[(1 - \eta_j(\boldsymbol{x}))(1 - h_j(\boldsymbol{x}))] + G^j_{01} \, \mathbb{E}[(1 - \eta_j(\boldsymbol{x}))h_j(\boldsymbol{x})]$$
$$+ G^j_{10} \, \mathbb{E}[\eta_j(\boldsymbol{x})(1 - h_j(\boldsymbol{x}))] + G^j_{11} \, \mathbb{E}[\eta_j(\boldsymbol{x})h_j(\boldsymbol{x})]$$
$$= \mathbb{E}[(a_j \eta_j(\boldsymbol{x}) + b_j)h_j(\boldsymbol{x})] + r_j, \tag{12}$$

where $a_j$ and $b_j$ are as stated in the theorem, while

$$r_j = G^j_{00} \, \mathbb{E}[1 - \eta_j(\boldsymbol{x})] + G^j_{10} \, \mathbb{E}[\eta_j(\boldsymbol{x})]. \tag{13}$$

Thus, we can rewrite the objective as:

$$\Psi(\boldsymbol{h}) = \sum_j \boldsymbol{G}^j \cdot \boldsymbol{C}^j(h_j) = \mathbb{E}\left[ \sum_{j=1}^m (a_j \eta_j(\boldsymbol{x}) + b_j)h_j(\boldsymbol{x}) \right] + R, \tag{14}$$

where $R = r_1 + \ldots + r_m$ does not depend on $\boldsymbol{h}$. For each $\boldsymbol{x} \in \mathcal{X}$, the objective can be independently maximized by the choice of $\boldsymbol{h}(\boldsymbol{x}) \in \Delta^k_m$ which maximizes $\sum_{j=1}^m (a_j \eta_j(\boldsymbol{x}) + b_j)h_j(\boldsymbol{x})$. But this is achieved by sorting $a_j \eta_j(\boldsymbol{x}) + b_j$ in descending order, and setting $h_j(\boldsymbol{x}) = 1$ for the top $k$ coordinates, and $h_j(\boldsymbol{x}) = 0$ for the remaining coordinates (with ties broken arbitrarily). $\qquad \square$

Let us notice that coefficients analogous to our $a_j$ and $b_j$ can also be found in the cost-sensitive prediction rule in binary classification (Elkan, 2001; Natarajan et al., 2018).

## C   THE OPTIMAL CLASSIFIER FOR GENERAL METRICS

In this section, we prove the existence and the form of the optimal classifier. Our results extend past results on binary classification (Koyejo et al., 2014) and multi-class classification (Narasimhan et al., 2015). We first show that the set of confusion tensors achievable by randomized $k$-budgeted classifiers is a compact set. Then, the statement of the main theorem follows from the first-order optimality conditions as well as the absolute continuity of marginal vector $\boldsymbol{\eta}(\boldsymbol{x})$. We stress that the results here are general and applicable to any mutli-label utility satisfying Assumption 4.3, which need not necessarily be a macro-averaged utility.

We remind that the set of confusion tensors achievable by randomized $k$-budgeted classifiers on distribution $\mathbb{P}$, is denoted as

$$\mathcal{C}_{\mathbb{P}} = \Big\{ \mathbf{C}(\boldsymbol{h}) \, : \, \boldsymbol{h} \in \mathcal{H} \Big\}, \tag{15}$$

and that optimizing the metric $\Psi(\boldsymbol{h})$ over $\boldsymbol{h} \in \mathcal{H}$ is equivalent to optimizing $\Psi(\mathbf{C})$ over $\mathbf{C} \in \mathcal{C}_{\mathbb{P}}$.

**Lemma C.1.** *$\mathcal{C}_{\mathbb{P}}$ is a convex set.*

*Proof.* Take any $\mathbf{C}_1, \mathbf{C}_2 \in \mathcal{C}_{\mathbb{P}}$ and any $\lambda \in [0, 1]$, and we show that $\mathbf{C}_\lambda = \lambda \mathbf{C}_1 + (1 - \lambda)\mathbf{C}_2 \in \mathcal{C}_{\mathbb{P}}$. Since $\mathbf{C}_1, \mathbf{C}_2 \in \mathcal{C}_{\mathbb{P}}$, there exist $k$-budgeted randomized classifiers $\boldsymbol{h}_1$ and $\boldsymbol{h}_2$, such that $\mathbf{C}_1 = \mathbf{C}(\boldsymbol{h}_1)$ and $\mathbf{C}_2 = \mathbf{C}(\boldsymbol{h}_2)$. Take $\boldsymbol{h}_\lambda$ defined as $\boldsymbol{h}_\lambda(\boldsymbol{x}) = \lambda \boldsymbol{h}_1(\boldsymbol{x}) + (1 - \lambda)\boldsymbol{h}_2(\boldsymbol{x})$ for any $\boldsymbol{x} \in \mathcal{X}$. Since $\Delta^k_m$ is convex and $\boldsymbol{h}_1(\boldsymbol{x}), \boldsymbol{h}_2(\boldsymbol{x}) \in \Delta^k_m$ for all $\boldsymbol{x} \in \mathcal{X}$, it also holds that $\boldsymbol{h}_\lambda(\boldsymbol{x}) \in \Delta^k_m$ for all $\boldsymbol{x} \in \mathcal{X}$, so $\boldsymbol{h}_\lambda$ is also $k$-budgeted randomized classifier. Since the confusion tensor is linear in predictions, we have $\mathbf{C}(\boldsymbol{h}_\lambda) = \lambda \mathbf{C}(\boldsymbol{h}_1) + (1 - \lambda)\mathbf{C}(\boldsymbol{h}_2) = \mathbf{C}_\lambda$, which proves that $\mathbf{C}_\lambda \in \mathcal{C}_{\mathbb{P}}$. $\qquad \square$

We now argue that for the analysis of $\mathcal{C}_{\mathbb{P}}$, it suffices to consider classifiers of the form $\boldsymbol{h} = \boldsymbol{f} \circ \boldsymbol{\eta}$, i.e. $\boldsymbol{h}(\boldsymbol{x}) = \boldsymbol{f}(\boldsymbol{\eta}(\boldsymbol{x}))$ for some function $\boldsymbol{f} \colon [0, 1]^m \to \Delta^k_m$.

**Lemma C.2.** *For any $\boldsymbol{h} \in \mathcal{H}$, there exists function $\boldsymbol{f} \colon [0, 1]^m \to \Delta^k_m$ such that $\boldsymbol{h}$ and $\boldsymbol{f} \circ \boldsymbol{\eta}$ have the same confusion tensors.*

*Proof.* If $\boldsymbol{h}$ is not of the form $\boldsymbol{f} \circ \boldsymbol{\eta}$, we define function $\boldsymbol{f}$ as:

$$\boldsymbol{f}(\boldsymbol{\eta}) = \mathbb{E}[\boldsymbol{h}(\boldsymbol{x})|\boldsymbol{\eta}(\boldsymbol{x}) = \boldsymbol{\eta}]. \tag{16}$$

Due to convexity of $\Delta_m^k$, we have $\boldsymbol{f}(\boldsymbol{\eta}) \in \Delta_m^k$. Moreover, it is easy to see that $\mathbf{C}(\boldsymbol{h}) = \mathbf{C}(\boldsymbol{f} \circ \boldsymbol{\eta})$; for instance,

$$
\begin{aligned}
C_{11}^j(h_j) &= \mathbb{E}[\eta_j(\boldsymbol{x}) h_j(\boldsymbol{x})] = \mathbb{E}[\mathbb{E}[\eta_j(\boldsymbol{x}) h_j(\boldsymbol{x}) \mid \boldsymbol{\eta}(\boldsymbol{x}) = \boldsymbol{\eta}]] \\
&= \mathbb{E}[\eta_j \, \mathbb{E}[h_j(\boldsymbol{x}) \mid \boldsymbol{\eta}(\boldsymbol{x}) = \boldsymbol{\eta}]] = \mathbb{E}[\eta_j f_j(\boldsymbol{\eta})] \\
&= \mathbb{E}[\eta_j(\boldsymbol{x}) f_j(\boldsymbol{\eta}(\boldsymbol{x}))] = C_{11}^j(f_j \circ \boldsymbol{\eta}),
\end{aligned}
\tag{17}
$$

etc. $\qquad \square$

Hence, any confusion tensor achievable by some $\boldsymbol{h}$ is also achievable by some $\boldsymbol{f} \circ \boldsymbol{\eta}$, so that the set of achievable confusion tensors can be written as $\mathcal{C}_{\mathbb{P}} = \{\mathbf{C}(\boldsymbol{f} \circ \boldsymbol{\eta}) \colon \boldsymbol{f} \in \mathcal{F}\}$, where we denote $\mathcal{F} = \{\boldsymbol{f} \colon [0,1]^m \to \Delta_m^k\}$. From this moment on, we thus, without loss of generality, consider optimizing the metrics over functions $\boldsymbol{f} \in \mathcal{F}$ of random vector $\boldsymbol{\eta}$, and make the relation $\boldsymbol{h} = \boldsymbol{f} \circ \boldsymbol{\eta}$ implicit, writing $\Psi(\boldsymbol{f})$ for $\Psi(\boldsymbol{f} \circ \boldsymbol{\eta})$ and using $\mathbf{C}(\boldsymbol{f})$ to denote the confusion tensors $\mathbf{C}(\boldsymbol{f} \circ \boldsymbol{\eta})$, that is

$$
\mathbf{C}(\boldsymbol{f}) = \left( \boldsymbol{C}^1(f_1), \ldots, \boldsymbol{C}^m(f_m) \right),
\tag{18}
$$

where

$$
\boldsymbol{C}^j(f_j) = \begin{pmatrix} \mathbb{E}_{\boldsymbol{\eta}}[(1 - \eta_j)(1 - f_j(\boldsymbol{\eta}))] & \mathbb{E}_{\boldsymbol{\eta}}[\eta_j(1 - f_j(\boldsymbol{\eta}))] \\ \mathbb{E}_{\boldsymbol{\eta}}[(1 - \eta_j) f_j(\boldsymbol{\eta})] & \mathbb{E}_{\boldsymbol{\eta}}[\eta_j f_j(\boldsymbol{\eta})] \end{pmatrix}.
\tag{19}
$$

**Lemma C.3.** *The mapping $\boldsymbol{f} \mapsto \mathbf{C}(\boldsymbol{f})$ is continuous: for any $\boldsymbol{f}, \boldsymbol{f}' \in \mathcal{F}$*

$$
\|\mathbf{C}(\boldsymbol{f}) - \mathbf{C}(\boldsymbol{f}')\|_F \le \sqrt{2 \mathbb{E}_{\boldsymbol{\eta}} \left[ \|\boldsymbol{f}(\boldsymbol{\eta}) - \boldsymbol{f}'(\boldsymbol{\eta})\|_2^2 \right]},
\tag{20}
$$

*where $\|\mathbf{C}(\boldsymbol{f}) - \mathbf{C}(\boldsymbol{f}')\|_F := \sqrt{\sum_{j=1}^m \|\boldsymbol{C}^j(f_j) - \boldsymbol{C}^j(f_j')\|_F^2}$*

*Proof.* Fix $j \in [m]$. Using $\delta_j(\boldsymbol{\eta}) = f_j(\boldsymbol{\eta}) - f_j'(\boldsymbol{\eta})$, we have from the definition:

$$
\begin{aligned}
\boldsymbol{C}^j(f_j) - \boldsymbol{C}^j(f_j') &= \begin{pmatrix} \mathbb{E}_{\boldsymbol{\eta}}[-(1 - \eta_j)\delta_j(\boldsymbol{\eta})] & \mathbb{E}_{\boldsymbol{\eta}}[-\eta_j \delta_j(\boldsymbol{\eta})] \\ \mathbb{E}_{\boldsymbol{\eta}}[(1 - \eta_j)\delta_j(\boldsymbol{\eta})] & \mathbb{E}_{\boldsymbol{\eta}}[\eta_j \delta_j(\boldsymbol{\eta})] \end{pmatrix} \\
&= \mathbb{E}_{\boldsymbol{\eta}} \left[ \delta_j(\boldsymbol{\eta}) \begin{pmatrix} -(1 - \eta_j) & -\eta_j \\ 1 - \eta_j & \eta_j \end{pmatrix} \right].
\end{aligned}
\tag{21}
$$

Since the squared Frobenious norm $\boldsymbol{X} \mapsto \|\boldsymbol{X}\|_F^2$ is convex, we can use Jensen's inequality $\|\mathbb{E}[\boldsymbol{X}]\|_F^2 \le \mathbb{E}[\|\boldsymbol{X}\|_F^2]$ to get

$$
\begin{aligned}
\|\boldsymbol{C}^j(f_j) - \boldsymbol{C}^j(f_j')\|_F^2 &\le \left\| \mathbb{E}_{\boldsymbol{\eta}} \left[ \delta_j(\boldsymbol{\eta}) \begin{pmatrix} -(1 - \eta_j) & -\eta_j \\ 1 - \eta_j & \eta_j \end{pmatrix} \right] \right\|_F^2 \\
&\le \mathbb{E}_{\boldsymbol{\eta}} \left[ (\delta_j(\boldsymbol{\eta}))^2 \left\| \begin{pmatrix} -(1 - \eta_j) & -\eta_j \\ 1 - \eta_j & \eta_j \end{pmatrix} \right\|_F^2 \right] \le 2 \mathbb{E}_{\boldsymbol{\eta}} \left[ (\delta_j(\boldsymbol{\eta}))^2 \right],
\end{aligned}
\tag{22}
$$

where we used

$$
\left\| \begin{pmatrix} -(1 - \eta_j) & -\eta_j \\ 1 - \eta_j & \eta_j \end{pmatrix} \right\|_F^2 = 2 \left( (1 - \eta_j)^2 + \eta_j^2 \right) \le 2 \max_{x \in [0,1]} ((1 - x)^2 + x^2) = 2.
\tag{23}
$$

Summing the inequality over $j = 1, \ldots, m$ and taking square root on both sides finishes the proof. $\qquad \square$

We will now show that the set of achievable confusion tensors $\mathcal{C}_{\mathbb{P}}$ is compact. To this end, we first prove a result from the functional analysis (which is false without the convexity assumption).

**Lemma C.4.** *Let $\mathcal{L} : \mathbb{H} \to \mathcal{V}$ be continuous affine operator between a Hilbert space $\mathbb{H}$ and a finite dimensional vector space $\mathcal{V}$. If $\mathcal{S} \subset \mathbb{H}$ is closed, bounded, and convex, then $\mathcal{L}(\mathcal{S})$ is compact.*

*Proof.* Observe that it suffices to prove this when $\mathcal{L}$ is linear since being compact is translation invariant.

The proof is inspired by an answer to a related question on Mathematics Stack Exchange. It suffices to prove every $\mathcal{L}(x_n)$ sequence in $\mathcal{L}$ has a convergent subsequence whose limit is in $\mathcal{L}(\mathcal{S})$.

By the Banach–Alaoglu theorem, balls in Hilbert spaces are weakly compact. For the convenience of the reader we will sketch this proof. Recall that weak convergence $x_n \to x$ in $\mathbb{H}$ means that for all linear functionals $\phi \in \mathbb{H}^*$ there is convergence $\phi(x_n) \to \phi(x)$. Likewise weakly compact means every sequence has a subsequence that converges weakly. Now onto the proof.

Let $x_n$ be a bounded sequence in $\mathbb{H}$. Let $\{e_1, e_2, \dots\}$ be a Hilbert basis for $\mathbb{H}$ and the dual vectors $\{\phi_1, \phi_2, \dots\}$ a Hilbert basis for $\mathbb{H}^*$ where $\phi_i(x) = \langle x, e_i \rangle$. Now apply the diagonal proof method, as in the Arzelà-Ascoli theorem, of successively passing to subsequences. Since the sequence is bounded we know that $\phi_1(x_n)$ is bounded in $\mathbb{R}$ and hence we can extract a subsequence $x_n$ so that $\phi_1(x_n) \to a_1$ where we may keep $x_1$. Now on this subsequence do the same for $\phi_2(x_n) \to a_2$ but keep $x_1$ and $x_2$. Continue this process, where at the $m$-th step one keeps the first $m$ terms from the previous subsequence. The resulting diagonal subsequence $x_n$ is such that $\phi_i(x_n) \to a_i$ for each $i = 1, 2, \dots$. The element $x = \sum_{i=1}^{\infty} a_i e_i$ is in $\mathbb{H}$ (by Bessel's inequality, the weak convergence results, and the fact that the original sequence was bounded). It remains to verify that $x_n \to x$ weakly, but for this it suffices to check $\phi_i(x_n) \to \phi_i(x)$ and by design this is the case.

Now, returning to the proof of the lemma since $\mathcal{S} \subset \mathbb{H}$ is bounded, it is contained in a ball, and hence by passing to a subsequence we have $x_n \to x$ in the weak topology for some $x \in \mathbb{H}$. Furthermore $x \in \mathcal{S}$. If $x$ wasn't, then since $\mathcal{S}$ is closed and convex, by the Hahn–Banach separation theorem there is a separating hyperplane $\phi \in \mathbb{H}^*$ so $\phi(x) < \inf \phi(\mathcal{S})$. But this contradicts that the weak convergence $x_n \to x$ since $x_n \in \mathcal{S}$.

So it remains to prove convergence $\mathcal{L}(x_n) \to \mathcal{L}(x)$. Since $\mathcal{L}$ is continuous we have convergence $\mathcal{L}(x_n) \to \mathcal{L}(x)$ in the weak topology, but this implies normal convergence since $\mathcal{V}$ is finite dimensional. $\qquad\square$

**Lemma C.5.** $\mathcal{C}_{\mathbb{P}}$ *is a compact set.*

*Proof.* To show that $\mathcal{C}_{\mathbb{P}}$ is compact, we will invoke Lemma C.4. To place ourselves in its setting, let the Hilbert space be $\mathbb{H} = L^2([0,1]^m, \mathbb{R}^m, \mu)$ where $\mu$ is the probability measure on $[0,1]^m$ associated with random vector $\boldsymbol{\eta}(\boldsymbol{x})$. The inner product for $\boldsymbol{f}, \boldsymbol{g} : [0,1]^m \to \mathbb{R}^m$ in $\mathbb{H}$ is

$$\langle \boldsymbol{f}, \boldsymbol{g} \rangle = \int_{[0,1]^m} \langle \boldsymbol{f}(\boldsymbol{\eta}), \boldsymbol{g}(\boldsymbol{\eta}) \rangle \, d\mu(\boldsymbol{\eta}) \tag{24}$$

where the inner product inside the integral is the normal dot product in $\mathbb{R}^m$.

We have the affine map defined via (1)

$$\mathcal{L} : \mathbb{H} \to \mathbb{R}^{m \times 2 \times 2} \quad \text{where} \quad \mathcal{L}(\boldsymbol{f}) = \mathbf{C}(\boldsymbol{f} \circ \boldsymbol{\eta}), \tag{25}$$

and let the subset $\mathcal{S} \subset \mathbb{H}$ be

$$\mathcal{S} = \{\boldsymbol{f} \in \mathbb{H} : \boldsymbol{f}([0,1]^m) \subset \Delta_m^k \text{ almost everywhere}\}. \tag{26}$$

Since $\mathcal{C}_{\mathbb{P}} = \mathcal{L}(\mathcal{S})$, it suffices to verify the assumptions in Lemma C.4.

The map $\mathcal{L}$ is continuous by Lemma C.3. The set $\mathcal{S}$ is convex since the set of $\Delta_m^k \subset \mathbb{R}^m$ is convex. Likewise for bounded using also that the we are working with a probability measure: If $\boldsymbol{f} \in \mathcal{S}$, then $\|\boldsymbol{f}(\boldsymbol{\eta})\|^2 \le m$ for all $\boldsymbol{\eta} \in [0,1]^m$ and hence

$$\|\boldsymbol{f}\|^2 = \int_{[0,1]^m} \|\boldsymbol{f}(\boldsymbol{\eta})\|^2 d\mu(\boldsymbol{\eta}) \le \int_{[0,1]^m} m \, d\mu(\boldsymbol{\eta}) = m. \tag{27}$$

Similarly the closedness of $\Delta_m^k \subset \mathbb{R}^m$ translates into the closedness of $\mathcal{S}$ as we now prove. Suppose there is a sequence of $\boldsymbol{f}_n \in \mathcal{S}$ with $\boldsymbol{f}_n \to \boldsymbol{f}$ and $\boldsymbol{f} \notin \mathcal{S}$. This means the set of points

$$A = \{\boldsymbol{\eta} \in [0,1]^m : \boldsymbol{f}(\boldsymbol{\eta}) \notin \Delta_m^k\} \tag{28}$$

that $\boldsymbol{f}$ maps out of $\Delta_m^k$ has positive measure $\mu(A) > 0$. In $\mathbb{R}^m$ there is a well defined distance function $d(\boldsymbol{z}, \Delta_m^k) = \inf_{\boldsymbol{v} \in \Delta_m^k} \|\boldsymbol{z} - \boldsymbol{v}\|$ and for $\epsilon > 0$ define the set

$$A_\epsilon = \{\boldsymbol{\eta} \in [0,1]^m : d(\boldsymbol{f}(\boldsymbol{\eta}), \Delta_m^k) > \epsilon\} \tag{29}$$

Note that $A = \bigcup_{j=1}^{\infty} A_{1/j}$ since $\Delta_m^k$ is closed and hence there is some $\epsilon > 0$ such that $\mu(A_\epsilon) > 0$. Therefore for all $n$

$$
\begin{aligned}
\|\boldsymbol{f} - \boldsymbol{f}_n\|^2 &= \int_{[0,1]^m} \|\boldsymbol{f}(\boldsymbol{\eta}) - \boldsymbol{f}_n(\boldsymbol{\eta})\|^2 d\mu(\boldsymbol{\eta}) \\
&\geq \int_{A_\epsilon} \|\boldsymbol{f}(\boldsymbol{\eta}) - \boldsymbol{f}_n(\boldsymbol{\eta})\|^2 d\mu(\boldsymbol{\eta}) \geq \int_{A_\epsilon} \epsilon^2 \, d\mu(\boldsymbol{\eta}) = \epsilon^2 \mu(A_\epsilon) > 0
\end{aligned}
\tag{30}
$$

where the second inequality uses that $f_n(\boldsymbol{\eta}) \in \Delta_m^k$ almost everywhere. That $\|\boldsymbol{f} - \boldsymbol{f}_n\|^2$ is uniformly bounded away from 0 contradicts that $\boldsymbol{f}_n \to \boldsymbol{f}$ in $\mathbb{H}$. $\qquad\square$

**Theorem 4.4.** *Let the data distribution $\mathbb{P}$ and metric $\Psi$ satisfy Assumption 4.2 and Assumption 4.3 respectively. Then, there exists an optimal $\mathbf{C}^\star \in \mathcal{C}_\mathbb{P}$, that is $\Psi(\mathbf{C}^\star) = \Psi^\star$. Moreover, any classifier $h^\star$ maximizing the linear utility $\mathbf{G} \cdot \mathbf{C}(h)$ over $h \in \mathcal{H}$ with $\mathbf{G} = (\boldsymbol{G}^1, \ldots, \boldsymbol{G}^m)$ given by $\boldsymbol{G}^j = \nabla_{\boldsymbol{C}^j} \Psi(\mathbf{C}^\star)$, also maximizes $\Psi(h)$ over $h \in \mathcal{H}$.*

*Proof.* Let $\mathbf{C}^\star = \operatorname{argmax}_{\mathbf{C} \in \mathcal{C}_\mathbb{P}} \Psi(\mathbf{C})$, which exists by the compactness of $\mathcal{C}_\mathbb{P}$ (Lemma C.5) and the continuity of $\Psi$. By the first order optimality and convexity of $\mathcal{C}_\mathbb{P}$, for all $\mathbf{C} \in \mathcal{C}_\mathbb{P}$

$$
\nabla \Psi(\mathbf{C}^\star) \cdot \mathbf{C}^\star \geq \nabla \Psi(\mathbf{C}^\star) \cdot \mathbf{C}.
\tag{31}
$$

which implies:

$$
\mathbf{C}^\star = \operatorname*{argmax}_{\mathbf{C} \in \mathcal{C}_\mathbb{P}} \mathbf{G} \cdot \mathbf{C}
\tag{32}
$$

for $\mathbf{G} = \nabla \Psi(\mathbf{C}^\star)$.

We now show that $\mathbf{C}^\star$ is the unique optimizer of (32). Using Assumption 4.3 applied to $\mathbf{C}^\star$, for all $j \in [m]$:

$$
\begin{aligned}
\frac{\partial}{\partial \epsilon} \Psi &\left( \boldsymbol{C}^{\star 1}, \ldots, \boldsymbol{C}^{\star j} + \epsilon \begin{pmatrix} 1 & -1 \\ -1 & 1 \end{pmatrix}, \ldots, \boldsymbol{C}^{\star m} \right) \bigg|_{\epsilon=0} = \nabla_{\boldsymbol{C}^j} \Psi(\mathbf{C}^\star) \cdot \begin{pmatrix} 1 & -1 \\ -1 & 1 \end{pmatrix} \\
&= G_{00}^j + G_{11}^j - G_{01}^j - G_{10}^j = a_j > 0,
\end{aligned}
\tag{33}
$$

with coefficients $a_j, j \in [m]$ defined in Theorem 4.1.

Now, since we just showed that $a_j \neq 0$ for all $j$, and $\boldsymbol{\eta}$ has a density, coordinates of $\boldsymbol{a} \odot \boldsymbol{\eta} + \boldsymbol{b}$ are all distinct with probability one. This means that, with probability one, $\operatorname{top}_k(\boldsymbol{a} \odot \boldsymbol{\eta} + \boldsymbol{b})$ is a singleton, and thus the optimizers of the linear utility $\mathbf{G} \cdot \mathbf{C}(h)$ can only differ on a zero measure set, so they all have the same confusion tensor. Thus, $\mathbf{C}^\star$ uniquely maximizes linear utility $\mathbf{G} \cdot \mathbf{C}$ over $\mathbf{C} \in \mathcal{C}_\mathbb{P}$.

This means, however, that any classifier $h^\star$ maximizing $\mathbf{G} \cdot \mathbf{C}(h)$ over $h \in \mathcal{H}$ has $\mathbf{C}(h^\star) = \mathbf{C}^\star$, and thus maximizes $\Psi$. $\qquad\square$

## D  CONSISTENCY OF FRANK-WOLFE

In this section, we provide the formal proof of consistency for the Frank-Wolfe algorithm. We prove convergence for a slightly modified version of Algorithm 1, in which we replace the line-search in line 13 with a fixed schedule, setting

$$
\alpha^i \leftarrow \frac{2}{t+1}.
\tag{34}
$$

For the experiments, we used the line-search instead, as we found it to give slightly better results.

### D.1  VC-DIMENSION LEMMA

**Lemma 5.2** (VC dimension for linear top-k classifiers). *For $\boldsymbol{\eta} \colon \mathcal{X} \longrightarrow [0,1]^m$, define*

$$
\mathcal{H}_{\boldsymbol{\eta}}^j := \bigcup_{\boldsymbol{a}, \boldsymbol{b} \in \mathbb{R}^m} \{h \colon \mathcal{X} \longrightarrow \{0,1\} : h(\boldsymbol{x}) = \mathbb{1}[j \in \operatorname{top}_k(\boldsymbol{a} \odot \boldsymbol{\eta} + \boldsymbol{b})]\}.
\tag{10}
$$

*The VC-complexity of this class is $\mathrm{VC}(\mathcal{H}_{\boldsymbol{\eta}}^j) \leq 6m \log(em)$.*

*Proof.* For any given $\boldsymbol{a}, \boldsymbol{b}$, the hypothesis predicts one, $h^j(\boldsymbol{x}) = 1$, iff exists a set of $m - k$ indices $\mathcal{I} \subset [m]$ with $|\mathcal{I}| = m - k$, $j \notin \mathcal{I}$, such that for all $i \in \mathcal{I}$ the score $a_i \eta_i + b_i \leq a_j \eta_j + b_j$ is not greater than the score of label $j$.

This computation can be realized as a two-layer network. In the first layer $\boldsymbol{z}$, we calculate an indicator to determine which labels' scores are below the threshold, that is $z_i = \mathbb{1}[(a_i - a_j)\eta_i + (b_i - b_j)]$. Then, for the output, we threshold the sum of all the intermediate units to determine if $j$ is predicted:

$$h(\boldsymbol{x}) = o(\boldsymbol{z}) := \mathbb{1}\Big[\sum_{i \neq j} z_i \geq m - k\Big]. \tag{35}$$

The resulting network has $2(m-1)$ edges and $m-1$ computation nodes. If we allow the output node to be more general—a generic linear threshhold function—, the VC-dimension of this extended function class $\mathcal{H}'$ can only grow. For this extended class, we can apply (Baum & Haussler, 1988, Corollary 3), which gives an upper bound for the VC-dimension of

$$\text{VC}(\mathcal{H}^j) \leq \text{VC}(\mathcal{H}') \leq 2(m - 1 + 2(m-1))\log(e(m-1)) \leq 6m\log(em). \tag{36}$$

$\square$

## D.2 ADDITIONAL LEMMAS

Before going into the main proof of Theorem 5.1, we provide two more helper lemmas:

**Lemma D.1** (Regret for Linear Macro Measures). *Let* $\mathbf{G}$ *be a* linear *macro-measure, that is,*

$$\mathbf{G}(\boldsymbol{h}; \boldsymbol{\eta}) = m^{-1} \sum_{j=1}^{m} \mathbb{E}_{\boldsymbol{x}}\Big[G_{00}^j(1 - \eta_j(\boldsymbol{x}))(1 - h_j(\boldsymbol{x}))$$
$$+ G_{01}^j(1 - \eta_j(\boldsymbol{x}))h_j(\boldsymbol{x}) + G_{10}^j \eta_j(\boldsymbol{x})(1 - h_j(\boldsymbol{x})) + G_{11}^j \eta_j(\boldsymbol{x})h_j(\boldsymbol{x})\Big]. \tag{37}$$

*Let* $\boldsymbol{h}^\star(\boldsymbol{x}) := \text{argmax}_{\boldsymbol{h}} \, \mathbf{G}(\boldsymbol{h}; \boldsymbol{\eta})$, *and* $\widehat{\boldsymbol{h}}(\boldsymbol{x}) := \text{argmax}_{\boldsymbol{h}} \, \mathbf{G}(\boldsymbol{h}; \widehat{\boldsymbol{\eta}})$. *Then*

$$\mathbf{G}(\boldsymbol{h}^\star; \boldsymbol{\eta}) - \mathbf{G}(\widehat{\boldsymbol{h}}; \boldsymbol{\eta}) \leq m^{-1} \max_j \|\boldsymbol{G}^j\|_{1,1} \, \mathbb{E}_{\boldsymbol{x}}[\|\boldsymbol{\eta}(\boldsymbol{x}) - \widehat{\boldsymbol{\eta}}(\boldsymbol{x})\|_1]. \tag{38}$$

*Proof.* As $\mathbf{G}(\boldsymbol{h}; \boldsymbol{\eta})$ is an affine function in its second argument, we can simplify differences to

$$\mathbf{G}(\boldsymbol{h}; \boldsymbol{\eta}) - \mathbf{G}(\boldsymbol{h}; \widehat{\boldsymbol{\eta}}) = m^{-1} \sum_{j=1}^{m} \mathbb{E}_{\boldsymbol{x}}\Big[-G_{00}^j(\eta_j - \widehat{\eta}_j)(1 - h_j) - G_{01}^j(\eta_j - \widehat{\eta}_j)h_j$$
$$+ G_{10}^j(\eta_j - \widehat{\eta}_j)(1 - h_j) + G_{11}^j(\eta_j - \widehat{\eta}_j)h_j\Big]$$
$$= m^{-1} \sum_{j=1}^{m} \mathbb{E}_{\boldsymbol{x}}\Big[(\eta_j - \widehat{\eta}_j)\left((G_{11}^j - G_{01}^j)h_j + (G_{10}^j - G_{00}^j)(1 - h_j)\right)\Big]. \tag{39}$$

We can use this property to bound the regret of $\widehat{\boldsymbol{h}}$ as

$$\mathbf{G}(\boldsymbol{h}^\star; \boldsymbol{\eta}) - \mathbf{G}(\widehat{\boldsymbol{h}}; \boldsymbol{\eta}) = \mathbf{G}(\boldsymbol{h}^\star; \boldsymbol{\eta}) - \mathbf{G}(\boldsymbol{h}^\star; \widehat{\boldsymbol{\eta}}) + \mathbf{G}(\boldsymbol{h}^\star; \widehat{\boldsymbol{\eta}}) - \mathbf{G}(\widehat{\boldsymbol{h}}; \boldsymbol{\eta})$$
$$\leq \mathbf{G}(\boldsymbol{h}^\star; \boldsymbol{\eta}) - \mathbf{G}(\boldsymbol{h}^\star; \widehat{\boldsymbol{\eta}}) + \mathbf{G}(\widehat{\boldsymbol{h}}; \widehat{\boldsymbol{\eta}}) - \mathbf{G}(\widehat{\boldsymbol{h}}; \boldsymbol{\eta})$$
$$= m^{-1} \sum_{j=1}^{m} \mathbb{E}_{\boldsymbol{x}}\Big[(\eta_j - \widehat{\eta}_j)\left((G_{11}^j - G_{01}^j)(h_j^\star - \widehat{h}_j) + (G_{10}^j - G_{00}^j)(\widehat{h}_j - h_j^\star)\right)\Big]$$
$$= m^{-1} \sum_{j=1}^{m} (G_{11}^j - G_{01}^j - G_{10}^j + G_{00}^j)\, \mathbb{E}_{\boldsymbol{x}}\Big[(\eta_j - \widehat{\eta}_j)(h_j^\star - \widehat{h}_j)\Big] \tag{40}$$

As $h_j \in [0, 1]$, we can bound $(\eta_j - \widehat{\eta}_j)(h_j^\star - \widehat{h}_j) \leq |(\eta_j - \widehat{\eta}_j)|$, resulting in

$$\mathbf{G}(\boldsymbol{h}^\star; \boldsymbol{\eta}) - \mathbf{G}(\widehat{\boldsymbol{h}}; \boldsymbol{\eta}) \leq m^{-1} \sum_{j=1}^{m} (G_{11}^j - G_{01}^j - G_{10}^j + G_{00}^j)\, \mathbb{E}_{\boldsymbol{x}}[|\eta_j - \widehat{\eta}_j|] \tag{41}$$

Using the notation of Theorem 4.1, we set $a_j = G_{11}^j - G_{01}^j - G_{10}^j + G_{00}^j$, so that we can further bound

$$\mathbf{G}(\boldsymbol{h}^\star; \boldsymbol{\eta}) - \mathbf{G}(\widehat{\boldsymbol{h}}; \boldsymbol{\eta}) \leq m^{-1} \max_j |a_j| \sum_{j=1}^{m} \mathbb{E}_{\boldsymbol{x}}[|\eta_j - \widehat{\eta}_j|] = m^{-1} \max_j |a_j|\, \mathbb{E}_{\boldsymbol{x}}[\|\boldsymbol{\eta} - \widehat{\boldsymbol{\eta}}\|_1] \tag{42}$$

Using

$$\max_j |a_j| \leq \max_j \|\boldsymbol{G}^j\|_{1,1} \tag{43}$$

yields the claim. $\qquad\square$

**Lemma D.2** (Uniform Convergence of Multi-label Confusion Tensors). *For* $\boldsymbol{\eta} \colon \mathcal{X} \longrightarrow [0,1]^m$, *let*

$$\mathcal{H}_{\boldsymbol{\eta}} := \bigcup_{\boldsymbol{a},\boldsymbol{b}\in\mathbb{R}^m} \{\boldsymbol{h} \colon \mathcal{X} \longrightarrow \{0,1\}^m \colon \boldsymbol{h}(\boldsymbol{x}) = \mathrm{top}_k\, \boldsymbol{a} \odot \boldsymbol{\eta} + \boldsymbol{b}\}, \tag{44}$$

*and let* $\mathcal{S} \in (\mathcal{X} \times \{0,1\}^m)^n$ *be an i.i.d. sample. Then for any* $\delta \in (0,1]$, *with probability at least* $1 - \delta$, *we have*

$$\sup_{\boldsymbol{h}\in\mathcal{H}_{\boldsymbol{\eta}}} \|\mathbf{C}(\boldsymbol{h},\mathbb{P}) - \widehat{\mathbf{C}}(\boldsymbol{h},\mathcal{S})\|_{\infty} \leq \tilde{\mathcal{O}}\left(\sqrt{\frac{m \cdot \log m \cdot \log n - \log \delta}{n}}\right). \tag{45}$$

*Proof.* Instead of showing uniform convergence for the entries of the confusion tensor directly, we show it for accuracy (0-1-error) $\mathrm{acc}^j = C_{11}^j + C_{00}^j$, condition positive rate $\mathrm{q}^j = C_{01}^j + C_{11}^j$ and predicted positive rate $\mathrm{p}^j = C_{10}^j + C_{11}^j$, first for a fixed $j \in [m]$.

To handle accuracy and predicted positives, consider

$$\sup_{\boldsymbol{h}\in\mathcal{H}_{\boldsymbol{\eta}}} \left|\mathrm{acc}^j(\boldsymbol{h},\mathbb{P}) - \widehat{\mathrm{acc}}^j(\boldsymbol{h},\mathcal{S})\right| = \sup_{\boldsymbol{h}\in\mathcal{H}_{\boldsymbol{\eta}}} \left|n^{-1}\sum_{i=1}^n \mathbb{1}[Y_{ij} = h_j(\boldsymbol{x}_i)] - \mathbb{P}[y_j = h_j(\boldsymbol{x})]\right|$$

$$= \sup_{h\in\mathcal{H}_{\boldsymbol{\eta}}^j} \left|n^{-1}\sum_{i=1}^n \mathbb{1}[Y_{ij} = h(\boldsymbol{x}_i)] - \mathbb{P}[y_j = h(\boldsymbol{x})]\right| \tag{46}$$

From Lemma 5.2, we know the VC-dimension of $\mathcal{H}_{\boldsymbol{\eta}}^j$ is some finite number $d$, thus, we can employ a standard bound for the 0-1 error to get, with probability $1 - \delta$, that

$$\sup_{\boldsymbol{h}\in\mathcal{H}_{\boldsymbol{\eta}}} |\mathrm{acc}^j(\boldsymbol{h},\mathbb{P}) - \widehat{\mathrm{acc}}^j(\boldsymbol{h},\mathcal{S})| \leq \sqrt{\frac{2d\log(2en/d) + 2\log(4/\delta)}{n}}. \tag{47}$$

As this bound holds for *all* distributions of targets $\boldsymbol{y}$, it holds in particular also for $\boldsymbol{y} \equiv 1$, in which case accuracy turns into predicted positive rate.

Finally, we can bound the error on the condition positive rate simply using Hoeffding's inequality, as it does not depend on the hypothesis $h$. We get, with probability $1 - \delta$

$$\sup_{\boldsymbol{h}\in\mathcal{H}_{\boldsymbol{\eta}}} |\mathrm{q}(\boldsymbol{h},\mathbb{P}) - \widehat{\mathrm{q}}(\boldsymbol{h},\mathcal{S})| \leq \sqrt{\frac{\log(\delta^{-1})}{2n}}. \tag{48}$$

Now we can reconstruct the actual entries of the confusion matrix. For example, the true positive rate as $\mathrm{tp} = \frac{1-\mathrm{acc}-\mathrm{q}-\mathrm{p}}{2}$. Thus, we can union bound, with probability $1 - \delta$

$$\sup_{\boldsymbol{h}\in\mathcal{H}_{\boldsymbol{\eta}}} |\mathrm{tp}^j(\boldsymbol{h},\mathbb{P}) - \widehat{\mathrm{tp}}^j(\boldsymbol{h},\mathcal{S})| \leq \sqrt{\frac{2d\log(2en/d) + 2\log(8/\delta)}{n}} + \sqrt{\frac{\log(3/\delta)}{2n}}. \tag{49}$$

Similar bounds can be constructed for the other entries. Taking a union bound over all $m$ labels:

$$\sup_{\boldsymbol{h}\in\mathcal{H}_{\boldsymbol{\eta}}} \|\mathbf{C}(\boldsymbol{h},\mathbb{P}) - \widehat{\mathbf{C}}(\boldsymbol{h},\mathcal{S})\|_{\infty} \leq \sqrt{\frac{2d\log(2en/d) + 2\log(8m/\delta)}{n}} + \sqrt{\frac{\log(3m/\delta)}{2n}}$$

$$= \sqrt{\frac{12m\log(em)\log(en/(3m(\log(em))) + 2\log(8m/\delta)}{n}} + \sqrt{\frac{\log(3m/\delta)}{2n}}. \tag{50}$$

In order to combine the two square-root terms, we can apply the arithmetic-quadratic mean inequality, to arrive at the claimed bound

$$\sup_{\boldsymbol{h}\in\mathcal{H}_{\boldsymbol{\eta}}} \|\mathbf{C}(\boldsymbol{h},\mathbb{P}) - \widehat{\mathbf{C}}(\boldsymbol{h},\mathcal{S})\|_{\infty} \leq \sqrt{\frac{48m\log(em)\log(en/(3m(\log(em))) + 10\log(\sqrt[5]{3\cdot 8^4}m/\delta)}{2n}}.$$

Finally, using $3m(\log(em)) \geq 1$, we simplify

$$\log(en/(3m(\log(em)))) \leq \log(en), \tag{51}$$

which results in

$$\sup_{\boldsymbol{h} \in \mathcal{H}_{\boldsymbol{\eta}}} \|\mathbf{C}(\boldsymbol{h}, \mathbb{P}) - \widehat{\mathbf{C}}(\boldsymbol{h}, \mathcal{S})\|_\infty \leq \sqrt{\frac{\mathcal{O}(m \log m \log n) + \mathcal{O}(\log(m/\delta))}{n}}. \tag{52}$$

$\square$

### D.3 BOUND FOR LINEAR OPTIMIZATION STEP

The preceding results allow to prove a bound on the approximation error for each linear optimization step that is performed as part of the Frank-Wolfe algorithm:

**Lemma D.3.** *Let $\Psi : \mathcal{C} \longrightarrow \mathbb{R}_{\geq 0}$ be concave over $\mathcal{C}_{\mathbb{P}}$, $\ell$-Lipschitz, and $\beta$-smooth w.r.t. the $\ell_1$-norm. Let $\boldsymbol{h} \in \mathcal{H}$ be some classifier, and denote $\mathbf{G} := \nabla\Psi(\widehat{\mathbf{C}}(\boldsymbol{h}, \mathcal{S}))$. Let $\widehat{\boldsymbol{g}}$ be the deterministic classifier that empirically optimizes the linear objective induced by $\Psi$ according to Theorem 4.1. For two classifiers $\boldsymbol{h}'$ and $\boldsymbol{g}'$, define*

$$\mathfrak{L}_{\mathbb{P}}(\boldsymbol{h}', \boldsymbol{g}') := \mathbf{C}(\boldsymbol{g}', \mathbb{P}) \cdot \nabla\Psi(\mathbf{C}(\boldsymbol{h}', \mathbb{P})), \tag{53}$$

*Then for any $\delta \in (0, 1]$, with probability at least $1 - \delta$ (over draws of $\mathcal{S}$ from $\mathbb{P}^n$), we have*

$$\mathfrak{L}_{\mathbb{P}}(\boldsymbol{h}, \widehat{\boldsymbol{g}}) \geq \max_{\boldsymbol{g}'} \mathfrak{L}_{\mathbb{P}}(\boldsymbol{h}, \boldsymbol{g}') - \epsilon_{\mathcal{S}} \tag{54}$$

*where*

$$\epsilon_{\mathcal{S}} = 8\ell m^{-1} \mathbb{E}_{\boldsymbol{x}}[\|\boldsymbol{\eta}(\boldsymbol{x}) - \widehat{\boldsymbol{\eta}}(\boldsymbol{x})\|_1] + 8m^2\beta \sup_{\boldsymbol{h}' \in \mathcal{H}} \tilde{\mathcal{O}}\left(\sqrt{\frac{m \cdot \log m \cdot \log n - \log \delta}{n}}\right). \tag{55}$$

*Proof.* Define an empirical counterpart to $\mathfrak{L}_{\mathbb{P}}$, the population-level utility of a classifier for an empirically estimated gradient, as

$$\mathfrak{L}_{\mathcal{S}}(\boldsymbol{h}', \boldsymbol{g}') := \mathbf{C}(\boldsymbol{g}', \mathbb{P}) \cdot \nabla\Psi(\widehat{\mathbf{C}}(\boldsymbol{h}', \mathcal{S})), \tag{56}$$

and the (population-level) optimal classifier $\boldsymbol{g}^\star \in \operatorname{argmax}_{\boldsymbol{g}'} \mathfrak{L}_{\mathbb{P}}(\boldsymbol{h}, \boldsymbol{g}')$ for the exact gradient, whose existence is guaranteed by Theorem 4.1. Then we can write

$$= \max_{\boldsymbol{g}'} \mathfrak{L}_{\mathbb{P}}(\boldsymbol{h}, \boldsymbol{g}') - \mathfrak{L}_{\mathbb{P}}(\boldsymbol{h}, \widehat{\boldsymbol{g}}) = \mathfrak{L}_{\mathbb{P}}(\boldsymbol{h}, \boldsymbol{g}^\star) - \mathfrak{L}_{\mathbb{P}}(\boldsymbol{h}, \widehat{\boldsymbol{g}})$$
$$= \mathfrak{L}_{\mathbb{P}}(\boldsymbol{h}, \boldsymbol{g}^\star) - \mathfrak{L}_{\mathcal{S}}(\boldsymbol{h}, \boldsymbol{g}^\star) + \mathfrak{L}_{\mathcal{S}}(\boldsymbol{h}, \boldsymbol{g}^\star) - \mathfrak{L}_{\mathcal{S}}(\boldsymbol{h}, \widehat{\boldsymbol{g}}) + \mathfrak{L}_{\mathcal{S}}(\boldsymbol{h}, \widehat{\boldsymbol{g}}) - \mathfrak{L}_{\mathbb{P}}(\boldsymbol{h}, \widehat{\boldsymbol{g}}) \tag{57}$$

Now we turn to bounding each of these terms. For the second, we get

$$\mathfrak{L}_{\mathcal{S}}(\boldsymbol{h}, \boldsymbol{g}^\star) - \mathfrak{L}_{\mathcal{S}}(\boldsymbol{h}, \widehat{\boldsymbol{g}}) = \mathbf{C}(\boldsymbol{g}^\star, \mathbb{P}) \cdot \mathbf{G} - \mathbf{C}(\widehat{\boldsymbol{g}}, \mathbb{P}) \cdot \mathbf{G}$$
$$\leq \max_{\boldsymbol{g}'} \mathbf{C}(\boldsymbol{g}', \mathbb{P}) \cdot \mathbf{G} - \mathbf{C}(\widehat{\boldsymbol{g}}, \mathbb{P}) \cdot \mathbf{G} \leq 2m^{-1} \max_j \|\boldsymbol{G}^j\|_{1,1} \mathbb{E}_{\boldsymbol{x}}[\|\boldsymbol{\eta}(\boldsymbol{x}) - \widehat{\boldsymbol{\eta}}(\boldsymbol{x})\|_1], \tag{58}$$

where the last step used that $\widehat{\boldsymbol{g}}$ is the empirical maximizer of the linear measure corresponding to $\mathbf{G}$, in order to apply Lemma D.1. Now, if $\Psi$ is $\ell$-Lipschitz w.r.t. the $\ell_1$-norm, then

$$\forall \mathbf{C}' : \ |\nabla\Psi(\mathbf{C}) \cdot \mathbf{C}'| \leq \|\mathbf{C}'\|_1. \tag{59}$$

Let $j \in [m]$, and applying (59) to $\mathbf{C}' = \tilde{\mathbf{C}}$ for which $\tilde{\mathbf{C}}^i = \mathbf{0}$ for all $i \neq j$, and $\tilde{\mathbf{C}}^i = 0.25 \cdot \mathbf{1}$, we get

$$0.25 \boldsymbol{G}^j \cdot \mathbf{1} \leq \ell \ \Leftrightarrow \ \|\boldsymbol{G}^j\|_{1,1} \leq 4\ell. \tag{60}$$

As this holds for all $j$, the upper bound turns into

$$\mathfrak{L}_{\mathcal{S}}(\boldsymbol{h}, \boldsymbol{g}^\star) - \mathfrak{L}_{\mathcal{S}}(\boldsymbol{h}, \widehat{\boldsymbol{g}}) \leq 8\ell m^{-1} \mathbb{E}_{\boldsymbol{x}}[\|\boldsymbol{\eta}(\boldsymbol{x}) - \widehat{\boldsymbol{\eta}}(\boldsymbol{x})\|_1] \tag{61}$$

To bound the other two terms, we can use Hölder's inequality:

$$
\begin{aligned}
\mathfrak{L}_{\mathbb{P}}(\boldsymbol{h}, \boldsymbol{g}^{\star}) - \mathfrak{L}_{\mathcal{S}}(\boldsymbol{h}, \boldsymbol{g}^{\star}) &= \mathbf{C}(\boldsymbol{g}^{\star}, \mathbb{P}) \cdot \nabla \Psi(\mathbf{C}(\boldsymbol{h}, \mathbb{P})) - \mathbf{C}(\boldsymbol{g}^{\star}, \mathbb{P}) \cdot \nabla \Psi(\widehat{\mathbf{C}}(\boldsymbol{h}, \mathcal{S})) \\
&= \mathbf{C}(\boldsymbol{g}^{\star}, \mathbb{P}) \cdot (\nabla \Psi(\mathbf{C}(\boldsymbol{h}, \mathbb{P})) - \nabla \Psi(\widehat{\mathbf{C}}(\boldsymbol{h}, \mathcal{S}))) \\
&\leq \|\nabla \Psi(\mathbf{C}(\boldsymbol{h}, \mathbb{P})) - \nabla \Psi(\widehat{\mathbf{C}}(\boldsymbol{h}, \mathcal{S}))\|_{\infty} \cdot \|\mathbf{C}(\boldsymbol{g}^{\star}, \mathbb{P})\|_1 \quad \text{(Hölder)} \\
&= m\|\nabla \Psi(\mathbf{C}(\boldsymbol{h}, \mathbb{P})) - \nabla \Psi(\widehat{\mathbf{C}}(\boldsymbol{h}, \mathcal{S}))\|_{\infty} \quad \text{(Normalization of } \mathbf{C}) \\
&\leq m\beta\|\mathbf{C}(\boldsymbol{h}, \mathbb{P}) - \widehat{\mathbf{C}}(\boldsymbol{h}, \mathcal{S})\|_1 \quad (\beta\text{-smoothness}) \\
&\leq 4m^2\beta\|\mathbf{C}(\boldsymbol{h}, \mathbb{P}) - \widehat{\mathbf{C}}(\boldsymbol{h}, \mathcal{S})\|_{\infty} \\
&\leq 4m^2\beta \sup_{\boldsymbol{h}' \in \mathcal{H}} \|\mathbf{C}(\boldsymbol{h}', \mathbb{P}) - \widehat{\mathbf{C}}(\boldsymbol{h}', \mathcal{S})]\|_{\infty} \quad (62)
\end{aligned}
$$

The same argument can be employed to bound the third term. Thus, applying Lemma D.2, we get with probability at least $1 - \delta$

$$
\mathfrak{L}_{\mathbb{P}}(\boldsymbol{h}, \boldsymbol{g}^{\star}) - \mathfrak{L}_{\mathbb{P}}(\boldsymbol{h}, \widehat{\boldsymbol{g}}) \leq 8\ell m^{-1} \, \mathbb{E}_{\boldsymbol{x}}[\|\boldsymbol{\eta}(\boldsymbol{x}) - \widehat{\boldsymbol{\eta}}(\boldsymbol{x})\|_1] + 
$$
$$
8m^2\beta \sup_{\boldsymbol{h}' \in \mathcal{H}} \tilde{\mathcal{O}} \left( \sqrt{\frac{m \cdot \log m \cdot \log n - \log \delta}{n}} \right) . \quad (63)
$$

$\square$

## D.4 CONSISTENCY OF FIXED-STEP-SCHEDULE FRANK-WOLFE

**Theorem 5.1** (Consistency of Frank-Wolfe). *Assume the utility function $\Psi \colon [0, 1]^{m \times 2 \times 2} \longrightarrow \mathbb{R}_{\geq 0}$ is concave over $\mathcal{C}_{\mathbb{P}}$, L-Lipschitz, and $\beta$-smooth w.r.t. the 1-norm. Let $\mathcal{S} = (\mathcal{S}_1, \mathcal{S}_2)$ be a sample drawn i.i.d. from $\mathbb{P}$. Further, let $\widehat{\eta}$ be a label probability estimator learned from $\mathcal{S}_1$, and $\boldsymbol{h}_{\mathcal{S}}^{\mathrm{FW}}$ be the classifier obtained after $\kappa n$ iterations. Then, for any $\delta \in (0, 1]$, with probability of at least $1 - \delta$ over draws of $\mathcal{S}$,*

$$
\Delta \Psi\big(\boldsymbol{h}_{\mathcal{S}}^{\mathrm{FW}}\big) \leq \mathcal{O}(\mathbb{E}_{\boldsymbol{x}}[\|\boldsymbol{\eta}(\boldsymbol{x}) - \widehat{\boldsymbol{\eta}}(\boldsymbol{x})\|_1]) + \tilde{\mathcal{O}} \left( m^2 \sqrt{\frac{m \cdot \log m \cdot \log n - \log \delta}{n}} \right) + \frac{8\beta m}{\kappa n + 2} . \quad (9)
$$

*Proof.* Define a curvature constant for the loss $\Psi$ as

$$
\begin{aligned}
C_{\Psi} &:= \sup_{\mathbf{C}^1, \mathbf{C}^2 \in \mathcal{C}_{\mathbb{P}}, \gamma \in [0,1]} \frac{2}{\gamma^2} \left( \Psi\big(\mathbf{C}^1 + \gamma(\mathbf{C}^2 - \mathbf{C}^1)\big) - \Psi\big(\mathbf{C}^1\big) - \gamma \left(\mathbf{C}^2 - \mathbf{C}^1\right) \cdot \nabla \Psi(\mathbf{C}^1)\right) \\
&\leq \sup_{\mathbf{C}^1, \mathbf{C}^2 \in \mathcal{C}_{\mathbb{P}}, \gamma \in [0,1]} \frac{2}{\gamma^2} \left( \frac{\beta}{2} \gamma^2 \|\mathbf{C}^2 - \mathbf{C}^1\|_1^2 \right) = \beta \sup_{\mathbf{C}^1, \mathbf{C}^2 \in \mathcal{C}_{\mathbb{P}}} \|\mathbf{C}^2 - \mathbf{C}^1\|_1^2 \leq 4\beta m , \quad (64)
\end{aligned}
$$

and let $\epsilon_{\mathcal{S}}$ be defined as in Lemma D.3. Set $\delta_{\mathrm{apx}} = (t + 1)\epsilon_{\mathcal{S}}/C_{\Psi}$ and $\hat{h}^i$ as in Algorithm 1. Let $\hat{f}^i$ be the classifier implicitly defined in iteration $i$, that is,

$$
\hat{f}^i := \sum_{j=1}^{i} \alpha^j \hat{h}^j . \quad (65)
$$

For $1 \leq i \leq t$, we can apply Lemma D.3 to $\hat{f}^{i-1}$ and $\hat{h}^i$, which gives

$$
\begin{aligned}
\mathbf{C}(\hat{h}^i, \mathbb{P}) \cdot \nabla \psi\Big(\mathbf{C}(\hat{f}^{i-1}, \mathbb{P})\Big) &\geq \max_{\boldsymbol{g}'} \mathbf{C}(\boldsymbol{g}', \mathbb{P}) \cdot \nabla \psi\Big(\mathbf{C}(\hat{f}^{i-1}, \mathbb{P})\Big) - \epsilon_{\mathcal{S}} \\
&= \max_{\mathbf{C} \in \mathcal{C}_{\mathbb{P}}} \mathbf{C} \cdot \nabla \psi\Big(\mathbf{C}(\hat{f}^{i-1}, \mathbb{P})\Big) - \epsilon_{\mathcal{S}} = \max_{\mathbf{C} \in \overline{\mathcal{C}_{\mathbb{P}}}} \mathbf{C} \cdot \nabla \psi\Big(\mathbf{C}(\hat{f}^{i-1}, \mathbb{P})\Big) - \epsilon_{\mathcal{S}} \\
&= \max_{\mathbf{C} \in \overline{\mathcal{C}_{\mathbb{P}}}} \mathbf{C} \cdot \nabla \psi\Big(\mathbf{C}(\hat{f}^{i-1}, \mathbb{P})\Big) - \frac{1}{2}\delta_{\mathrm{apx}} \frac{2}{t + 1} C_{\Psi} \\
&\geq \max_{\mathbf{C} \in \overline{\mathcal{C}_{\mathbb{P}}}} \mathbf{C} \cdot \nabla \psi\Big(\mathbf{C}(\hat{f}^{i-1}, \mathbb{P})\Big) - \frac{1}{2}\delta_{\mathrm{apx}} \frac{2}{i + 1} C_{\Psi} . \quad (66)
\end{aligned}
$$

As we consider, for the proof, a Frank-Wolfe implementation with fixed step schedule $\frac{2}{i+1}$, the confusion tensors are related through

$$\mathbf{C}(\hat{f}^i, \mathbb{P}) = \left(1 - \frac{2}{i+1}\right) \mathbf{C}(\hat{f}^{i-1}, \mathbb{P}) + \frac{2}{i+1} \mathbf{C}(\hat{h}^i, \mathbb{P}). \tag{67}$$

With results (66) and (67), we now have the exact same situation as in Narasimhan et al. (2015, Proof of Theorem 16). In particular, an application of Jaggi (2013, Theorem 1) gives the desired result. □

## E    LABEL DEPENDENCE AND OPTIMIZATION OF MACRO-AT-$k$ METRICS

The "budgeted-at-$k$" constraint couples the label-wise binary problems, resulting in their inability to be independently optimized. To demonstrate this coupling effect, we present a simple example. We consider the macro Jaccard similarity, defined below, and assume budget $k = 2$:

$$\Psi_{\text{Jaccard}}(\mathbf{C}(\boldsymbol{h})) = m^{-1} \sum_{j=1}^{m} \frac{C_{11}^j}{C_{11}^j + C_{01}^j + C_{10}^j}. \tag{68}$$

Let us consider two simple distributions, both with two different instances $\boldsymbol{x}$ of equal probability and three labels:

<table>
<tr><td colspan="5" align="center">Distribution $A$:</td></tr>
<tr><td></td><td>$P(\boldsymbol{x})$</td><td>$\eta_1(\boldsymbol{x})$</td><td>$\eta_2(\boldsymbol{x})$</td><td>$\eta_3(\boldsymbol{x})$</td></tr>
<tr><td>$\boldsymbol{x}_1$</td><td>0.5</td><td>0.4</td><td>0.2</td><td>0.6</td></tr>
<tr><td>$\boldsymbol{x}_2$</td><td>0.5</td><td>0.8</td><td>0.4</td><td>**0.4**</td></tr>
</table>

<table>
<tr><td colspan="5" align="center">Distribution $B$:</td></tr>
<tr><td></td><td>$P(\boldsymbol{x})$</td><td>$\eta_1(\boldsymbol{x})$</td><td>$\eta_2(\boldsymbol{x})$</td><td>$\eta_3(\boldsymbol{x})$</td></tr>
<tr><td>$\boldsymbol{x}_1$</td><td>0.5</td><td>0.4</td><td>0.2</td><td>0.6</td></tr>
<tr><td>$\boldsymbol{x}_2$</td><td>0.5</td><td>0.8</td><td>0.4</td><td>**0.8**</td></tr>
</table>

Notice that both distributions only differ on the marginal conditional probability of the third label of the second instance $\boldsymbol{x}_2$ ($\eta_3(\boldsymbol{x}_2)$). We find the optimal randomized classifiers for both distributions:

<table>
<tr><td colspan="4" align="left">Optimal $\boldsymbol{h}_A^\star(\boldsymbol{x})$ for distribution $A$:</td></tr>
<tr><td></td><td>$\pi_1(\boldsymbol{x})$</td><td>$\pi_2(\boldsymbol{x})$</td><td>$\pi_3(\boldsymbol{x})$</td></tr>
<tr><td>$\boldsymbol{x}_1$</td><td>**1.0**</td><td>**0.0**</td><td>1.0</td></tr>
<tr><td>$\boldsymbol{x}_2$</td><td>1.0</td><td>1.0</td><td>0.0</td></tr>
<tr><td colspan="4">$\Psi_{\text{Jaccard}}(\mathbf{C}(\boldsymbol{h}_A^\star, A)) \approx 0.453962$</td></tr>
</table>

<table>
<tr><td colspan="4" align="left">Optimal $\boldsymbol{h}_B^\star(\boldsymbol{x})$ for distribution B:</td></tr>
<tr><td></td><td>$\pi_1(\boldsymbol{x})$</td><td>$\pi_2(\boldsymbol{x})$</td><td>$\pi_3(\boldsymbol{x})$</td></tr>
<tr><td>$\boldsymbol{x}_1$</td><td>**0.0**</td><td>**1.0**</td><td>1.0</td></tr>
<tr><td>$\boldsymbol{x}_2$</td><td>1.0</td><td>0.0</td><td>1.0</td></tr>
<tr><td colspan="4">$\Psi_{\text{Jaccard}}(\mathbf{C}(\boldsymbol{h}_B^\star, B)) \approx 0.471423$</td></tr>
</table>

We can notice that despite changing only one marginal conditional probability, the optimal solution is different on the other instance for the two other labels. If it were possible to find the solution for each label separately, the change in the distribution on one label would not affect the order of other labels, as it happened in the above example.

## F    EXPERIMENTAL SETUP

### F.1    TRAINING AND SELECTION OF MARGINAL PROBABILITY ESTIMATORS

In our experiments, we use two types of models for the estimation of marginal conditional probabilities of labels $\boldsymbol{\eta}(\boldsymbol{x})$:

1. For MEDIAMILL, FLICKR, and RCV1X datasets, we use multi-layer fully connected neural network (ranging from 1 to 3 layers with hidden layer size from (128 to 2048) implemented in Pytorch Paszke et al. (2019). We perform a search for the best hyper-parameters (number and size of layers, learning rate, number of epochs) using a validation set created from the train set for each loss used (binary cross-entropy, focal and asymmetric loss). Then, the model is retrained on the whole training set. We use Adam optimizer (Kingma & Ba, 2015). For Focal and Asymmetric loss, we use default parameters suggested by the authors in (Ridnik et al., 2021).

Table 3: Mean results with standard deviation of different inference strategies on measure calculated at $\{3, 5, 10\}$ Notation: P—precision, R—recall, F1—F1-measure. The green color indicates cells in which the strategy matches the metric. The best results are in **bold** and the second best are in *italic*.

| Inference strategy | Instance @3 P±std | R±std | Macro @3 P±std | R±std | F1±std | Instance @5 P±std | R±std | Macro @5 P±std | R±std | F1±std | Instance @10 P±std | R±std | Macro @10 P±std | R±std | F1±std |
|---|---|---|---|---|---|---|---|---|---|---|---|---|---|---|---|
| **MEDIAMILL** | | | | | | | | | | | | | | | |
| TOP-K | **66.25** ±0.00 | 49.55 ±0.00 | 8.96 ±0.00 | 4.81 ±0.00 | 4.95 ±0.00 | *51.96* ±0.00 | 62.04 ±0.00 | 12.85 ±0.00 | 8.75 ±0.00 | 7.71 ±0.00 | **33.63** ±0.00 | 76.60 ±0.00 | 11.46 ±0.00 | 19.68 ±0.00 | 11.28 ±0.00 |
| TOP-K+$w$^POW | 57.36 ±0.00 | 42.51 ±0.00 | 15.31 ±0.00 | 11.84 ±0.00 | *10.54* ±0.00 | 47.68 ±0.00 | 56.62 ±0.00 | 13.00 ±0.00 | 17.37 ±0.00 | *12.64* ±0.00 | 32.18 ±0.00 | 72.98 ±0.00 | 9.64 ±0.00 | 29.43 ±0.00 | *13.07* ±0.00 |
| TOP-K+$w$^LOG | 39.72 ±0.00 | 27.32 ±0.00 | 14.43 ±0.00 | 10.10 ±0.00 | 9.41 ±0.00 | 35.40 ±0.00 | 39.96 ±0.00 | 11.38 ±0.00 | 15.33 ±0.00 | 10.95 ±0.00 | 28.45 ±0.00 | 63.36 ±0.00 | 9.86 ±0.00 | 26.25 ±0.00 | 12.26 ±0.00 |
| TOP-K+$\ell$_FOCAL | 65.87 ±0.00 | **49.60** ±0.00 | 10.08 ±0.00 | 4.87 ±0.00 | 4.94 ±0.00 | **52.08** ±0.00 | **62.16** ±0.00 | 11.99 ±0.00 | 8.93 ±0.00 | 7.90 ±0.00 | *33.61* ±0.00 | 76.65 ±0.00 | 10.76 ±0.00 | 20.08 ±0.00 | 11.37 ±0.00 |
| TOP-K+$\ell$_ASYM | *65.88* ±0.00 | 49.48 ±0.00 | 10.31 ±0.00 | 4.58 ±0.00 | 4.80 ±0.00 | 51.55 ±0.00 | 61.87 ±0.00 | 11.10 ±0.00 | 8.50 ±0.00 | 7.48 ±0.00 | 33.54 ±0.00 | **76.75** ±0.00 | 10.73 ±0.00 | 19.55 ±0.00 | 11.16 ±0.00 |
| MACRO-P_FW | 7.94 ±0.09 | 6.13 ±0.08 | **19.33** ±0.92 | 6.06 ±0.32 | 2.87 ±0.13 | 6.99 ±0.07 | 8.96 ±0.09 | **17.29** ±1.22 | 8.79 ±0.20 | 3.17 ±0.11 | 6.02 ±0.03 | 14.14 ±0.10 | **17.38** ±1.60 | 17.24 ±0.49 | 5.23 ±0.09 |
| MACRO-R_PRIOR | 6.37 ±0.00 | 3.67 ±0.00 | 8.81 ±0.00 | **19.82** ±0.00 | 5.31 ±0.00 | 7.38 ±0.00 | 7.25 ±0.00 | 8.91 ±0.00 | **26.50** ±0.00 | 6.71 ±0.00 | 8.31 ±0.00 | 17.42 ±0.00 | 10.53 ±0.00 | **39.24** ±0.00 | 8.85 ±0.00 |
| MACRO-R_FW | 6.37 ±0.00 | 3.67 ±0.00 | 8.81 ±0.00 | **19.82** ±0.00 | 5.31 ±0.00 | 7.38 ±0.00 | 7.25 ±0.00 | 8.91 ±0.00 | **26.50** ±0.00 | 6.71 ±0.00 | 8.31 ±0.00 | 17.42 ±0.00 | 10.53 ±0.00 | **39.24** ±0.00 | 8.85 ±0.00 |
| MACRO-F1_FW | 45.20 ±0.12 | 33.05 ±0.11 | *15.42* ±0.24 | 11.17 ±0.10 | **12.21** ±0.10 | 43.57 ±0.03 | 51.60 ±0.05 | *15.20* ±0.47 | 15.05 ±0.11 | **13.82** ±0.14 | 28.12 ±0.02 | 64.23 ±0.04 | *13.93* ±0.16 | 23.32 ±0.51 | **14.81** ±0.09 |
| **FLICKR** | | | | | | | | | | | | | | | |
| TOP-K | **23.94** ±0.00 | **56.96** ±0.00 | 23.04 ±0.00 | 38.41 ±0.00 | 26.56 ±0.00 | **16.99** ±0.00 | **66.01** ±0.00 | 17.12 ±0.00 | 47.03 ±0.00 | 23.49 ±0.00 | **10.16** ±0.00 | **77.35** ±0.00 | 10.72 ±0.00 | 59.37 ±0.00 | 17.24 ±0.00 |
| TOP-K+$w$^POW | 22.35 ±0.00 | 53.44 ±0.00 | 17.96 ±0.00 | 44.26 ±0.00 | 24.21 ±0.00 | 16.10 ±0.00 | 62.80 ±0.00 | 13.76 ±0.00 | 52.39 ±0.00 | 20.68 ±0.00 | 9.77 ±0.00 | 74.54 ±0.00 | 9.08 ±0.00 | 63.98 ±0.00 | 15.08 ±0.00 |
| TOP-K+$w$^LOG | 23.57 ±0.00 | 56.17 ±0.00 | 19.86 ±0.00 | 41.36 ±0.00 | 25.49 ±0.00 | 16.76 ±0.00 | 65.21 ±0.00 | 15.05 ±0.00 | 49.75 ±0.00 | 22.00 ±0.00 | *10.06* ±0.00 | 76.63 ±0.00 | 9.79 ±0.00 | 61.80 ±0.00 | 16.10 ±0.00 |
| TOP-K+$\ell$_FOCAL | *23.64* ±0.00 | 56.27 ±0.00 | 24.90 ±0.00 | 36.67 ±0.00 | 26.42 ±0.00 | *16.89* ±0.00 | 65.62 ±0.00 | 18.53 ±0.00 | 45.67 ±0.00 | *24.16* ±0.00 | 10.05 ±0.00 | 76.63 ±0.00 | 11.77 ±0.00 | 57.90 ±0.00 | *18.14* ±0.00 |
| TOP-K+$\ell$_ASYM | 23.37 ±0.00 | 55.65 ±0.00 | 23.09 ±0.00 | 37.00 ±0.00 | 26.12 ±0.00 | 16.74 ±0.00 | 65.04 ±0.00 | 17.39 ±0.00 | 45.61 ±0.00 | 23.60 ±0.00 | 10.06 ±0.00 | *76.63* ±0.00 | 10.91 ±0.00 | 58.36 ±0.00 | 17.48 ±0.00 |
| MACRO-P_FW | 4.65 ±0.03 | 11.49 ±0.09 | **39.34** ±1.25 | 6.63 ±0.05 | 8.06 ±0.12 | 5.66 ±0.02 | 22.75 ±0.07 | **41.74** ±1.48 | 9.70 ±0.11 | 10.57 ±0.13 | 2.83 ±0.01 | 22.26 ±0.06 | **37.59** ±1.21 | 10.68 ±0.06 | 8.50 ±0.09 |
| MACRO-R_PRIOR | 16.14 ±0.00 | 38.62 ±0.00 | 17.58 ±0.00 | **45.50** ±0.00 | 22.27 ±0.00 | 12.17 ±0.00 | 47.48 ±0.00 | 13.98 ±0.00 | **53.83** ±0.00 | 19.72 ±0.00 | 7.89 ±0.00 | 60.42 ±0.00 | 9.57 ±0.00 | **64.66** ±0.00 | 15.07 ±0.00 |
| MACRO-R_FW | 16.14 ±0.00 | 38.62 ±0.00 | 17.58 ±0.00 | **45.50** ±0.00 | 22.27 ±0.00 | 12.17 ±0.00 | 47.48 ±0.00 | 13.98 ±0.00 | **53.83** ±0.00 | 19.72 ±0.00 | 7.89 ±0.00 | 60.42 ±0.00 | 9.57 ±0.00 | **64.66** ±0.00 | 15.07 ±0.00 |
| MACRO-F1_FW | 17.59 ±0.00 | 41.60 ±0.00 | *35.28* ±0.00 | 29.28 ±0.00 | **29.43** ±0.00 | 12.22 ±0.01 | 47.31 ±0.07 | *34.13* ±0.15 | 32.70 ±0.05 | **29.43** ±0.04 | 5.92 ±0.00 | 45.77 ±0.00 | *34.55* ±0.00 | 33.08 ±0.00 | **29.02** ±0.00 |
| **RCV1X** | | | | | | | | | | | | | | | |
| TOP-K | **72.99** ±0.00 | **75.32** ±0.00 | 13.06 ±0.00 | 4.67 ±0.00 | 5.43 ±0.00 | **52.30** ±0.00 | **81.96** ±0.00 | 12.77 ±0.00 | 7.61 ±0.00 | 7.64 ±0.00 | **32.98** ±0.00 | **89.70** ±0.00 | 11.35 ±0.00 | 14.75 ±0.00 | 10.28 ±0.00 |
| TOP-K+$w$^POW | 65.99 ±0.00 | 69.11 ±0.00 | 18.58 ±0.00 | 12.78 ±0.00 | **13.09** ±0.00 | 48.48 ±0.00 | 77.18 ±0.00 | 14.69 ±0.00 | 17.66 ±0.00 | 17.66 ±0.00 | 31.43 ±0.00 | 87.14 ±0.00 | 10.63 ±0.00 | 26.05 ±0.00 | *12.82* ±0.00 |
| TOP-K+$w$^LOG | 70.70 ±0.00 | 73.37 ±0.00 | 19.97 ±0.00 | 8.10 ±0.00 | 9.80 ±0.00 | 51.18 ±0.00 | 80.49 ±0.00 | 16.03 ±0.00 | 11.75 ±0.00 | 11.29 ±0.00 | 32.66 ±0.00 | 89.14 ±0.00 | 11.96 ±0.00 | 19.01 ±0.00 | 12.06 ±0.00 |
| TOP-K+$\ell$_FOCAL | *71.99* ±0.00 | 74.38 ±0.00 | 14.06 ±0.00 | 4.83 ±0.00 | 5.76 ±0.00 | *51.46* ±0.00 | *80.94* ±0.00 | 12.49 ±0.00 | 7.65 ±0.00 | 7.75 ±0.00 | 32.38 ±0.00 | 88.75 ±0.00 | 10.59 ±0.00 | 14.42 ±0.00 | 10.06 ±0.00 |
| TOP-K+$\ell$_ASYM | 71.14 ±0.00 | 73.60 ±0.00 | 14.40 ±0.00 | 5.44 ±0.00 | 6.46 ±0.00 | 50.81 ±0.00 | 80.13 ±0.00 | 12.27 ±0.00 | 8.52 ±0.00 | 8.41 ±0.00 | 31.88 ±0.00 | 87.85 ±0.00 | 9.64 ±0.00 | 15.16 ±0.00 | 10.03 ±0.00 |
| MACRO-P_FW | 46.36 ±0.03 | 50.11 ±0.02 | *21.11* ±0.32 | 5.61 ±0.07 | 5.84 ±0.07 | 29.40 ±0.02 | 49.81 ±0.03 | *21.69* ±0.29 | 5.72 ±0.05 | 5.31 ±0.05 | 19.45 ±0.01 | 60.40 ±0.02 | *21.66* ±0.21 | 6.03 ±0.06 | 5.78 ±0.05 |
| MACRO-R_PRIOR | 44.26 ±0.00 | 46.10 ±0.00 | 14.60 ±0.00 | *18.24* ±0.00 | 12.04 ±0.00 | 34.77 ±0.00 | 56.28 ±0.00 | 13.13 ±0.00 | *24.59* ±0.00 | 12.77 ±0.00 | 24.08 ±0.00 | 70.51 ±0.00 | 10.66 ±0.00 | *34.34* ±0.00 | 12.39 ±0.00 |
| MACRO-R_FW | 43.28 ±0.00 | 44.99 ±0.00 | 14.56 ±0.00 | **18.41** ±0.00 | 11.95 ±0.00 | 34.15 ±0.00 | 55.24 ±0.00 | 13.15 ±0.00 | **24.89** ±0.00 | 12.73 ±0.00 | 23.78 ±0.00 | 69.71 ±0.00 | 10.76 ±0.00 | **34.66** ±0.00 | 12.44 ±0.00 |
| MACRO-F1_FW | 58.20 ±0.03 | 61.22 ±0.03 | **21.45** ±0.14 | 10.37 ±0.09 | *12.09* ±0.05 | 44.42 ±0.01 | 71.86 ±0.01 | **21.96** ±0.11 | 12.25 ±0.06 | **13.68** ±0.03 | 27.26 ±0.00 | 78.88 ±0.00 | **22.10** ±0.03 | 14.86 ±0.02 | **15.12** ±0.01 |
| **AMAZONCAT** | | | | | | | | | | | | | | | |
| TOP-K | **78.29** ±0.00 | 59.29 ±0.00 | 35.73 ±0.00 | 12.44 ±0.00 | 16.52 ±0.00 | **63.63** ±0.00 | **74.54** ±0.00 | 46.43 ±0.00 | 32.72 ±0.00 | 35.06 ±0.00 | **39.16** ±0.00 | 85.18 ±0.00 | 39.52 ±0.00 | 51.69 ±0.00 | 40.39 ±0.00 |
| TOP-K+$w$^POW | 66.32 ±0.00 | 49.76 ±0.00 | 50.21 ±0.00 | 45.79 ±0.00 | **45.70** ±0.00 | 57.12 ±0.00 | 69.47 ±0.00 | 44.85 ±0.00 | 53.78 ±0.00 | **46.30** ±0.00 | 37.31 ±0.00 | 82.20 ±0.00 | 30.13 ±0.00 | 63.53 ±0.00 | 37.15 ±0.00 |
| TOP-K+$w$^LOG | 72.56 ±0.00 | 54.56 ±0.00 | 50.30 ±0.00 | 32.06 ±0.00 | 36.94 ±0.00 | *61.15* ±0.00 | 71.83 ±0.00 | 48.93 ±0.00 | 42.87 ±0.00 | *43.05* ±0.00 | *38.71* ±0.00 | 84.49 ±0.00 | 36.84 ±0.00 | 56.71 ±0.00 | *40.60* ±0.00 |
| MACRO-P_FW | 47.00 ±0.02 | 35.57 ±0.02 | *56.47* ±0.12 | 23.74 ±0.06 | 29.62 ±0.07 | 41.04 ±0.01 | 50.74 ±0.01 | *55.85* ±0.08 | 27.45 ±0.07 | 30.23 ±0.07 | 30.66 ±0.01 | 69.67 ±0.02 | **55.27** ±0.11 | 29.09 ±0.06 | 34.51 ±0.05 |
| MACRO-R_PRIOR | 48.58 ±0.00 | 34.93 ±0.00 | 37.16 ±0.00 | **59.97** ±0.00 | 42.02 ±0.00 | 40.67 ±0.00 | 47.35 ±0.00 | 28.17 ±0.00 | 66.98 ±0.00 | 35.75 ±0.00 | 28.06 ±0.00 | 62.91 ±0.00 | 17.62 ±0.00 | **73.98** ±0.00 | 25.04 ±0.00 |
| MACRO-R_FW | 48.58 ±0.00 | 34.93 ±0.00 | 37.15 ±0.00 | *59.97* ±0.00 | 42.02 ±0.00 | 40.67 ±0.00 | 47.35 ±0.00 | 28.17 ±0.00 | **66.98** ±0.00 | 35.75 ±0.00 | 28.06 ±0.00 | 62.91 ±0.00 | 17.62 ±0.00 | **73.98** ±0.00 | 25.04 ±0.00 |
| MACRO-F1_FW | 68.59 ±0.01 | 51.49 ±0.00 | **56.75** ±0.04 | 34.68 ±0.03 | **40.90** ±0.02 | 55.73 ±0.00 | 65.60 ±0.00 | **56.62** ±0.01 | 36.40 ±0.01 | **41.92** ±0.01 | 35.30 ±0.00 | 78.34 ±0.00 | *54.67* ±0.01 | 39.93 ±0.01 | **43.26** ±0.01 |

2. For AMAZONCAT dataset, we use probabilistic label tree (PLT) with LIBLINEAR models trained with $L_2$-regularized logistic loss (Fan et al., 2008). We make it sparse by truncating all the weights whose absolute value is above the threshold of 0.01, as introduced in (Babbar & Schölkopf, 2017), to reduce the model size and inference time. Use the implementation provided in NAPKINXC library (Jasinska-Kobus et al., 2020) and use the library's default parameters.

## F.2 SPARSE MARGINALS IN FRANK-WOLFE ALGORITHM

Materializing the $\widehat{\boldsymbol{\eta}}(\boldsymbol{x})$ for all instances in the form of a dense matrix requires a considerable amount of memory for datasets like AMAZONCAT (over 58 Gb using 32-bit floats). Because of that, we instead use a sparse matrix in *compressed sparse row* (CSR) format with only top-$k'$ values of marginals kept for each instance, where $k \ll k' \ll m$. All the other marginals are being treated as zeros. In CSR format, the row vectors are represented as a list of tuples $\boldsymbol{a}_i^{\text{csr}} := \{(\text{index}, \text{value}) : \text{value} \neq 0\}$ and allow for efficient element-wise multiplication between both dense and sparse vectors needed in Frank-Wolfe procedure. By using the sparse matrix of marginals with exactly $k'$ non-zero values, we effectively reduce the complexity of one iteration from $\mathcal{O}(nk')$ instead of $\mathcal{O}(nm)$. We use $k' = 200$ for both RCV1X and AMAZONCAT datasets. We found that using $k' > 100$ has no negative impact on predictive performance compared to using a full dense matrix.

## F.3 HARDWARE

All the experiments were conducted on a workstation with 64 GB of RAM and Nvidia V100 16Gb GPU. However, the experiments can be also reproduced with smaller amount of memory.

## G EXTENDED RESULTS

In this section, we include extended results of our empirical experiments. Here, we include tables with standard deviations. In Table 3, we present the results of the main experiment from Section 6 with standard deviations. In Table 4, we present the results on balanced accuracy, for which the

Table 4: Comparison of two classifiers for macro-balanced accuracy calculated at $\{3, 5, 10\}$ – a closed form classifier (MACRO-BA$_{\text{PRIOR}}$) and 2) classifier found using Frank-Wolfe algorithm (MACRO-BA$_{\text{FW}}$). The  green color  indicates cells in which the strategy matches the metric.

| Inference strategy | Instance @3 | | Macro @3 | | | BA | Instance @5 | | Macro @5 | | | BA | Instance @10 | | Macro @10 | | | BA |
|---|---|---|---|---|---|---|---|---|---|---|---|---|---|---|---|---|---|---|
| | P | R | P | R | F1 | | P | R | P | R | F1 | | P | R | P | R | F1 | |
| **MEDIAMILL** | | | | | | | | | | | | | | | | | | |
| MACRO-BA$_{\text{PRIOR}}$ | 7.43 | 4.41 | 10.70 | 19.86 | 5.65 | 58.54 | 9.05 | 9.34 | 11.30 | 26.54 | 7.25 | 60.98 | 11.53 | 26.05 | 10.65 | 39.33 | 9.88 | 65.15 |
| MACRO-BA$_{\text{FW}}$ | 7.43 | 4.41 | 10.70 | 19.86 | 5.65 | 58.54 | 9.05 | 9.34 | 11.30 | 26.54 | 7.25 | 60.98 | 11.53 | 26.05 | 10.65 | 39.33 | 9.88 | 65.15 |
| **FLICKR** | | | | | | | | | | | | | | | | | | |
| MACRO-BA$_{\text{PRIOR}}$ | 16.33 | 39.10 | 17.56 | 45.50 | 22.31 | 72.10 | 12.35 | 48.20 | 13.96 | 53.84 | 19.77 | 75.80 | 7.98 | 61.19 | 9.57 | 64.67 | 15.09 | 79.97 |
| MACRO-BA$_{\text{FW}}$ | 16.33 | 39.10 | 17.56 | 45.50 | 22.31 | 72.10 | 12.35 | 48.20 | 13.96 | 53.84 | 19.77 | 75.80 | 7.98 | 61.19 | 9.57 | 64.67 | 15.09 | 79.97 |
| **RCV1X** | | | | | | | | | | | | | | | | | | |
| MACRO-BA$_{\text{PRIOR}}$ | 44.15 | 46.00 | 14.62 | 18.40 | 12.01 | 59.17 | 34.71 | 56.27 | 13.18 | 24.86 | 12.78 | 62.37 | 24.07 | 70.61 | 10.77 | 34.64 | 12.47 | 67.17 |
| MACRO-BA$_{\text{FW}}$ | 44.15 | 46.00 | 14.62 | 18.40 | 12.01 | 59.17 | 34.71 | 56.27 | 13.18 | 24.86 | 12.78 | 62.37 | 24.07 | 70.61 | 10.77 | 34.64 | 12.47 | 67.17 |
| **AMAZONCAT** | | | | | | | | | | | | | | | | | | |
| MACRO-BA$_{\text{PRIOR}}$ | 47.02 | 33.74 | 35.27 | 61.70 | 40.42 | 80.84 | 39.28 | 45.74 | 26.77 | 67.95 | 34.15 | 83.97 | 27.45 | 61.62 | 17.36 | 73.37 | 24.55 | 86.66 |
| MACRO-BA$_{\text{FW}}$ | 47.02 | 33.74 | 35.27 | 61.70 | 40.42 | 80.84 | 39.28 | 45.74 | 26.77 | 67.95 | 34.15 | 83.97 | 27.45 | 61.62 | 17.36 | 73.37 | 24.55 | 86.66 |

Table 5: Means with standard deviations of running times and numbers of iterations performed by the Frank-Wolfe algorithm for different objective measures calculated at $3, 5, 10$. Notation: T—total time in seconds, I—number of iterations.

| Inference strategy | $T$@3 ±std | $I$@3 ±std | $T$@5 ±std | $I$@5 ±std | $T$@10 ±std | $I$@10 ±std |
|---|---|---|---|---|---|---|
| **MEDIAMILL** | | | | | | |
| MACRO-P$_{\text{FW}}$ | 1.21 ±0.19 | 9.90 ±1.87 | 1.15 ±0.21 | 8.70 ±1.27 | 1.57 ±0.39 | 10.70 ±2.69 |
| MACRO-R$_{\text{FW}}$ | 0.52 ±0.09 | 3.00 ±0.00 | 0.53 ±0.08 | 3.00 ±0.00 | 0.58 ±0.10 | 3.00 ±0.00 |
| MACRO-F1$_{\text{FW}}$ | 1.44 ±0.14 | 10.40 ±1.02 | 1.42 ±0.27 | 8.90 ±1.14 | 1.34 ±0.35 | 9.10 ±2.91 |
| **FLICKR** | | | | | | |
| MACRO-P$_{\text{FW}}$ | 2.37 ±0.65 | 8.30 ±1.35 | 2.40 ±0.52 | 10.20 ±1.47 | 2.44 ±0.42 | 9.40 ±1.02 |
| MACRO-R$_{\text{FW}}$ | 1.17 ±0.19 | 3.00 ±0.00 | 1.07 ±0.09 | 3.00 ±0.00 | 1.17 ±0.15 | 3.00 ±0.00 |
| MACRO-F1$_{\text{FW}}$ | 1.39 ±0.19 | 4.60 ±0.80 | 2.00 ±0.34 | 7.60 ±1.43 | 2.01 ±0.47 | 7.20 ±1.89 |
| **RCV1X** | | | | | | |
| MACRO-P$_{\text{FW}}$ | 25.33 ±3.13 | 7.40 ±0.92 | 23.08 ±1.80 | 6.70 ±0.46 | 20.04 ±2.38 | 5.20 ±0.40 |
| MACRO-R$_{\text{FW}}$ | 20.70 ±1.95 | 6.00 ±0.00 | 21.94 ±2.49 | 6.00 ±0.00 | 22.41 ±2.30 | 6.00 ±0.00 |
| MACRO-F1$_{\text{FW}}$ | 15.05 ±1.04 | 4.20 ±0.40 | 15.03 ±1.02 | 4.00 ±0.00 | 15.75 ±1.20 | 4.00 ±0.00 |
| **AMAZONCAT** | | | | | | |
| MACRO-P$_{\text{FW}}$ | 21.97 ±2.47 | 4.50 ±0.50 | 20.80 ±3.91 | 3.80 ±0.40 | 23.44 ±2.78 | 4.40 ±0.66 |
| MACRO-R$_{\text{FW}}$ | 30.89 ±3.86 | 6.00 ±0.00 | 29.70 ±3.33 | 6.00 ±0.00 | 32.25 ±4.10 | 6.00 ±0.00 |
| MACRO-F1$_{\text{FW}}$ | 15.76 ±0.53 | 3.00 ±0.00 | 18.18 ±3.44 | 3.40 ±0.92 | 34.02 ±6.84 | 6.60 ±1.20 |

introduced Frank-Wolfe algorithm retrieves the same solution as the closed form classifier described in Section 4, similarly to the case with macro-recall. In Table 5, we present the mean number of iterations and running time of the Frank-Wolfe algorithm for different objective measures, as well as different values of $k$. The values were calculated based on 10 runs of the algorithm. We use the same value of stopping condition $\epsilon = 0.001$ for all the experiments. The Frank-Wolfe algorithm requires only a small number of 3-10 iterations, and thanks to the usage of sparse matrices as described in Section F.2, it needs, in most cases, less than a minute even for larger benchmark datasets like AMAZONCAT. In further subsections, we also present the results for different splits and initialization strategies for the Frank-Wolfe algorithm.

## G.1 IMPACT OF SPLITTING STRATEGY IN THE FRANK-WOLFE ALGORITHM

In this experiment, we test different ratios of splitting training datasets into the sets used for training the $\boldsymbol{\eta}$ estimator and estimating confusion matrix $\boldsymbol{C}$ (50/50 or 75/25 split), as well as a variant where we use the whole training set for both training the estimator and estimating $\boldsymbol{C}$ (100/100 split). The initial classifier $\boldsymbol{h}^0$ is initialized with the top-k $\widehat{\eta}_j$ classifier for all the experiments here. We present the result of this comparison in Table 6. The results suggest that more data used for training is beneficial for the quality of the final randomized classifier.

Table 6: Comparison of different splitting strategies for the Frank-Wolfe algorithm on measures calculated at $\{3, 5, 10\}$. Notation: P—precision, R—recall, F1—F1-measure. The green color indicates cells in which the strategy matches the metric.

| Inference strategy | Instance @3 | | Macro @3 | | | Instance @5 | | Macro @5 | | | Instance @10 | | Macro @10 | | |
|---|---|---|---|---|---|---|---|---|---|---|---|---|---|---|---|
| | P | R | P | R | F1 | P | R | P | R | F1 | P | R | P | R | F1 |
| **MEDIAMILL** | | | | | | | | | | | | | | | |
| MACRO-P_FW + 50/50 split | 7.76 | 5.87 | 15.41 | 6.65 | 3.89 | 9.22 | 11.35 | 13.60 | 11.27 | 5.38 | 3.56 | 8.58 | 15.22 | 19.08 | 5.44 |
| MACRO-P_FW + 75/25 split | 6.74 | 5.03 | 14.68 | 6.48 | 3.58 | 9.00 | 10.70 | 13.79 | 13.07 | 5.21 | 4.54 | 10.56 | 14.47 | 19.30 | 6.09 |
| MACRO-P_FW + 100/100 split | 7.94 | 6.13 | 19.33 | 6.06 | 2.87 | 6.99 | 8.96 | 17.29 | 8.79 | 3.17 | 6.02 | 14.14 | 17.38 | 17.24 | 5.23 |
| MACRO-R_FW + 50/50 split | 5.14 | 3.05 | 8.42 | 15.56 | 4.58 | 6.04 | 6.06 | 9.55 | 22.20 | 5.93 | 6.75 | 14.26 | 9.97 | 33.82 | 7.52 |
| MACRO-R_FW + 75/25 split | 4.41 | 2.52 | 7.89 | 15.30 | 4.30 | 4.93 | 4.75 | 7.02 | 21.83 | 5.40 | 5.83 | 12.01 | 9.40 | 34.91 | 7.37 |
| MACRO-R_FW + 100/100 split | 6.37 | 3.67 | 8.81 | 19.82 | 5.31 | 7.38 | 7.25 | 8.91 | 26.50 | 6.71 | 8.31 | 17.42 | 10.53 | 39.24 | 8.85 |
| MACRO-F1_FW + 50/50 split | 45.25 | 33.24 | 14.19 | 10.67 | 10.93 | 41.36 | 49.55 | 12.97 | 15.24 | 12.59 | 26.90 | 62.50 | 12.19 | 23.43 | 13.50 |
| MACRO-F1_FW + 75/25 split | 43.07 | 31.33 | 13.80 | 10.65 | 10.91 | 40.06 | 47.64 | 12.61 | 15.06 | 12.49 | 27.84 | 64.08 | 11.86 | 23.77 | 13.64 |
| MACRO-F1_FW + 100/100 split | 46.77 | 34.21 | 15.61 | 11.16 | 12.35 | 43.48 | 51.65 | 14.93 | 14.98 | 13.69 | 27.92 | 64.11 | 12.14 | 28.42 | 14.63 |
| **FLICKR** | | | | | | | | | | | | | | | |
| MACRO-P_FW + 50/50 split | 5.66 | 13.32 | 37.62 | 12.69 | 12.84 | 4.32 | 17.11 | 37.80 | 15.02 | 13.99 | 2.26 | 17.43 | 36.93 | 16.40 | 12.71 |
| MACRO-P_FW + 75/25 split | 6.73 | 16.18 | 39.33 | 16.33 | 16.37 | 3.84 | 15.28 | 38.07 | 15.61 | 15.13 | 2.22 | 17.40 | 37.73 | 17.95 | 14.97 |
| MACRO-P_FW + 100/100 split | 4.65 | 11.49 | 39.34 | 6.63 | 8.06 | 5.66 | 22.75 | 41.74 | 9.70 | 10.57 | 2.83 | 22.26 | 37.59 | 10.68 | 8.50 |
| MACRO-R_FW + 50/50 split | 14.90 | 35.66 | 18.30 | 42.72 | 21.83 | 11.41 | 44.65 | 14.71 | 51.16 | 19.80 | 7.50 | 57.46 | 9.97 | 62.17 | 15.34 |
| MACRO-R_FW + 75/25 split | 15.34 | 36.81 | 17.65 | 43.89 | 21.84 | 11.62 | 45.44 | 13.78 | 52.03 | 19.10 | 7.58 | 58.07 | 9.33 | 63.70 | 14.50 |
| MACRO-R_FW + 100/100 split | 16.14 | 38.62 | 17.58 | 45.50 | 22.27 | 12.17 | 47.48 | 13.98 | 53.83 | 19.72 | 7.89 | 60.42 | 9.57 | 64.66 | 15.07 |
| MACRO-F1_FW + 50/50 split | 19.06 | 45.28 | 31.30 | 31.55 | 28.63 | 12.24 | 47.51 | 31.10 | 34.24 | 29.02 | 6.18 | 47.82 | 31.37 | 35.89 | 28.73 |
| MACRO-F1_FW + 75/25 split | 17.33 | 41.28 | 31.93 | 32.64 | 29.65 | 11.46 | 44.54 | 31.62 | 34.78 | 29.74 | 5.86 | 45.22 | 30.52 | 38.03 | 29.37 |
| MACRO-F1_FW + 100/100 split | 18.21 | 43.06 | 34.89 | 29.51 | 29.41 | 11.78 | 45.91 | 34.68 | 30.87 | 29.45 | 7.14 | 55.21 | 34.00 | 33.17 | 29.11 |

Table 7: Comparison of different initialization strategies for the Frank-Wolfe algorithm on measures calculated at $\{3, 5, 10\}$ Notation: P—precision, R—recall, F1—F1-measure. The green color indicates cells in which the strategy matches the metric.

| Inference strategy | Instance @3 | | Macro @3 | | | Instance @5 | | Macro @5 | | | Instance @10 | | Macro @10 | | |
|---|---|---|---|---|---|---|---|---|---|---|---|---|---|---|---|
| | P | R | P | R | F1 | P | R | P | R | F1 | P | R | P | R | F1 |
| **MEDIAMILL** | | | | | | | | | | | | | | | |
| MACRO-P_FW + top-k init. | 7.49 | 5.35 | 16.54 | 9.43 | 3.54 | 9.61 | 11.18 | 17.10 | 11.30 | 4.37 | 5.66 | 12.90 | 17.06 | 17.31 | 5.72 |
| MACRO-P_FW + rnd init. | 7.94 | 6.13 | 19.33 | 6.06 | 2.87 | 6.99 | 8.96 | 17.29 | 8.79 | 3.17 | 6.02 | 14.14 | 17.38 | 17.24 | 5.23 |
| MACRO-R_FW + top-k init. | 6.37 | 3.67 | 8.81 | 19.82 | 5.31 | 7.38 | 7.25 | 8.91 | 26.50 | 6.71 | 8.31 | 17.42 | 10.53 | 39.24 | 8.85 |
| MACRO-R_FW + rnd init. | 6.37 | 3.67 | 8.81 | 19.82 | 5.31 | 7.38 | 7.25 | 8.91 | 26.50 | 6.71 | 8.31 | 17.42 | 10.53 | 39.24 | 8.85 |
| MACRO-F1_FW + top-k init. | 46.77 | 34.21 | 15.61 | 11.16 | 12.35 | 43.48 | 51.65 | 14.93 | 14.98 | 13.69 | 27.92 | 64.11 | 12.14 | 28.42 | 14.63 |
| MACRO-F1_FW + rnd init. | 45.20 | 33.05 | 15.42 | 11.17 | 12.21 | 43.57 | 51.60 | 15.20 | 15.05 | 13.82 | 28.12 | 64.23 | 13.93 | 23.32 | 14.81 |
| **FLICKR** | | | | | | | | | | | | | | | |
| MACRO-P_FW + top-k init. | 4.46 | 10.84 | 40.41 | 7.87 | 9.33 | 2.53 | 10.44 | 38.11 | 7.84 | 8.04 | 1.75 | 13.99 | 37.31 | 10.69 | 8.42 |
| MACRO-P_FW + rnd init. | 4.65 | 11.49 | 39.34 | 6.63 | 8.06 | 5.66 | 22.75 | 41.74 | 9.70 | 10.57 | 2.83 | 22.26 | 37.59 | 10.68 | 8.50 |
| MACRO-R_FW + top-k init. | 16.14 | 38.62 | 17.58 | 45.50 | 22.27 | 12.17 | 47.48 | 13.98 | 53.83 | 19.72 | 7.89 | 60.42 | 9.57 | 64.66 | 15.07 |
| MACRO-R_FW + rnd init. | 16.14 | 38.62 | 17.58 | 45.50 | 22.27 | 12.17 | 47.48 | 13.98 | 53.83 | 19.72 | 7.89 | 60.42 | 9.57 | 64.66 | 15.07 |
| MACRO-F1_FW + top-k init. | 18.21 | 43.06 | 34.89 | 29.51 | 29.41 | 11.78 | 45.91 | 34.68 | 30.87 | 29.45 | 7.14 | 55.21 | 34.00 | 33.17 | 29.11 |
| MACRO-F1_FW + rnd init. | 17.59 | 41.60 | 35.28 | 29.28 | 29.43 | 12.22 | 47.31 | 34.13 | 32.70 | 29.43 | 5.92 | 45.77 | 34.55 | 33.08 | 29.02 |
| **RCV1X** | | | | | | | | | | | | | | | |
| MACRO-P_FW + top-k init. | 32.66 | 34.63 | 20.70 | 3.62 | 3.77 | 31.39 | 52.39 | 21.41 | 5.80 | 6.58 | 16.15 | 49.66 | 21.45 | 7.84 | 5.96 |
| MACRO-P_FW + rnd init. | 46.36 | 50.11 | 21.11 | 5.61 | 5.84 | 29.40 | 49.81 | 21.69 | 5.72 | 5.31 | 19.45 | 60.40 | 21.66 | 6.03 | 5.78 |
| MACRO-R_FW + top-k init. | 43.28 | 44.99 | 14.56 | 18.41 | 11.95 | 34.15 | 55.24 | 13.15 | 24.89 | 12.73 | 23.78 | 69.71 | 10.76 | 34.66 | 12.44 |
| MACRO-R_FW + rnd init. | 43.28 | 44.99 | 14.56 | 18.41 | 11.95 | 34.15 | 55.24 | 13.15 | 24.89 | 12.73 | 23.78 | 69.71 | 10.76 | 34.66 | 12.44 |
| MACRO-F1_FW + top-k init. | 58.14 | 61.31 | 21.58 | 10.36 | 12.03 | 44.29 | 71.93 | 22.37 | 12.26 | 13.65 | 27.96 | 80.26 | 22.53 | 13.76 | 15.20 |
| MACRO-F1_FW + rnd init. | 58.20 | 61.22 | 21.45 | 10.37 | 12.09 | 44.42 | 71.86 | 21.96 | 12.25 | 13.68 | 27.26 | 78.88 | 22.10 | 14.86 | 15.12 |

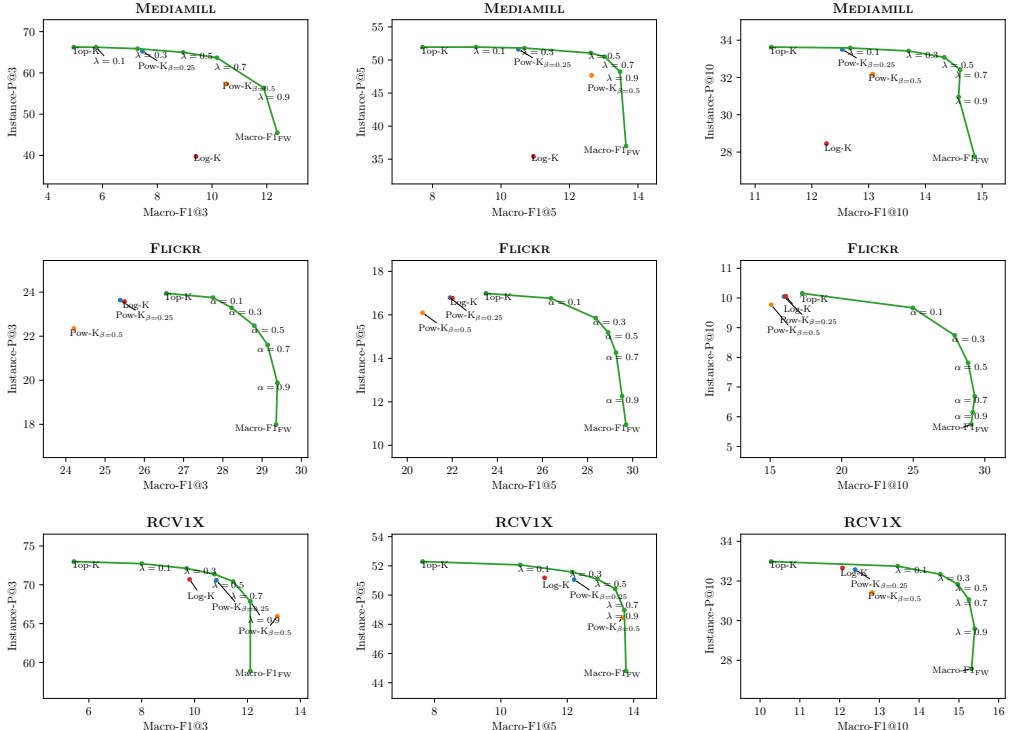

Figure 1: Comparison of the baseline algorithms with the PU inference with mixed objectives for $k \in \{3, 5, 10\}$. The green line shows the results for different interpolations between two measures.

## G.2 TOP-K VS RANDOM-INITIALIZATION

In this experiment, we investigate the impact of the initialization strategy in the Frank-Wolfe algorithm on the results. They consider two strategies for the initialization of initial classifier $\boldsymbol{h}^0$; one initialize the classifier that weights $\widehat{\eta}_j$ by a random positive number (rnd init.), second strategy initialize $\boldsymbol{h}^0$ with the top-k $\widehat{\eta}_j$ classifier (top-k init.). For all the experiments here, we use the same dataset for both training the $\boldsymbol{\eta}$ estimator and estimating confusion tensor $\mathbf{C}$ (100/100). We present the result of this comparison in Table 7. The results show that the initialization strategy has an impact on the results for $\text{MACRO-P}_{\text{FW}}$ variant of the algorithm. However it is not clear from the results which initialization variant is better and in which circumstances for $\text{MACRO-P}_{\text{FW}}$.

## G.3 RESULTS WITH MIXED UTILITIES

It can be noticed in the presented results that the optimization of macro-measures comes with the cost of a significant drop in performance on instance-wise measures, which in some cases may not be acceptable. To achieve the desired trade-off between tail and head label performance, one can optimize a mixed utility that is a linear combination of instance-wise measures and selected macro-measures. As an example, we present the results for such mixed utility that is a combination of instance-wise precision@$k$ with macro F1-measure@$k$:

$$\Psi(\mathbf{C}) := (1 - \lambda)\Psi_{\text{Instance-P}}(\mathbf{C}) + \lambda\Psi_{\text{Macro-F1}}(\mathbf{C})$$
$$= \sum_{j=1}^{m} (1 - \lambda)\psi_{\text{Instance-P}}(\boldsymbol{C}^j) + \lambda\psi_{\text{Macro-F1}}(\boldsymbol{C}^j) \tag{69}$$

In Figure 1, we present the plots with results on two combined measures for different values of $\lambda$. Once again, the presented results are the mean values over 10 runs of the inference. The plots show that the instance-vs-macro curve has a nice concave shape that dominates simple baselines in most cases. In particular, we can initially improve macro-measures significantly with only a minor drop in

instance-measures, and only if we want to optimize even more strongly for macro-measures, we get larger drops in instance-wise measures. A particularly notable feature of the plug-in approach is that the curves in the figure are cheap to produce since there is no requirement for expensive re-training of the entire architecture, so one can easily select an optimal interpolation constant according to some criteria, such as a maximum decrease of instance-wise performance.

## H  CONFUSION TENSOR MEASURES

In this section, we will take a closer look at the definitions of confusion tensor metrics, and provide some structural results. First, let us recall the definitions from the main text:

**Definition 3.1** (Binary Confusion Matrix Measure). *Let $\mathcal{C} = \left\{ \boldsymbol{C} \in [0,1]^{2 \times 2} \mid \|\boldsymbol{C}\|_{1,1} = 1 \right\}$ be the set of all possible binary confusion matrices, and $\boldsymbol{C}, \boldsymbol{C}' \in \mathcal{C}$. Then we say that $\boldsymbol{C}'$ is* at least as good *as $\boldsymbol{C}$, $\boldsymbol{C}' \succeq \boldsymbol{C}$, if there exists constants $\epsilon_1, \epsilon_2$ such that*

$$\boldsymbol{C}' = \begin{pmatrix} C_{00} + \epsilon_1 & C_{01} - \epsilon_1 \\ C_{10} - \epsilon_2 & C_{11} + \epsilon_2 \end{pmatrix}, \tag{2}$$

*i.e., if $\boldsymbol{C}'$ can be generated from $\boldsymbol{C}$ by turning some false positives to true negatives and false negatives to true positives. A function $\psi \colon \mathcal{C} \longrightarrow [0,1]$ is called a* binary confusion matrix measure *(Singh & Khim, 2022) if it respects that ordering, i.e., if for $\boldsymbol{C}' \succeq \boldsymbol{C}$ we have $\psi(\boldsymbol{C}') \geq \psi(\boldsymbol{C})$.*

**Definition 3.2** (Confusion Tensor Measure). *For a given number of labels $m \in \mathbb{N}$, and two confusion tensors $\mathbf{C}, \mathbf{C}' \in \mathcal{C}^m$, we say that $\mathbf{C}'$ is* at least as good *as $\mathbf{C}$, $\mathbf{C}' \succeq \mathbf{C}$, if for all labels $j \in [m]$ it holds that $\boldsymbol{C}^{j\prime} \succeq \boldsymbol{C}^j$. A function $\Psi \colon \mathcal{C}^m \longrightarrow [0,1]$ is called a* confusion tensor measure *if it respects this ordering, i.e., if for $\mathbf{C}' \succeq \mathbf{C}$ we have $\Psi(\mathbf{C}') \geq \Psi(\mathbf{C})$.*

Our first claim is that these form partial orders.

**Lemma H.1** (Partial order of confusion matrices). *The relation $\succeq$ introduced in Definition 3.1 forms a partial order on $\mathcal{C}$. Similarly, $\succeq$ from Definition 3.2 forms a partial order on $\mathcal{C}^m$.*

*Proof.* We start with the binary case. We need to show reflexivity, antisymmetry, and transitivity:
*Reflexivity:* By choosing $\epsilon_1 = \epsilon_2 = 0$, we see that $\boldsymbol{C} \succeq \boldsymbol{C}$.
*Antisymmetry:* Let $\boldsymbol{C} \succeq \boldsymbol{C}'$ and $\boldsymbol{C}' \succeq \boldsymbol{C}$. This implies $\epsilon_1 = \epsilon_2 = 0$, meaning $\boldsymbol{C} = \boldsymbol{C}'$. *Transitivity:* $\boldsymbol{C} \succeq \boldsymbol{C}'$ with coefficients $\epsilon_1, \epsilon_2$, and $\boldsymbol{C}' \succeq \boldsymbol{C}''$ with $\epsilon_1', \epsilon_2'$, then $\boldsymbol{C} \succeq \boldsymbol{C}''$ by choosing $\epsilon_1 + \epsilon_1'$, $\epsilon_2 + \epsilon_2'$.

The multi-label case follows directly, as it is just an $m$-fold Cartesian product of the binary case. $\square$

Next, we show a systematic way of turning binary confusion matrix measures into confusion tensor measures, using either *micro-* or *macro*-aggregation.

**Definition H.2** (Aggregation function). *For $n \in \mathbb{N}$, we call a function $f \colon [0,1]^n \longrightarrow [0,1]$ an* aggregation function *if it is nondecreasing in each of its arguments.*

**Theorem H.3** (Macro-Aggregation). *Let $\psi_1, \ldots, \psi_m$ be a collection of binary confusion matrix measures, and $\phi \colon [0,1]^m \longrightarrow [0,1]$ be an aggregation function. Then the* macro-aggregation

$$\Psi(\mathbf{C}) := \phi(\psi_1(\boldsymbol{C}^1), \ldots, \psi_m(\boldsymbol{C}^m)) \tag{70}$$

*is a confusion tensor measure.*

*Proof.* Let $\mathbf{C}' \succeq \mathbf{C}$. Then, for all labels $j$, $\boldsymbol{C}^{j\prime} \succeq \boldsymbol{C}^j$, which implies $\psi_j(\boldsymbol{C}^{j\prime}) \geq \psi_j(\boldsymbol{C}^j)$. As $\phi$ is nondecreasing in all of its arguments, this implies $\Psi(\mathbf{C}') \geq \Psi(\mathbf{C})$, concluding the proof. $\square$

**Theorem H.4** (Micro-Averaging). *Let $\psi$ be a binary confusion matrix measures, and $\phi$ be a* linear *aggregation function. Define the averaged confusion matrix by applying aggregation to each entry separately,*

$$\overline{\boldsymbol{C}} = \phi(\mathbf{C}) := \begin{pmatrix} \phi(C_{00}^1, \ldots, C_{00}^m) & \phi(C_{01}^1, \ldots, C_{01}^m) \\ \phi(C_{10}^1, \ldots, C_{10}^m) & \phi(C_{11}^1, \ldots, C_{11}^m) \end{pmatrix} \tag{71}$$

*Then the* micro-average

$$\Psi(\mathbf{C}) := \psi(\phi(\mathbf{C})), \tag{72}$$

*is a confusion tensor measure.*

*Proof.* Let $\mathbf{C}' \succeq \mathbf{C}$. Then, for all labels $j$, $\mathbf{C}^{j'} \succeq \mathbf{C}^j$, i.e., there exists a collection $(\epsilon_1^1, \epsilon_2^1), \ldots, (\epsilon_1^m, \epsilon_2^m)$ which transform $\mathbf{C}$ into $\mathbf{C}'$. Denote $\overline{\mathbf{C}'} = \phi(\mathbf{C}')$, and similarly $\overline{\mathbf{C}} = \phi(\mathbf{C})$. Due to the linearity of $\phi$, we have

$$\overline{C}'_{00} = \phi(C_{00}^{1'}, \ldots, C_{00}^{m'}) = \phi(C_{00}^1 + \epsilon_1^1, \ldots, C_{00}^m + \epsilon_1^m) = \phi(C_{00}^1, \ldots, C_{00}^m) + \phi(\epsilon_1^1, \ldots, +\epsilon_1^m).$$

Similar calculations can be done for the other components. This implies that

$$\overline{\mathbf{C}}' = \begin{pmatrix} \overline{C}_{00} + \phi(\epsilon_1^1, \ldots, +\epsilon_1^m) & \overline{C}_{01} - \phi(\epsilon_1^1, \ldots, +\epsilon_1^m) \\ \overline{C}_{10} + \phi(\epsilon_2^1, \ldots, +\epsilon_2^m) & \overline{C}_{11} - \phi(\epsilon_2^1, \ldots, +\epsilon_2^m) \end{pmatrix}, \tag{73}$$

i.e., $\overline{\mathbf{C}}' \succeq \overline{\mathbf{C}}$, and thus $\Psi(\mathbf{C}') \geq \Psi(\mathbf{C})$. $\qquad\square$

If the aggregation function is chosen to be the arithmetic mean, the two cases above reduce to regular macro- and micro-averaging. This justifies our choice of (3) in the main paper, proving that this indeed does result in an admissible confusion tensor metric.

Note that for micro-aggregation, we had to be much more strict in what aggregation functions to admit, essentially limiting to weighted arithmetic mean, because we need to ensure that the component-wise averaging of confusion matrices results in matrices that are comparable using the partial order.

Finally, we can also provide the following structural result:

**Theorem H.5.** *Let $\Psi_1, \ldots, \Psi_n$ be a collection of confusion tensor losses, and $\phi$ an aggregation function. Then*

$$\Psi(\mathbf{C}) = \phi(\Psi_1(\mathbf{C}), \ldots, \Psi_n(\mathbf{C})) \tag{74}$$

*is a confusion tensor loss.*

*Proof.* Let $\mathbf{C}' \succeq \mathbf{C}$, then $\Psi_i(\mathbf{C}') \geq \Psi_i(\mathbf{C})$. Thus, by monotonicity of $\phi$, we get $\Psi(\mathbf{C}') \geq \Psi(\mathbf{C})$. $\quad\square$

This latter result implies that, e.g., calculating the harmonic mean of macro-precision and macro-recall is also a confusion tensor loss.

## I    SENSITIVITY OF MACRO-AT-$k$ METRICS TO TAIL LABELS

The experiment described in this section has been motivated by a similar one given in (Wei & Li, 2019). In Table 8 we compare different metrics for budgeted at $k$ predictions. We train a probabilistic label tree (PLT) model on the full AMAZONCAT dataset (McAuley et al., 2015) and on a reduced version with the 1000 most popular labels only. The test is performed for both models on the full set of labels. The standard metrics are only slightly perturbed by reducing the label space to the head labels. This holds even for propensity-scored precision (Jain et al., 2016), a popular measure for evaluating tail labels in extreme multi-label classification, which decreases by just 1%-20% despite discarding over 90% of the label space. In contrast, the macro-at-$k$ measures decrease between 60% and 90% if tail labels are ignored.

Table 8: Performance measures (%) on AmazonCat-13k of a classifier trained on the full set of labels and a classifier trained with only 1k head labels.

| Metric | full labels | | | head labels | | |
|---|---|---|---|---|---|---|
| | @1 | @3 | @5 | @1 (diff.) | @3 (diff.) | @5 (diff.) |
| Precision | 93.03 | 78.51 | 63.74 | 93.08 (+0.05%) | 76.42 (-2.66%) | 58.21 (-8.67%) |
| nDCG | 93.03 | 87.25 | 85.35 | 93.08 (+0.05%) | 85.75 (-1.71%) | 80.91 (-5.19%) |
| PS-Precision | 49.76 | 62.63 | 70.35 | 49.07 (-1.39%) | 57.71 (-7.84%) | 57.41 (-18.40%) |
| Macro-Precision | 13.28 | 32.65 | 44.16 | 4.31 (-67.54%) | 5.28 (-83.82%) | 4.32 (-90.21%) |
| Macro-Recall | 1.38 | 11.06 | 30.57 | 0.47 (-65.61%) | 2.69 (-75.71%) | 4.10 (-86.59%) |
| Macro-F1 | 2.26 | 14.67 | 32.84 | 0.74 (-67.37%) | 3.10 (-78.88%) | 3.77 (-88.51%) |

