# OpenReview forum: "Consistent algorithms for multi-label classification with macro-at-$k$ metrics"
_ICLR.cc/2024/Conference — ICLR 2024 poster_

### Official Review · Reviewer_s4jL · 2023-10-27

**Soundness:** 3 good
**Presentation:** 3 good
**Contribution:** 3 good
**Rating:** 8
**Confidence:** 4

**Summary:**

The manuscript studies the problem of multi-label classification with budgeted (top-k) prediction metrics. The problem is very well established by now, and of much practical relevance. The paper is overall well motivated, well-written, and the contributions are par for the venue.

I see two key contributions: 1) formalizing the empirical uitility maximation notion of '@k' metrics as that of optimizing over certain classes of confusion matrices, which decomposes over individual labels, but the optimization problem itself doesn't decompose easily because of the constraint that any classifier in the hypothesis space must output exactly k labels per instance; 2) deriving a neat form for the Bayes optimal, which yields interpretable closed form solutions for well-known metrics like recall and balanced accuracy; the most-general closed form solution is not very useful because it depends on the optimal values --- this observation by itself is not novel, as pretty much every paper that talks about non-decomposable losses for binary/multi-label classification problems (several of which are clearly cited in the paper) have developed similar results.

Given the form of the Bayes optimal, Frank-Wolfe based algorithm for estimating the classifiers can be written down, following the work of Narasimhan et al. (2015). The empirical results demonstrate the effectiveness of the algorithm, and the consistency between optimal Bayes rule and the algorithmic convergence for certain measures, compared to several natural baselines.

Overall, I like the work and I am inclined to accept. I've some minor concerns which I outline under 'weaknesses' section. It would be good to hear from the authors on these questions in their rebuttal.

-- Post rebuttal --
Most of my concerns are addressed. I'm more positive about the paper now.

**Strengths:**

1. Clearly formulated problem -- it's non-trivial to set up the problem, and I really like the simplicity of the formulation in terms of confusion matrices/tensors, and the authors have done a great job of presenting notation-heavy material with careful development of ideas.
2. Technical rigor -- the theoretical results are well established, and the supporting key lemmas are included in the main paper, which is very helpful.

**Weaknesses:**

The intro neatly positions the problem in the context of several closely-related work in this space, but I felt those connections didn't surface as much as I'd have loved to see in the main paper.

For instance, the first main result in Theorem 4.1. Just looking at the constants a_j and b_j, one can see, not surprisingly, that these are the same constants one would see for weighted binary classification problems, see Lemma 2 of https://www.jmlr.org/papers/volume18/15-226/15-226.pdf (of course, there's no top-k notion here in binary classification). It would be good to draw these connections, position the main results in the context of known results, and tease apart new observations and insights.

Similarly, with respect to, Theorem 4.4, it would be good to see some form of the result, say special cases, that can be dotted lined to known results for multi-label problems.

**Questions:**

It would be good to address the points raised in the weaknesses section. And a note on the challenges or novelty in the proofs for Thm 4.4 over and above known work.

It's nice to see the consistency between Macro-R_prior and Macro-R_FW in the experiments for the recall metric. Did the authors also experiment with balanced accuracy in the experiments, where the closed form solution is also easy to compute?

**Details Of Ethics Concerns:**

A lot of the formulation developed in the paper reminded me of this result I came across a few years ago https://arxiv.org/pdf/1908.09057.pdf. It turns out there is a lot in common (Corollary 3 and 4 in this arXiv I believe are analogous to Theorems 4.1 and 4.4), but the current submission also some algorithmic contributions. Perhaps the authors are overlapping, but I just wanted to make this note here.

---

> ### Author Response · Authors · 2023-11-17
> **Response to Reviewer s4jL (1/2)**
>
> We sincerely thank the reviewer for the insightful review. We appreciate the hard work. Below, we address the main questions and concerns:
>
> ---
>
> > For instance, the first main result in Theorem 4.1. Just looking at the constants a_j and b_j, one can see, not surprisingly, that these are the same constants one would see for weighted binary classification problems, see Lemma 2 of https://www.jmlr.org/papers/volume18/15-226/15-226.pdf (of course, there's no top-k notion here in binary classification). It would be good to draw these connections, position the main results in the context of known results, and tease apart new observations and insights.
>
> The cost-sensitive binary classification is indeed a special case of our linear metric setup with a single label, which is why these constants turn out to be the same. In our work, we choose top-$k$ labels with respect to a linear transformation of marginals, while in binary classification a single label is selected whenever that linear transformation of class conditional probability is non-negative. Having said that, we agree that we should make a better connection between the results presented in our paper and known results.
>
> ---
>
> > Similarly, with respect to, Theorem 4.4, it would be good to see some form of the result, say special cases, that can be dotted lined to known results for multi-label problems.
>
> The easiest special case are linear/weighted measures, i.e., $\Psi(\mathbf{C}) = \mathbf{L} \cdot \mathbf{C} + d$.  In that case, $\nabla \Psi = \mathbf{L}$, so the statement reduces to the trivial claim that any $\mathbf{C}$ maximizing $\mathbf{L} \cdot \mathbf{C}$ also maximizes $\mathbf{L} \cdot \mathbf{C} + d$.
>
> For a more interesting case, consider macro-recall: Here,
> $$
> \Psi(\mathbf{C}) = \sum_j C^j_{11} / (C^j_{11} + C^j_{10}),
> $$
> and
> $$
> \nabla_{\boldsymbol{C}^j} \Psi = \begin{pmatrix}
> 0 & 0 \\\\
> C^j_{11} / (C^j_{11} + C^j_{10})^2 & -C^j_{10} / (C^j_{11} + C^j_{10})^2
> \end{pmatrix}.
> $$
>
> Thus, the theorem claims that one needs to optimize the linear measure
> $$\sum_j (C^{\star j}\_{11} + C^{\star j}\_{10})^{-2} (C^{\star j}\_{11} C^j\_{10}
> -C^{\star j}\_{10} C^j\_{11}).
> $$
>
> Note that, in fact, for any given problem, $C^j_{10} + C^j_{11} = \pi_j$ is a constant, so we can simplify the expression:
> $$
> \sum_j \pi_j^{-2} C^{\star j}\_{11} (\pi_j - C^j\_{11}) - (\pi_j - C^{\star j}\_{11}) C^j\_{11} = \sum_j \pi_j^{-1} (C^{\star j}\_{11} - C^j\_{11}).
> $$
> As macro-recall is formulated as a loss function, our goal is to minimize the expression above. As $\pi_j^{-1} C^{\star j}\_{11}$ is a constant, the optimization objective becomes
> $$
> \min \sum_j - \pi_j^{-1} C^j\_{11} = \max \sum_j \pi_j^{-1} C^j\_{11}.
> $$
>
> Thus, this recovers the solution for optimizing macro-recall.
>
> ---
>
> > It's nice to see the consistency between Macro-R_prior and Macro-R_FW in the experiments for the recall metric. Did the authors also experiment with balanced accuracy in the experiments, where the closed form solution is also easy to compute?
>
> We have performed such an experiment, and we can confirm that the proposed Frank-Wolfe algorithm (Macro-BA$\_{FW}$) also recovers the optimal analytical solution for Balanced Accuracy obtained directly from label priors (Macro-BA$_{PRIOR}$). We present the results below, showing that both approaches get almost the same performance.
>
> **Mediamill:**
> | method                         | mBA@3      | mBA@5      | mBA@10     |
> |:-------------------------------|-----------:|-----------:|-----------:|
> | Macro-BA$_{PRIOR}$             | *58.51*    | **60.93**  | **65.03**  |
> | Macro-BA$_{FW}$                | **58.51**  | *60.93*    | *65.03*    |
>
> **Flickr:**
> | method                         | mBA@3      | mBA@5      | mBA@10     |
> |:-------------------------------|-----------:|-----------:|-----------:|
> | Macro-BA$_{PRIOR}$             | **72.02**  | **75.70**  | **79.86**  |
> | Macro-BA$_{FW}$                |*71.93*     | *75.63*    | *79.73*    |
>
> **RCV1x:**
> | method                         | mBA@3      | mBA@5      | mBA@10     |
> |:-------------------------------|-----------:|-----------:|-----------:|
> | Macro-BA$_{PRIOR}$             | **58.58**  | *61.54*    | *66.02*    |
> | Macro-BA$_{FW}$                | *58.58*    | **61.55**  | **66.03**  |
>
> **AmazonCat:**
> | method                         | mBA@3      | mBA@5      | mBA@10     |
> |:-------------------------------|-----------:|-----------:|-----------:|
> | Macro-BA$_{PRIOR}$             | *71.62*    | **74.04**  | **76.28**  |
> | Macro-BA$_{FW}$                | **71.63**  | *74.01*    | *76.26*    |

---

> > ### Comment · Reviewer_s4jL · 2023-11-22
> > **Thanks for the responses**
> >
> > Thank you, it's good to see that the method is competitive to optimal strategy in case of balanced error as well. It would be great if you could add some of these observations to the main text.

---

> ### Author Response · Authors · 2023-11-17
> **Response to Reviewer s4jL (2/2)**
>
> ---
>
> ### **Regarding ethical concerns (comparison to [Wang et al](https://arxiv.org/pdf/1908.09057.pdf))**
>
> We thank the reviewer for bringing this work to our attention. We were not aware of it (if not wrong, it was not published in a peer-reviewed venue or journal). After reading it, we see substantial differences from our submission.
>
> The reviewer is correct that this paper does look similar to ours. However, this is not surprising; both papers are different extensions of earlier works, such as [Narasimhan et al](https://proceedings.mlr.press/v37/narasimhanb15.html). We tried to clearly indicate in our submission that it is based on this line of work. Below we discuss the differences between our paper and [Wang et al](https://arxiv.org/pdf/1908.09057.pdf).
>
>
> That paper considers _multi-output_ classifiers with $K$ classes in each output, and makes $M$ such outputs, which is conceptually quite different from at-$k$ prediction considered in our paper. This can already be seen in the respective definitions of the confusion tensor: In our case, it is an $m \times 2 \times 2$ object, and in the other paper, it is an $M \times K \times K$ object.
>
> In case of multi-output classification, each of the outputs chooses a class _independently_ of the other. Of course, an $m$-label multilabel problem can be considered as a multi-output problem, with $K=2$ and $M=m$. In that case, though, the setting of Wang et al. does not contain the at-$k$ constraint, which is what makes this problem really challenging.
>
> On a technical side, our formulation of the theoretical results and our proof techniques are different (as a side note, their proof of the compactness of the feasible confusion set is incorrect as it falsely assumes the space of bounded functions to be compact).
>
> On the basis of the above arguments, we hope that we have assured the reviewer that we do not plagiarize any other work. Nevertheless, we admit that paper of Weng et al. is related to our submission and we should cite it.

---

> > ### Comment · Reviewer_s4jL · 2023-11-22
> > **Thanks**
> >
> > Thank you, I see that the notation and set up do draw commonly from Narasimhan et al. I wanted to register my concern because the overlap seemed particularly striking. Thanks for calling out the differences -- I do realize that the submission makes enough technical advances and is sufficiently different compared to the arXiv work.
> >
> > I'm raising the score to clear accept.

---

### Official Review · Reviewer_gGLD · 2023-10-31

**Soundness:** 3 good
**Presentation:** 3 good
**Contribution:** 4 excellent
**Rating:** 8
**Confidence:** 3

**Summary:**

In the paper "Consistent algorithms for multi-label classification with macro-at-k metrics", the authors propose a framework of consistent multi-label learning algorithms for targeting macro-averaged metrics that are however budgeted for a k-subset of labels. The presented approach is based on the Frank-Wolfe algorithm and represents a principled extension towards multi-label classification with corresponding theoretical guarantees. An empirical study confirms the consistency with the targeted metric.

**Strengths:**

- Theoretical sound and underpinned approach for targeting macro-averaged at k metrics for multi-label classification.
- An empirical study confirms the theoretical findings for four datasets and various metrics.
- The overall presentation and language of the paper are very good.

**Weaknesses:**

- The empirical evaluation is relatively limited as only four datasets are considered. However, the main focus should remain on the theoretical findings here.
- The theoretical results could be accompanied by an intuition to ease understanding of the results.
- A comparison to something like binary relevance learning treating each label independently for demonstrating also empirically that these measures cannot be sufficiently tackled by such an approach would be desirable.

minor:
p. 4 "define in Table 1"
p. 7 "tensor measure measure"

**Questions:**

- In https://doi.org/10.1007/s10994-021-06107-2 an @k metric interpolating between  Hamming and subset 0/1 loss is presented. To what extent would this relate to the considered @k-metrics in this paper?

---

> ### Author Response · Authors · 2023-11-17
> **Response to Reviewer gGLD**
>
> We sincerely thank the reviewer for the review. We appreciate the hard work. Below, we address the question and concern:
>
> ---
>
> > The theoretical results could be accompanied by an intuition to ease understanding of the results.
>
> We thank the reviewer for the suggestion; we will try to incorporate more intuitions in the next revision. If the reviewer has specific suggestions regarding which results would benefit the most from them, we would be happy to focus on these parts.
>
> ---
>
> >A comparison to something like binary relevance learning treating each label independently for demonstrating also empirically that these measures cannot be sufficiently tackled by such an approach would be desirable.
>
> For metrics for which macro averages can be written in a linear form (see page 5), the optimal solution can also be obtained by binary relevance-like approaches. In the experiments, we explicitly include methods (e.g., Top-K, Macro-R$_{\text{PRIOR}}$) of this type. In the general case, binary relevance methods do not lead to the optimal solution, as discussed in Appendix E.
>
> ---
>
> > In https://doi.org/10.1007/s10994-021-06107-2 an @k metric interpolating between Hamming and subset 0/1 loss is presented. To what extent would this relate to the considered @k-metrics in this paper?
>
> While the Hamming loss is a part of the framework we consider in our work, the 0/1 loss is not, as it cannot be represented as a function of the confusion tensor we use. That also applies to the family of losses introduced by [Hüllermeier et al. 2022](https://doi.org/10.1007/s10994-021-06107-2), which are a generalization of these two losses. Therefore, we cannot consider them in our framework.
>
> ---
>
> > minor: p. 4 "define in Table 1" p. 7 "tensor measure measure"
>
> We thank the reviewer for noticing and letting us know.

---

> > ### Comment · Reviewer_gGLD · 2023-11-20
> > **Re: Response to Reviewer gGLD**
> >
> > Thank you for your brief and to-the-point response.
> >
> > > While the Hamming loss is a part of the framework we consider in our work, the 0/1 loss is not, as it cannot be represented as a function of the confusion tensor we use. That also applies to the family of losses introduced by Hüllermeier et al. 2022, which are a generalization of these two losses. Therefore, we cannot consider them in our framework.
> >
> > Maybe I get it wrong but your work and the work by Hüllermeier et al. both focus on predicing a subset of labels correctly. While in your work this is fixed for a specific K, in Hüllermeier et al. the K is a parameter of choice for the user ranging from 1 resembling Hamming loss and the number of labels to obtain subset 0/1 loss. Of course, this work here looks at macro-averaged metrics but still assigns a degree of importance to a subset of labels, namely the top-K labels. So I do believe that there is a connection although it might be subtle and that you cannot consider subset 0/1 loss in your framework.

---

> ### Author Response · Authors · 2023-11-22
> **The difference between the approach of Hüllermeier et al. and ours**
>
> The key difference between the approach of Hüllermeier et al. and ours is that
> the former measures performance *jointly on each instance*, while we focus on macro-averages.
>
> In our work, $k$ is a strict requirement on the number of predicted positive labels per instance. In case of the other work, $k$ is a parameter of a loss function that indicates the minimal size of correctly predicted subsets of labels for an instance that will be rewarded by the loss function. It does not limit in any way the number of predicted positive labels.
>
> To see the difference, consider $k = 1$. In our case, we will predict only one label per instance. The Bayes classifier will be a label with the highest value of the linear transformation of the label marginal (with parameters $\mathbf{a}$ and $\mathbf{b}$ depending on the base loss function and data distribution). In case of Hüllermeier et al., $k=1$ leads to the standard Hamming loss without any restriction on the number of predicted labels. The optimal prediction is then a set of labels with marginal probabilities being greater than (or equal) 0.5.

---

### Official Review · Reviewer_XNZA · 2023-11-01

**Soundness:** 3 good
**Presentation:** 2 fair
**Contribution:** 3 good
**Rating:** 6
**Confidence:** 3

**Summary:**

This paper aims to find consistent algorithms for macro-at-k metrics, which is widely-used in many long-tailed multi-label classification problems. For the multi-label problem, the author shows such optimal classifiers can be derived by selecting top-k scoring labels based on an affine transformation of the marginal label probabilities (which is unknown in practice). They further presents a Frank-Wolfe algorithm to empirically find the optimal classifiers.

**Strengths:**

- The technical derivation of this paper seems solid

**Weaknesses:**

- The writing of this paper is not easy to follow
- Some baseline methods are missing in experiments

**Questions:**

- Q1: From the results of Table 2, the proposed method seems to greatly sacrifice instance-wise metrics to trade for gains in macro-average metrics. Is it possible for the proposed method to optimize a interpolated version of the objective that flexibly control the performance tradeoff between instance-wise metrics and macro-averaged metrics?

- Q2: Some baseline methods that claim to also perform good on tail-labels are not discussed in related work, or compared in the experiment section. For example [1] and [2], to name just a few.

- Q3: To improve the clarity of the proposed methods, the author may consider a toy synthetic dataset where data distributions are known, and show derivations of the proposed method, and verify the consistency property through simulations.

- Q4: This submission also seems highly related to [3]. The author should discuss what's the difference, and compare it empirically.


### Reference
- [1] Menon et al. Long-Tail Learning via Logit Adjustment. ICLR 2020.
- [2] Zhang et al. Long-tailed Extreme Multi-label Text Classification by the Retrieval of Generated Pseudo Label Descriptions. EACL 2023.
- [3] Schultheis et al. Generalized test utilities for long-tail performance in extreme multi-label classification. NeurIPS 2023.

---

> ### Author Response · Authors · 2023-11-17
> **Response to Reviewer XNZA (1/2)**
>
> We sincerely thank the reviewer for the review. We appreciate the hard work. Below, we address the questions:
>
> ---
>
> > From the results of Table 2, the proposed method seems to greatly sacrifice instance-wise metrics to trade for gains in macro-average metrics. Is it possible for the proposed method to optimize a interpolated version of the objective that flexibly control the performance tradeoff between instance-wise metrics and macro-averaged metrics?
>
> Yes, it is possible to optimize a measure that interpolates between instance-wise metric and macro-metric as long as the resulting objective is a function of confusion tensor $\mathbf{C}$ and ideally meets all assumptions required by the Frank Wolfe procedure.
> Such objective is, for example, a linear interpolation between instance-precision@k and macro-f1@k:
> $$
> \Psi(\mathbf{C}) = (1 - \alpha) \text{Precision}@k(\mathbf{C}) + \alpha \text{Macro-F1}(\mathbf{C}).
> $$
>
> Below, we link to the image presenting plots for two datasets, Mediamill and Flickr. The results show that, in this case, the instance-vs-macro curve has a nice concave shape that dominates simple baselines. In particular, we can initially improve macro-measures significantly with only a small drop in instance-measures. We will include this discussion and more results in the appendix.
>
>
> Plots: https://i.imgur.com/LJTTxyq.png
>
> ---
>
> > Some baseline methods that claim to also perform good on tail-labels are not discussed in related work, or compared in the experiment section. For example [1] and [2], to name just a few.
>
> In the paper, we included a comparison with some popular techniques that are domain-agnostic. However, we will be happy to include a comparison with more methods of that type if the reviewer has some particular in mind. Regarding the ones already mentioned by the reviewer:
>
> - [1] considers multi-class ($y \in [m]$) setting, while in this work we consider multi-label ($\mathbf{y} \in \\{0,1\\}^m$) setting. Post-hoc logit adjustment presented in Section 4 of [1] is the simple weighting by the inverse of priors, which we also use in experiments (denoted as $\textrm{Macro-R}_{\text{PRIOR}}$).
>
> - [2] heavily relies on the problem domain (textual data) and cannot be applied to all datasets we consider. Please also note that the approach proposed in our submission can be used on top of any method aiming at improving the probability estimates of tail labels. We will try to highlight it more in the revised version of the paper.
>
> ---
>
> > To improve the clarity of the proposed methods, the author may consider a toy synthetic dataset where data distributions are known, and show derivations of the proposed method, and verify the consistency property through simulations.
>
> We thank the reviewer for the suggestion! We will add an experiment on synthetic data to improve the presentation of our paper.

---

> ### Author Response · Authors · 2023-11-17
> **Response to Reviewer XNZA (2/2)**
>
> ---
>
> > This submission also seems highly related to [3]. The author should discuss what's the difference, and compare it empirically.
>
> Our submission and paper [3] look at the same problem setup, but using two different optimization frameworks. The latter follows  the _Expected Test Utility_ (ETU) framework where one assumes a _given fixed_ test set $\boldsymbol{X}$ and then tries to predict the corresponding labels $\boldsymbol{Y}$. This has important consequences, both theoretical and practical.
>
> From a theoretical perspective, by having a fixed test set, the optimal classifier is always deterministic, and one essentially solves a discrete optimization problem over the space of all at-$k$ predictions. As this becomes computationally intractable except for trivial cases, [3] needs to make an approximation, actually optimizing $\Psi(\mathbb{E}[\mathbf{C}(\boldsymbol{Y}, \hat{\boldsymbol{Y}}) \mid \boldsymbol{X}])$ instead of the actual ETU objective, $\mathbb{E}[\Psi(\mathbf{C}(\boldsymbol{Y}, \hat{\boldsymbol{Y}})) \mid \boldsymbol{X}]$. This objective is then minimized using a block-coordinate-ascent algorithm.
>
> In contrast, the _Population Utility_ (PU) framework adopted in this work means we are looking for (potentially stochastic) classifiers $\boldsymbol{h}$ that work on the level of _a single instance_. Thus, we want to optimize $\Psi(\mathbb{E}[\mathbf{C}(\boldsymbol{y}, \boldsymbol{h}(\boldsymbol{x}))])$. While this resembles the approximation above, there are important differences. Firstly, the expectation is no longer conditioned on the test set $\boldsymbol{X}$. Secondly, instead of matrices, $\boldsymbol{Y}$ and $\boldsymbol{X}$, we now only need vectors $\boldsymbol{y}$ and $\boldsymbol{x}$, as the expectation is taken over single instances.
>
> From a practical point of view, this means that the PU-classifier introduced in this paper can be employed in an online inference setting, where queries are presented one-by-one, instead of being in one large batch. In contrast, if you apply the ETU algorithm [3] in online inference, i.e., independently apply to test sets of size 1, it degenerates back to the top-k prediction.
> However, when evaluating on a given test set, the ETU solution can play its strengths, since it is possible to exploit the actual instances present in the test set, which PU cannot do. Thus, when using the same label probability estimator $\boldsymbol{\eta}$, we generally expect ETU to perform better than PU.
>
> To illustrate our expectation empirically, we present below the results obtained for the Flickr data set (the method introduced in [3] is denoted in the table below with $_{BCA}$ subscript).
>
> **Flickr:**
> | method                         | iP@5       | iR@5       | mP@5       | mR@5       | mF@5       |
> |:-------------------------------|-----------:|-----------:|-----------:|-----------:|-----------:|
> | Top-K                          | **16.99**  | **65.90**  | 17.11      | 46.88      | 23.47      |
> | Top-K + $w^{\text{POW}}$       | 16.10      | 62.69      | 13.76      | 52.22      | 20.67      |
> | Top-K + $w^{\text{LOG}}$       | 16.76      | 65.10      | 15.05      | 49.60      | 21.99      |
> | Top-K + $\ell_{\text{FOCAL}}$  | *16.89*    | *65.50*    | 18.54      | 45.53      | 24.14      |
> | Top-K + $\ell_{\text{ASYM}}$   | 16.73      | 64.91      | 17.39      | 45.47      | 23.58      |
> | Macro-P$_{FW}$                 | 5.66       | 22.71      | *41.74*    | 9.67       | 10.55      |
> | Macro-R$_{BCA}$                | 0.39       | 1.63       | **65.80**  | 3.74       | 3.29       |
> | Macro-R$_{PRIOR}$              | 12.17      | 47.40      | 13.98      | *53.64*    | 19.71      |
> | Macro-R$_{FW}$                 | 12.17      | 47.40      | 13.98      | *53.64*    | 19.71      |
> | Macro-R$_{BCA}$                | 12.63      | 49.12      | 12.93      | **55.44**  | 18.86      |
> | Macro-F1$_{FW}$                | 11.78      | 45.83      | 34.68      | 30.77      | *29.40*    |
> | Macro-F1$_{BCA}$               | 11.23      | 43.53      | 32.36      | 33.21      | **30.70**  |

---

> > ### Comment · Reviewer_XNZA · 2023-11-22
> >
> > I am satisfied with the authors response. I will keep my score the same.

---

### Official Review · Reviewer_gTuy · 2023-11-01

**Soundness:** 3 good
**Presentation:** 3 good
**Contribution:** 3 good
**Rating:** 8
**Confidence:** 3

**Summary:**

Paper presents an approach for multi-label classification with a budget using so called macro-at-k metrics which are linearly decomposable into a sum of binary classification utilities, and could be useful in the case of extreme classification with large number of imbalanced labels. This leads to challenging optimisation problem which is tackled using Frank-Wolfe method on label-wise confusion matrices. Theoretical underpinnings of producing consistent classifier are analysed, and performance improvement of macro-at-k against top-k strategies are shown on four different multi-label benchmark datasets with different number of labels and label distribution.

**Strengths:**

The proposal is theoretical sound, and detailed analysis of the properties of proposed metrics as well as how to build the consistent classifier based on confusion matrix measure and tensor representation, are shown. One practical solution for optimising the classifiers with proposed macro metrics, is derived. In empirical evaluations, four benchmark dataset with two base classifiers (MLP, Sparse linear model) are used to compare macro-at-k metrics with more straightforward top-k heuristics. In most cases, proposed approach could improve the precision and recall (and F1 score) in different k-budget levels, whereas on the largest dataset more simple baseline heuristics seems to work better. To my knowledge, the setting of considering budgeted macro-at-k in multi-label classification is novel, providing interesting approach and new knowledge to extreme multi-label problems.

**Weaknesses:**

Although paper shows good theoretical background and promising results, it lacks some of the detailed analysis and discussion of the results, especially in a broader sense. For instance, the discussion which of proposed metrics and macro-at-k approach should be chosen for different problems from practitioners perspectives, would strengthen the presentation. Also, manuscript is missing the analysis of computational complexity and computational times (of optimising the classifiers) and how these relate to size of the budget and other chosen parameters, as well as how these compare between different heuristics.

**Questions:**

- What would be the conclusions or "rule of thumb" of selecting particular heuristics from the practitioners' point of view for certain applications or multi-label classification problem?
- What are the computational costs of proposed approach and how these relate  the size of k?

---

> ### Author Response · Authors · 2023-11-17
> **Response to Reviewer gTuy (1/2)**
>
> We sincerely thank the reviewer for the insightful review. We appreciate the hard work. Below, we address the main questions and concerns:
>
> ---
>
> > It lacks (...) the discussion which of proposed metrics and macro-at-k approach should be chosen for different problems from practitioners perspectives
>
> > What would be the conclusions or "rule of thumb" of selecting particular heuristics from the practitioners' point of view for certain applications or multi-label classification problem?
>
> The basic heuristics might be more robust than the FW method in the case of datasets with very long tails and a very small number of positive instances. In this case, simplicity might be preferred over the sophisticated optimization procedure (which is the general rule of thumb in statistics and machine learning). Nevertheless, we believe that by improving the probability estimation for long tail labels (what we do not discuss in this paper), we should also get a more stable and better performance of the FW method.
>
> Regarding the metrics, macro-averaging treats all the labels equally important. This prevents ignoring tail labels (when used with an imbalance-sensitive binary loss). The budget of $k$ predictions requires the prediction algorithm to choose the labels “wisely.” The choice of the base metric (e.g., precision, recall, F-measure, etc.) depends on a given application, exactly in the same way as in the case of binary classification applications. For example, suppose a user wants to cover as many as possible relevant labels. In that case, they should use recall, but in our setting, the budget per each instance is precisely defined and recall for each label has the same importance.

---

> > ### Author Response · Authors · 2023-11-17
> > **Response to Reviewer gTuy (2/2)**
> >
> > ---
> >
> > > Manuscript is missing the analysis of computational complexity and computational times (of optimizing the classifiers) and how these relate to the size of the budget and other chosen parameters, as well as how these compare between different heuristics
> >
> > > What are the computational costs of proposed approach and how these relate the size of k?
> >
> > During the inference, the probabilistic classifier returned by the Frank-Wolfe procedure selects top $k$ labels according to selected $\boldsymbol{a}$ and $\boldsymbol{b}$ parameters, what has time complexity the same as other baselines (that is $O(m)$ per instance).
> >
> > A single iteration of Frank-Wolfe (FW) for macro-at-k metrics has both time and space complexity of $O(n_2m + m)$, where $m$ is the number of labels, $n_2$ is the size of a dataset used to estimate the confusion matrix in the FW algorithm ($\mathcal{S}_2$ in Algorithm 1).
> > In case of time complexity, we need $m$ operations for calculating the gradient of $\mathbf{C}$ and values of $\boldsymbol{a}$ and $\boldsymbol{b}$, and $n_2m$ operations for selecting top $k$ predictions for each instance in $\mathcal{S}_2$ according to $\boldsymbol{a} \odot \boldsymbol{\eta}(\boldsymbol{x}) + \boldsymbol{b}$ and estimation of next $\mathbf{C}$.
> > In case of space complexity, we need $n_2m$ units to store $m$ probability estimates and labels for each instance in $\mathcal{S}_2$, and additional $m$ units are required for $\mathbf{C}$, its gradients, $\boldsymbol{a}$ and $\boldsymbol{b}$, etc.
> >
> > The Frank-Wolfe algorithm usually requires a few iterations (in our implementation, we stop it if the number of iterations reaches a maximum, which was 20 in our experiments, or $\alpha \le \epsilon$, which was 0.0001 in our experiments). We include running times (in seconds) and a number of iterations in the table below (for a single run of the Frank-Wolfe procedure).
> >
> > From the presented data, it is hard to observe a simple relation of $k$ and running time. The required number of iterations depends on the difficulty of the optimization problem that depends not only on $k$ but also on label distribution and the quality of achieved probability estimates.
> >
> > We implemented the Frank-Wolfe algorithm in Python using Numpy and Pytorch; we believe that further speed-up can be achieved using lower-level programming language and programming methods.
> > For datasets with a very large number of labels, we can truncate probability estimates very close to 0 and use sparse data structures to speed up the algorithm further.
> >
> > We will include the above analysis and results in the appendix.
> >
> > **Mediamill:**
> > | method | time@3/s| iter.@3 | time@5/s | iter.@5 | time@10/s  | iter.@10|
> > |:----|--:|--:|--:|--:|--:|--:|
> > | Macro-P$_{FW}$  | 67.59| 11| 70.33| 10| 56.60| 7 |
> > | Macro-R$_{FW}$  | 27.69| 4 | 27.70| 4 | 28.41| 4 |
> > | Macro-F1$_{FW}$ | 45.60| 11| 42.44| 10| 29.33| 5 |
> >
> > **Flickr:**
> > | method | time@3/s| iter.@3 | time@5/s | iter.@5 | time@10/s  | iter.@10|
> > |:----|--:|--:|--:|--:|--:|--:|
> > | Macro-P$_{FW}$  | 85.52| 9 | 87.73| 9 | 110.90  | 12|
> > | Macro-R$_{FW}$  | 50.80| 4 | 51.55| 4 | 50.84| 4 |
> > | Macro-F1$_{FW}$ | 55.23| 5 | 50.10| 4 | 61.97| 6 |
> >
> > **RCV1x:**
> > | method | time@3/s| iter.@3 | time@5/s | iter.@5 | time@10/s  | iter.@10|
> > |:----|--:|--:|--:|--:|--:|--:|
> > | Macro-P$_{FW}$  | 753.32  | 6 | 799.90  | 6 | 754.67  | 5 |
> > | Macro-R$_{FW}$  | 645.35  | 4 | 656.32  | 4 | 605.59  | 4 |
> > | Macro-F1$_{FW}$ | 541.44  | 5 | 537.67  | 5 | 606.54  | 6 |
> >
> > **AmazonCat:**
> > | method | time@3/s| iter.@3 | time@5/s | iter.@5 | time@10/s  | iter.@10|
> > |:----|--:|--:|--:|--:|--:|--:|
> > | Macro-P$_{FW}$  | 1181.99 | 4 | 840.41  | 2 | 1134.49 | 4 |
> > | Macro-R$_{FW}$  | 1112.24 | 4 | 1126.27 | 4 | 1081.46 | 4 |
> > | Macro-F1$_{FW}$ | 891.25  | 4 | 1454.04 | 9 | 885.40  | 4 |

---

> > > ### Comment · Reviewer_gTuy · 2023-11-22
> > > **Response to rebuttal**
> > >
> > > Thank you for the response. I have read all the rebuttals and reviews. I am satisfied with the clarifications and additional experiments to be added to revised version of the manuscript. The work gives nice theoretical contributions to at-$k$ constraint multi-label classification with practical optimisation strategy with competitive empirical results. I'll raise my score accordingly.

---

### Comment · Area_Chair_5gMc · 2023-11-20
**reviewers, please acknowledge the responses from the authors**

Dear reviewers: Please read the replies from the authors carefully, and submit your reactions. Please be open-minded in deciding whether to change your scores for the submission, taking into account the explanations and additional results provided by the authors.

Thank you!

---

### Meta-Review · Area_Chair_5gMc · 2023-12-15

**Metareview:**

All reviewers are positive about this paper, which is of high quality.

**Justification For Why Not Higher Score:**

The contribution is quite technical, so it will be of interest to a limited audience.

**Justification For Why Not Lower Score:**

All reviewers agree that the paper is a clear contribution.

---

### Decision · Program_Chairs · 2024-01-16

Accept (poster)